# Time Series Forecasting Through the Lens of Dynamics

**Alexis-Raja Brachet** [1] [2]   **Pierre-Yves Richard** [2]   **Céline Hudelot** [1]

## Abstract

While deep learning is facing an homogenization across modalities led by Transformers, they are still challenged by shallow linear models in the time series forecasting task. Our hypothesis is that models should learn a direct link from past to future data points, which we identify as a learning dynamics capability. We develop an original `PRO-DYN` nomenclature to analyze existing models through the lens of dynamics. Two observations thus emerge: **1.** under-performing architectures learn dynamics at most partially, **2.** the location of the dynamics block at the model end is of prime importance. Our systemic and empirical studies both confirm our observations on a set of performance-varying models with diverse backbones. We propose a simple plug-and-play methodology guiding model designs and improvements.

## 1. Introduction

In recent years, data-driven models, especially deep learning ones, have successfully processed data in various tasks. While specific models were designed regarding the modality, we face a model homogenization (Bommasani et al., 2021): Transformer models (Vaswani et al., 2017) originating from the text modality are becoming state-of-the-art (SOTA) across various fields (Veličković et al., 2018; Dosovitskiy et al., 2021; Chen et al., 2023). Boundaries between modalities are vanishing. However, in the specific case of time series forecasting, these usual models are still challenged by quite simple ones (Zeng et al., 2023; Xu et al., 2024; Tan et al., 2024).

Most of the text-based models, including RNNs (Elman, 1990), LSTMs (Hochreiter & Schmidhuber, 1997), Trans-

formers (Vaswani et al., 2017), or more recently, State-Space Models (SSMs) (Gu & Dao, 2024), follow the sequence-to-sequence paradigm, turning one sequence into another sequence. They achieved significant success in text generation, which fits quite well to this modality: the larger the model capacity, the better the performance, as seen in (Hoffmann et al., 2022). As time series forecasting (TSF) can also be seen as a sequence-to-sequence task, or more precisely, a series-to-series task, the previous text-based models have naturally been adapted to it (Zhou et al., 2021; 2022a). However, it has been shown that these models are well challenged by shallow linear ones, among which the Long-term time series forecasting (LTSF)-Linear models (Zeng et al., 2023) or FITS (Xu et al., 2024). These simple models map the input and output data by a linear layer (with eventual normalization or decomposition). Recent SOTA approaches are now built on basic linear models, with complex backbones serving as pre-processing units (Nie et al., 2023; Liu et al., 2024b; Hu et al., 2024a; Wang et al., 2025a).

These observations raise two points: **a.** generating time series is inherently different from generating text, even though they have a similar sequential structure;[1] **b.** the problem does not seem to come from using text-based architectures but from the way we use them on this kind of data. To our knowledge, no systematic study to explain these observations has been proposed in the literature. Previous work on Transformer failure in TSF (Ke et al., 2025) focused only on the attention mechanism, without explaining recent Transformer-based model achievements (Nie et al., 2023; Liu et al., 2024b).

Text-based models were designed to replicate the text generation mechanism (Rayner, 1998; Cheng et al., 2016). **We argue that TSF models should replicate the time series generation mechanism**. This mechanism is, in a majority of fields, modeled as a data-evolution law called a dynamical system, also termed as **physical dynamics** in this paper, a priori known in physics (Raissi et al., 2019; Li et al., 2021; Kovachki et al., 2023) or economics (Liu et al., 2024a), or estimated a posteriori (Shojaee et al., 2025). It legitimates the modeling of time series evolution based on their underlying dynamics. **We thus hypothesize that TSF models**

---

[1]MICS, CentraleSupélec, Université Paris-Saclay, France [2]CentraleSupélec, IETR UMR CNRS 6164, France. Correspondence to:
Alexis-Raja Brachet <alexisraja.brachet@centralesupelec.fr>.

*Proceedings of the 43rd International Conference on Machine Learning*, Seoul, South Korea. PMLR 306, 2026. Copyright 2026 by the author(s).

[1]In classification or anomaly detection, we don't observe this phenomenon, see results in (Xu et al., 2024).

**should be able to learn these physical dynamics**. We observe that TSF models differ in how they propagate past observations into future predictions. We argue that this propagation mechanism, which we term as the **model dynamics**,[2] is then critical to their performance. While some models rely on fixed placeholders as Informer (Zhou et al., 2021) or non-learnable functions as FEDformer (Zhou et al., 2022a), others learn a direct mapping from observations to forecasts. This distinction seems to explain why models like iTransformer (Liu et al., 2024b) beat shallow baselines, and motivates our focus on learning model dynamics as a key property of TSF models.

In this context, we identify three families of TSF models, based on their design and learning principles: **pattern-recognition models**, which focus on identifying and extrapolating temporal patterns without explicitly modeling the underlying physical dynamics, either from a single dataset as Informer or iTransformer (family 1), or from a large corpus of datasets as Chronos (Ansari et al., 2024) or Toto (Cohen et al., 2025) (family 2); and **dynamics-based models** which are explicitly designed to learn the underlying physical dynamics governing the time series, as Koopa (Liu et al., 2023) or Attraos (Hu et al., 2024a) (family 3). We restrict our study to family 1 models, as they present a foundation setting to extend this work to the other families. Indeed, they present characteristics we also find in foundation models (transformer backbone and computing blocks configuration), and this first study provides a common framework based on dynamics to compare non-dynamics-based models (families 1 and 2) with family 3 ones.[3]

We develop an original nomenclature named `PRO-DYN`. It enables making explicit how dynamics is involved in a model (`DYN` function), surrounded by pre- and post-processing units (`PRO` functions). We identify LTSF-Linear models (Zeng et al., 2023) as a relaxed version of discrete linear time-delay dynamical systems. We then perform a systemic study of existing TSF models (Qiu et al., 2024). We derive two main observations: **1.** under-performing models have no (or partial) learnable dynamics modeling (supporting our hypothesis); **2.** SOTA architectures among family 1 do learn a dynamics (again supporting our hypothesis), combining deep blocks as pre-processing units and a dynamics block at the end as the predictor, denoted as a `PRE-DYN` configuration, giving clues to model design considerations.

Motivated by the identification of linear layers for prediction to linear dynamical systems, we first incorporate linear dy-

namics, without any structural hyperparameter modification, into targeted models that have no or partial dynamics modeling capabilities: two Transformer-based ones, Informer and FEDformer, the CNN-based MICN (Wang et al., 2023a), and the SSM-based FiLM (Zhou et al., 2022b). Our experiments show tangible performance improvements, which support that learnable dynamics modeling capabilities drive the performance. Then, to study the second observation, we add a Linear dynamics layer at the entry of recent well-performing models, iTransformer, PatchTST (Nie et al., 2023), and Crossformer (Zhang & Yan, 2023) to employ them as post-processing units, again without any structural hyperparameter modification. Our experiments show that pre-processing-like architectures are the best choice as they take better advantage of longer look-back windows.

We decide to focus on linear layers as `DYN` functions for model dynamics for two reasons. First, they are a general computation unit and carry little explicit dynamical inductive bias compared, for instance, to autoregressive mechanisms; thus, they constitute a minimal and relatively unfavorable setting for a dynamics-based interpretation, making the emergence of a physical dynamical bias in this setting challenging. Then, they are sufficient to characterize the main architectural differences highlighted in family 1 by `PRO-DYN`.

## 2. Related Work

**TSF by adapting text-based Transformers** The main concern when adapting text-based models for time series forecasting was efficiency; the development of the LogSparse attention was the pioneer work in Transformer-based TSF (Li et al., 2019), but it kept the slow autoregressive process. The most influential work became Informer, where they defined the ProbSparse attention and generated predictions in one forward pass (Zhou et al., 2021). Models like Autoformer (Wu et al., 2021), FEDformer (Zhou et al., 2022a), and Non-stationary Transformers (Liu et al., 2022b), inherited from Informer computations: initialize a decoder with a simple non-learnable prediction (mean or zero-padding). Later, these models were beaten by the LTSF-Linear models (Zeng et al., 2023). The focus of complex deep model failure has been on attention mechanism (Zeng et al., 2023; Ke et al., 2025), but recent well-performing models are still attention-based (Nie et al., 2023; Liu et al., 2024b; Cohen et al., 2025). Our work focuses on the learning dynamics capabilities of TSF models, a possible major performance driver.

**Models inheriting from LTSF-Linear models** LTSF-Linear models (Zeng et al., 2023) were introduced in earlier works on Direct Multi-Step (DMS) forecasting (Chevillon, 2005), again for simplicity and efficiency, avoiding accumu-

---

[2]In the following, the term *dynamics* employed alone refers to a *model dynamics*.

[3]We apply our nomenclature to both dynamics-based and foundation models in Appendix D to open the discussion on future research axis.

lated errors from Iterated Multi-step (IMS). They have been automatically adopted by the vast majority of diverse models for their performance and efficiency, as in TiDE (Das et al., 2023), iTransformer (Liu et al., 2024b), or Attraos (Hu et al., 2024a), beating previous LTSF-Linear models. To our knowledge, no systemic justification has been proposed to support the integration of Linear functions. Our work proposes one based on Linear learning dynamics capabilities.

**Models in TSF based on physical dynamics**   A group of works on TSF assumes that the studied data is governed by an evolution law, i.e., a physical dynamics (family 3). Thus, it draws on theories from the dynamical field, such as the Koopman theory (Koopman, 1931), which shows that an infinite-dimensional space exists where the physical dynamics becomes linear. These methods (Liu et al., 2023; Gupta et al., 2024; Wu et al., 2025) then project the data into a latent space, in which they proceed forward in time with an approximation of the Koopman operator, which is linear, and then return to the initial space. Other models build on other theories, such as Takens' theorem (Takens, 1981) from the chaotic field (Hu et al., 2024a; Majeedi et al., 2025) or on state-space representations (Zhang et al., 2023), or try to align diffusion principles and temporal evolution (Cachay et al., 2023; Guo et al., 2025). These works support the idea of analyzing the TSF task through the lens of dynamics, with results in (Hu et al., 2024b) aligning with our hypothesis that physics-inspired considerations are specifically relevant for time series generating tasks. We leave the study of these models for future work, where we plan to compare various `DYN` functions and their impact on the design and performance.

## 3. Systemic Analysis Through the Lens of Dynamics

We introduce the task with the associated notations and definitions used in the rest of the paper.

**Time series and time interval**   We consider a time series $\mathbf{X} = \{x_d(t_1), \ldots, x_d(t_L)\}_{d=1}^{D} \in \mathbb{R}^{L \times D}$ which is the historical data of $D$ variates along $L$ regularly sampled timestamps $t_i \in \mathbb{R}^{+}$ with $t_i < t_j, \forall i, j \in \{1, \ldots, L\} | i < j$. A time interval of such time series $\mathbf{X}$ is the smallest interval containing $\{t_1, \ldots, t_L\}$, which is $\mathcal{T}_{\mathbf{X}} = [t_1, t_L]$, called the historical time interval.

**TSF task**   The time series forecasting task of $\mathbf{X}$ is to infer the $H$ future timestamps $\mathbf{Y} = \{x_d(t_{L+1}), \ldots, x_d(t_{L+H})\}_{d=1}^{D}$ based on the $L$ historical ones, i.e. $\mathbf{X}$. We denote $\mathcal{T}_{\mathbf{Y}} = [t_{L+1}, t_{L+H}]$ the prediction time interval.

**Definition 3.1. (Dynamics in TSF models)** We define the dynamics of a TSF model, called **model dynamics**, as its

mechanism for propagating past observations into future predictions. A model has **learning dynamics capabilities** when its mechanism is **learnable** and **directly maps** past data points to future ones. A model does not have such capabilities when its mechanism is indirect or not learnable.

**Definition 3.2. (Dynamical systems)** A dynamical system describes the evolution over time of a state denoted as $\mathbf{x} \in \mathbb{R}^{D}$. It relates past observations to future ones by an evolution operator $F$. Imposing structures on $\mathbf{x}$ and $F$ can represent specific properties of the dynamical system.

**Definition 3.3. (Discrete time-delay dynamical systems)** Based on a current system state $\mathbf{x}(t) \in \mathbb{R}^{D}$ at time $t \in \mathbb{R}^{+}$, let suppose we sample at a rate of $\Delta \in \mathbb{R}^{+*}$ such as $\mathbf{x}(t_n) = \mathbf{x}(n\Delta), n \in \mathbb{N}$. $\mathbf{x}$ is governed by a discrete time-delay dynamical system if there exists a map $F : \mathbb{R}^{K \times D} \to \mathbb{R}^{D}$, $K \in \mathbb{N}$, such that:

$$\mathbf{x}(t_n) = F(\mathbf{x}(t_{n-1}), \ldots, \mathbf{x}(t_{n-K}))$$

**Definition 3.4. (Discrete linear time-delay dynamical systems)** Without loss of generality, let us suppose $D = 1$. Keeping the same notations as above, if $F$ is linear, we can define $M \in \mathbb{R}^{L \times L}$, $L \in \mathbb{N}$ with $L \geq K$ such that:

$$[\mathbf{x}(t_n), \ldots, \mathbf{x}(t_{n-L+1})]^T = M[\mathbf{x}(t_{n-1}), \ldots, \mathbf{x}(t_{n-L})]^T$$

$M$ is the discrete dynamical matrix defining the system evolution of $L$ successive observations. For $D > 1$, $M \in \mathbb{R}^{(L \times D) \times (L \times D)}$ would be defined as a block matrix.

A basic example of such a system is provided in Appendix C. In the following, we suppose the lookback window is wide enough such that $L \geq K$.

### 3.1. The `PRO-DYN` Nomenclature

Our nomenclature is based on how computations are performed along the time axis in TSF models.

We consider a function $f$ mapping a time series from a time interval $\mathcal{T}_{\mathcal{E}}$ to $\mathcal{T}_{\mathcal{F}}$. Depending on the task assigned to $f$, it involves temporal relations between $\mathcal{T}_{\mathcal{E}}$ and $\mathcal{T}_{\mathcal{F}}$. In this paper, we rely on the popular Allen's interval algebra (Allen, 1983) that defines relations between two time intervals (illustrated in Appendix A). We introduce the notions of `PRO` (PROcessing) and `DYN` (DYNamics) function, based on Allen's temporal interval relations: $f$ is `PRO` if and only if $\mathcal{T}_{\mathcal{E}}$ **contains, started by, finished by, or equals** $\mathcal{T}_{\mathcal{F}}$ (not moving forward in time). $f$ is `DYN` if and only if $\mathcal{T}_{\mathcal{E}}$ **starts, overlaps, meets, or before** $\mathcal{T}_{\mathcal{F}}$ (moving forward in time).[4] Non-learnable invertible functions employed both

---

[4] Relying on Allen's algebra provides a consistent and rigorous way to characterize how models manipulate temporal information. It provides a precise formal criterion for the `PRO`/`DYN` distinction by specifying whether a computational block preserves a temporal interval or advances it.

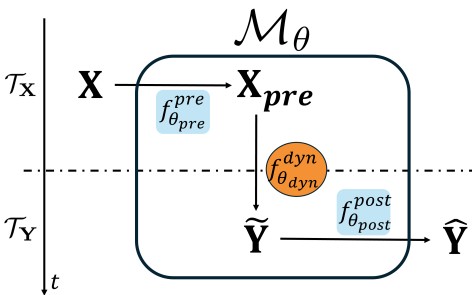

*Figure 1.* PRO and DYN functions illustrated in the processing chain of a TSF model $\mathcal{M}_\theta$ where "$\mathcal{T}_\mathbf{X}$ **before** $\mathcal{T}_\mathbf{Y}$". PRO functions are framed and blue while DYN function is encircled and orange. Solid lines represent the main data flow. Skip connections are not drawn for better clarity.

at the entry and inverted in the end (e.g. normalization) are not taken into account in our nomenclature.

Based on time-evolution considerations, we propose to introduce three types of functions that can be used to decompose any model designed for a TSF task. More precisely, we consider that any model $\mathcal{M}_\theta$ designed for a TSF task, with learnable parameters $\theta$, that takes data points in the historical time interval $\mathcal{T}_\mathbf{X}$ and outputs predictions $\hat{\mathbf{Y}}$ in the future $\mathcal{T}_\mathbf{Y}$, can be decomposed as follows:

$$\mathcal{M}_\theta : \mathbf{X} \xrightarrow[f^{pre}_{\theta_{pre}}]{} \mathbf{X}_{pre} \xrightarrow[f^{dyn}_{\theta_{dyn}}]{\oplus\{\mathbf{X}\}} \tilde{\mathbf{Y}} \xrightarrow[f^{post}_{\theta_{post}}]{\oplus\{\mathbf{X},\mathbf{X}_{pre}\}} \hat{\mathbf{Y}} \quad (1)$$

where $f^{dyn}_{\theta_{dyn}}$ (in orange), a DYN function, defines $\mathcal{M}_\theta$ dynamics performing a prediction going from $\mathcal{T}_\mathbf{X}$ to $\mathcal{T}_\mathbf{Y}$ (or $\mathcal{T}_\mathbf{X} \rightarrow \mathcal{T}_\mathbf{X} \cup \mathcal{T}_\mathbf{Y}$ in a start/overlap case); $f^{pre}_{\theta_{pre}}$, $f^{post}_{\theta_{post}}$, two PRO functions, are pre- and post- (relatively to $f^{dyn}_{\theta_{dyn}}$) processing functions, performing computations while staying in their input time interval. $\oplus\{\}$ over arrows illustrate eventual skip connections. We illustrate our framework in Figure 1.

Based on this, we introduce our original PRO-DYN nomenclature. For any TSF model : **1.** we decompose it as a composition of PRO and DYN functions; **2.** we identify the nature of the DYN function;

From the PRO-DYN nomenclature, we first analyze the LTSF-Linear models (Zeng et al., 2023), the basic models challenging deep complex ones, through the lens of dynamics.

## 3.2. LTSF-Linear Dynamics

Without loss of generality, let us suppose $H \leq L$ and $D = 1$ (see Appendix B for the $H > L$ case). Let suppose the true physical dynamics is $[\mathbf{x}(t_n), \ldots, \mathbf{x}(t_{n-L+1})]^T = M[\mathbf{x}(t_{n-1}), \ldots, \mathbf{x}(t_{n-L})]^T$

and denote $\mathbf{Y} = [\mathbf{x}(t_{L+1}), \ldots, \mathbf{x}(t_{L+H})]^T$. We then have : $\mathbf{Y} = (M^H)_{-H:,:}\mathbf{X}$ with $M^H$ is $M$ repeated $H$ times, where $(M^H)_{-H:,:} \in \mathbb{R}^{H \times L}$ corresponds to $M^H$ where only the last $H$ rows are kept (following Python notation).

LTSF-Linear models are the composition of: a non-learnable invertible pre-processing step $g_{pre} : \mathbf{X} \mapsto \mathbf{X}_g$ (where $g_{pre}$ is identity for Linear, normalization for NLinear, or seasonal-trend decomposition[5] for DLinear), a learnable DYN function Linear$_\theta$ - highlighted in orange -, and a non-learnable post-processing step $g_{post} = g^{-1}_{pre}$ such as:

$$\hat{\mathbf{Y}} = g^{-1}_{pre} \circ \boxed{\text{Linear}_\theta} \circ g_{pre}(\mathbf{X}) \quad (2)$$

$$= g^{-1}_{pre}(\boxed{W_\theta}\,\mathbf{X}_g + \boxed{b_\theta}) \quad (3)$$

where $W_\theta \in \mathbb{R}^{H \times L}$ and $b_\theta \in \mathbb{R}^{H \times D}$ - a $\mathbb{R}^H$ vector repeated $D$ times along the variate dimension - are the parameters of Linear$_\theta$ (for DLinear, $g^{-1}_{pre}$ is the sum of seasonal and trend linear layer outputs). We can identify $W_\theta$ to $(M^H)_{-H:,:}$: two matrices with the same dimensions having the same temporal impact on a group of historical values. By doing so, **LTSF-Linear models can be viewed as a relaxed version of discrete linear time-delay dynamical systems**, as there are no constraints on the learnt coefficients (system identification is then difficult, while dynamics-based models are designed to do so). The bias term in the linear layer can be seen as a corrective step due to the absence of constraints. Thus, the relaxed identification of LTSF-Linear models to such dynamical systems would explain their performance.[6]

## 3.3. TSF Models Through the PRO-DYN Nomenclature

The goal here is to identify features from the PRO-DYN nomenclature driving model performance. We analyze deep models tested in the benchmark (Qiu et al., 2024) (chosen for its diversity of datasets). We end up with Table 1, keeping the same row-order performance as in the benchmark on the multivariate TSF task. When there are multiple linear layers in a model, if a specific linear layer is explicitly defined as the prediction layer, it is the DYN function. Other linear layers along the time dimension are considered as processing layers. They can be seen as up- or down-sampling steps (as in iTrasformer or PatchTST). Linear layers along the variate dimension are usual projection functions.

There are from Table 1 two *performance-based* groups: models better ($\uparrow$) than NLinear (chosen as the reference as

---

[5]The trend component $\mathbf{T}$ is a moving average over the input and the seasonal component $\mathbf{S}$ is the input without the trend component, such as $\mathbf{X} = \mathbf{S} + \mathbf{T}$.

[6]In LTSF-Linear models, they go forward in time without mixing information between variates. They usually either learn one global dynamics for each variate or learn one dynamics per variate, falling into the $D = 1$ case.

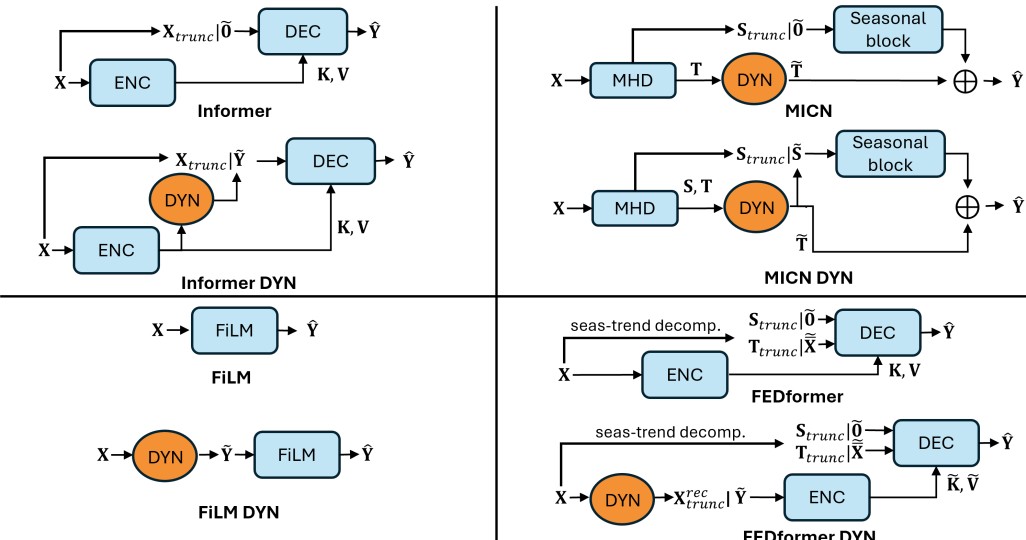

*Figure 2.* RQ1 models with now full learnable dynamics capabilities. `DYN` is linear dynamics layer, ENC-DEC is encoder-decoder, MHD is multi-scale hybrid decomposition (defined in MICN, see (Wang et al., 2023a)). $\mathbf{A}|\mathbf{B}$ means $\mathbf{A}$ concatenated with $\mathbf{B}$ along the time axis, $\hat{\mathbf{Y}}$ is the output, tilde refers to intermediate variables in the prediction time interval, $\tilde{\mathbf{Y}}$ an intermediate prediction from a `DYN` function, $\overline{\mathbf{X}}$ is the mean of $\mathbf{X}$, *trunc* subscript corresponds to input start tokens.

it is the best performing simple model), and models worse ($\downarrow$) than it. From the `PRO-DYN` nomenclature, models $\uparrow$ have two features (identified with green color) in common: a **complete learnable `DYN` function** and a **`PRO` function for pre-processing only**, while in the second group, the main shared features (identified with magenta color) are an, at most **partially**, non-learnable `DYN` function and learnable `PRO` functions for both pre- and post-processing.

We thus identify, directly in Table 1, two *feature-based* groups: in green/bold, models with two green features and, in magenta/italic, models with at least one magenta feature. They almost coincide with the performance-based groups. We thus derive two observations: **1.** a (partially) non-learnable `DYN` function drowns the performance, and **2.** a learnable `PRO` function for pre-processing just before the final `DYN` function drives the performance.

We derive these two observations into two research questions (RQ) to validate them experimentally: (**RQ1**) Can we enhance model performance by adding a full learnable dynamics?; (**RQ2**) Is (Pre-processing)-`DYN` the best-performing configuration? If so, why?

**(RQ1) Dynamics addition** We choose to study Informer (Zhou et al., 2021), FiLM (Zhou et al., 2022b), MICN (Wang et al., 2023a), and FEDformer (Zhou et al., 2022a), for their diverse performance, `DYN` functions, and backbones. We incorporate full learnable dynamics, always after the eventual non-learnable normalization step, for prediction in these models by adding a linear `DYN` layer (`Linear`$_\theta$ in Section 3.2), while keeping the original struc-

*Table 1.* TSF deep models (family 1) through the `PRO-DYN` nomenclature ranked by performance. $\uparrow$ (resp. $\downarrow$) are models better (resp. worse) than NLinear. 0-pad. stands for 0-padding, discr. for discretization, NS for Non-stationary. Colors correspond to features identified as driving (green) or dragging (magenta) performance on the TSF task. In Appendix D, Table 4 is provided adding reference and the backbone type of these models; Table 5 is provided, including both foundation and dynamics-based models (family 2 & 3), showing the generality of our nomenclature.

| Model | Complete learnable dynamics (RQ1) | Config. (RQ2) | DYN function |
|---|:---:|:---:|:---:|
| **DUET** $\uparrow$ | ✓ | PRE-DYN | Linear |
| **PDF** $\uparrow$ | ✓ | PRE-DYN | Linear |
| **Pathformer** $\uparrow$ | ✓ | PRE-DYN | Linear |
| **iTransformer** $\uparrow$ | ✓ | PRE-DYN | Linear |
| **PatchTST** $\uparrow$ | ✓ | PRE-DYN | Linear |
| **Crossformer** $\uparrow$ | ✓ | PRE-DYN | Linear |
| **TimeMixer** $\uparrow$ | ✓ | PRE-DYN | Linear |
| NLinear | ✓ | DYN | Linear |
| *FITS* $\downarrow$ | ✗ | PRE-DYN | Linear & 0-pad. |
| *TimesNet* $\downarrow$ | ✓ | PRE-DYN-POST | Linear |
| **Triformer** $\downarrow$ | ✓ | PRE-DYN | Linear |
| *FEDformer* $\downarrow$ | ✗ | PRE-DYN-POST | Mean & 0-pad. |
| *MICN* $\downarrow$ | ✗ | PRE-DYN-POST | Linear & 0-pad. |
| *NS Transformer* $\downarrow$ | ✗ | PRE-DYN-POST | 0-padding |
| *FiLM* $\downarrow$ | ✗ | PRE-DYN-POST | Legendre discr. |
| *Informer* $\downarrow$ | ✗ | PRE-DYN-POST | 0-padding |

tures (see Figure 2): for Informer, a Transformer-based model, we feed the decoder with the encoder output, which is processed by a linear `DYN` layer. It replaces the zero-padding. For FiLM, an SSM-based model, we add a linear `DYN` layer at the model entry. FiLM turns into a `PRO` post-processing block. For MICN, a CNN-based model, we feed the seasonal block with the seasonal component processed by the trend block, which is a linear `DYN` layer, replacing the zero-padding. For FEDformer, a Transformer-based model,

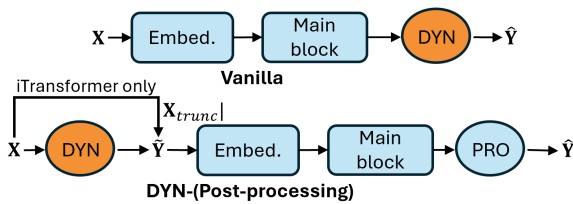

*Figure 3.* RQ2 models general derivation in their post-processing configuration. For iTransformer, part of the input is concatenated to the `DYN` output. Embed. stands for embedding layer.

we add a linear `DYN` layer before the encoder embedding layer, while recomputing the end of the input $\mathbf{X}_{trunc}$ to fit the original decoder temporal embedding size. The decoder input (zero-padding for the seasonality $\mathbf{S}$ and input mean for the trend $\mathbf{T}$) is not changed, but Keys ($\tilde{\mathbf{K}}$) and Values ($\tilde{\mathbf{V}}$) are now computed from initial predictions performed by the added learnable `DYN` function.[7]

Better performances of the `DYN` versions of the chosen models would answer positively to RQ1. The variety of studied models and locations to incorporate dynamics would validate the generality of dynamics considerations.

**(RQ2) `DYN`-(Post-processing) configuration**  We identify three well-performing models for their diversity in time consideration and structure: iTransformer (Liu et al., 2024b), an encoder-only model where time and variate dimensions are inverted, considers time as a latent dimension, losing its sequential property; PatchTST (Nie et al., 2023), a patching-based method, thus respecting time sequentially, in an encoder-only fashion; Crossformer (Zhang & Yan, 2023), also a patching-based method, with an encoder-decoder serving as a `PRO` function. In each model, illustrated in Figure 3, we add a learnable linear `DYN` layer just before the embedding one, without removing the linear `DYN` layer at the end, which becomes a linear `PRO` layer (not possible to remove it while keeping the same model design and hyperparameters): the main computation block acts now as a post-processing function. Only the `DYN` output feeds PatchTST and Crossformer, while part of the input is concatenated to it to feed iTransformer. A performance drop in the modified models would answer positively to RQ2.[8] A comparison between

---

[7]We studied multiple ways to add the linear `DYN` layer. The `DYN` layer takes an input living in the historical time interval. For Informer and FEDformer, we then had two options: before or after the encoder. We tested both designs and selected the best one for each configuration. For FiLM, we also had the same two options: before or after the whole FiLM block. The latter option involved adjusting the hyperparameters of the original model too much, thus making comparison with vanilla FiLM meaningless. We thus put the `DYN` layer before.

[8]In the case of the `PRE-DYN` configuration, the linear `DYN` layer maps observations in the latent space to predictions in the original space. The linear DYN layer returns to the input space

*Table 2.* Evolution through the `PRO-DYN` nomenclature of the studied models in our research questions. In RQ1, we add complete learnable dynamics capabilities, asserting whether it improves the performance. In RQ2, we add a linear `DYN` layer at the model entry, turning `PRE-DYN` models previously into `DYN-POST` ones. A performance drop would validate RQ2.

| Models | Vanilla | | Modified | |
|---|---|---|---|---|
| | Complete learn. dyn. | Config. | Complete learn. dyn. | Config. |
| Informer | ✗ | PRE-DYN-POST | ✓ | PRE-DYN-POST |
| FiLM | ✗ | PRE-DYN-POST | ✓ | DYN-POST |
| MICN | ✗ | PRE-DYN-POST | ✓ | PRE-DYN-POST |
| FEDformer | ✗ | PRE-DYN-POST | ✓ | DYN-POST |
| iTransformer | ✓ | PRE-DYN | ✓ | DYN-POST |
| PatchTST | ✓ | PRE-DYN | ✓ | DYN-POST |
| Crossformer | ✓ | PRE-DYN | ✓ | DYN-POST |

the modified models in the RQs and their vanilla version is proposed in Table 2.

# 4. Experiments

We conduct extensive experiments to answer RQ1 and RQ2. Modified models in RQ1 are referred to as "`DYN` added models", while ones in RQ2 are referred to as "post-processing models". Original models are referred to as "vanilla".

## 4.1. Experimental Setup

**Datasets**  We consider TSF on the 25 datasets of the TFB benchmark (Qiu et al., 2024), with 4 forecasting horizons each, including the well-established ETTs, Exchange, Weather, Electricity, ILI, Traffic, and Solar datasets (Wu et al., 2021). Details on datasets are shown in Appendix E.

**Settings**  To evaluate the TSF task, we compute the Mean Square Error (MSE) and Mean Average Error (MAE) across each dataset and forecasting horizon (200 scores per model) for the modified models and compare them to the vanilla results obtained in the TFB benchmark. We keep the same architecture hyperparameters as their vanilla versions. We only adjust by hand learning hyperparameters (epochs, learning rate, patience), and also test them on their vanilla versions for fair comparison. Each configuration can be found in the code,[9] and the implementation details are in Appendix F. Raw results can be found in Appendix G.1, and prediction visualizations in Appendix H.

**Results**  We count the number of cases where the modified models are better, equal (iso), worse by at most 1% (low degradation), or worse by at least 1%, than their vanilla version. For MSE and MAE, the lower, the better. Resulting distributions are shown in Figure 4. For RQ1 `DYN` added

---

as it progresses forward in time. We can view the `PRE` block as linearizing the dynamics between historical latent variables and the predicted ones. System identification is thus challenging.

[9]https://github.com/ARBrachet/PRO-DYN/

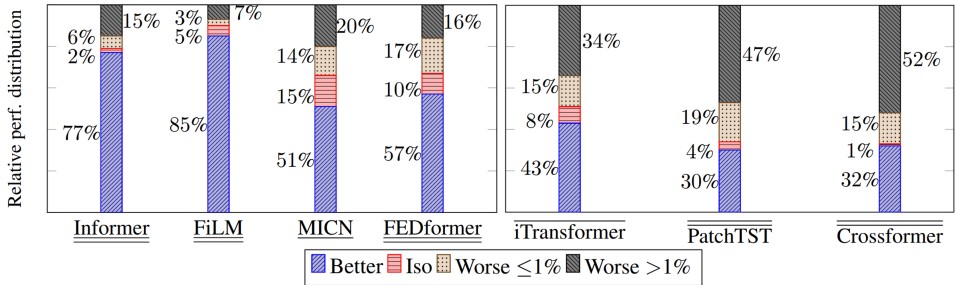

*Figure 4.* Performance distribution of RQ1 (left) and RQ2 (right) models. A model name is underlined (resp. double-underlined) when the `DYN` added model is statistically better than its vanilla version on either MSE or MAE (resp. both). Similarly, it is overlined (resp. double-overlined) when the vanilla model is statistically better than the post-processing version on one (resp. both) metric.

models, we compute the p-values of the unilateral Wilcoxon test to assess if the MSE and MAE are lower than the vanilla versions with statistical significance (p-value $< 0.05$). For RQ2 models, we perform the opposite test to assess if the vanilla versions are better than the post-processing ones with statistical significance. In addition, for RQ1, we compute in Table 3 the average score normalized by NLinear to measure the quantitative impact of the `DYN` addition, while comparing each model to the NLinear baseline. Detailed results can be found in Appendix G.2.

### 4.2. First Analysis

**RQ1**  Results seem to **support to have full learning dynamics capabilities** for TSF. Indeed, from Figure 4, all `DYN` added models **are better or comparable in more than** $80\%$ **of the cases** than their vanilla versions, and **better with statistical significance** on at least one metric. In particular, Informer and FiLM are greatly improved by the linear `DYN` layer addition. With just a data flow update, MICN gets better or equal scores $66\%$ of the time. Moving to Table 3, for each model, **the mean performance is better** with the `DYN` layer addition, supporting our model update. Moreover, FiLM `DYN` gets slightly better results than NLinear on average. However, `DYN` models (except FiLM) are still worse than NLinear - normalized scores below zero -, where possible reasons are: keeping the same hyperparameters is constraining, or `DYN` models are not in a pre-processing configuration (not feasible to do so while keeping the same original architecture).

**RQ2**  Results from Figure 4 **support the (Pre-processing)-`DYN` configuration** as PatchTST and Crossformer in the `DYN`-(Post-processing) version **get worse results with statistical significance**. However, post-processing iTransformer is only statistically worse on one metric, with better or equal results in $51\%$ of the time, supporting the possibility of using such models as post-processing blocks. As PatchTST and Crossformer take into account the time sequential aspect in their

*Table 3.* Comparison of average performance normalized by NLinear scores: $\frac{\text{score(NLinear)} - \text{score(Model)}}{\text{score(NLinear)}}$, where score is MSE or MAE, for each dataset and forecasting horizon. Best score between `DYN` added model and its vanilla version is in **bold**. Some outliers are removed for consistency (see Appendix G.2 for further details).

| Model | Informer | FiLM | MICN | FEDformer |
|---|---|---|---|---|
| DYN added | **−0.228** | **0.006** | **−0.164** | **−0.360** |
| Vanilla | −0.333 | −0.036 | −0.176 | −0.398 |

computations, they are more responsive to modifications with a temporal meaning, explaining the difference with iTransformer in response to the `DYN` layer addition.

**Intermediate conclusion**  Overall, current results seem to answer positively to both RQs. However, adding a linear layer comes with two side effects: a slight parameter addition and data length modification, which have an impact on performance. The modified models (except Informer and MICN) process inputs of different lengths than vanilla ones. We thus conduct an additional study to identify the performance drivers.

### 4.3. Performance Driver Analysis

We perform additional experiments on two side effects of the `DYN` layer addition:

- **parameter addition** can be the principal performance driver in the RQ1 case. To isolate that side effect, we compare `DYN` added models to their `PRO` added version, in which the added linear `DYN` layer is replaced by a feed-forward one which does not change the time dimension, turning it into a linear `PRO` layer. MICN is excluded here as no layer is added. For Informer `PRO`, we either pad with zeros if $H > L$ or truncate the output if $H < L$ to fit the decoder input dimension. RQ2 vanilla models are still compared to their post-processing versions, as it is a worst-case scenario for (Pre-processing)-`DYN` configuration (they have less parameters than their modified version);

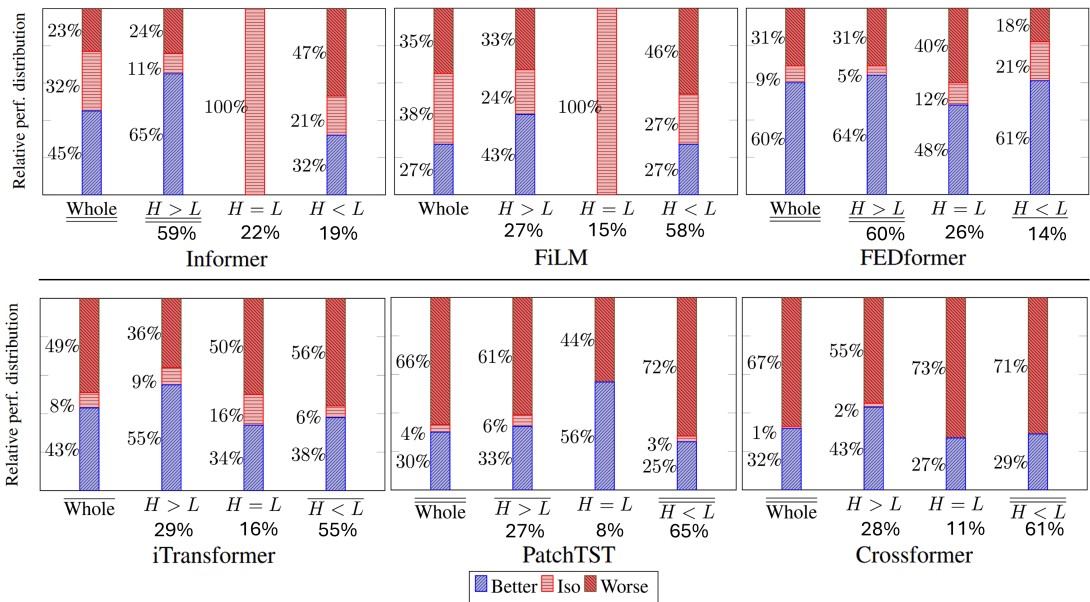

*Figure 5.* DYN added model performance distribution against their PRO version (up); Post-processing model performance distribution against their vanilla version (down), with setup conditioning. *Whole* bar is the overall distribution. Different from Figure 4, *Worse* $\leq 1\%$ and *Worse* $> 1\%$ are put together under *Worse*. Again, a setup is underlined (resp. double-underlined) when the DYN model is statistically better than the PRO one on either (resp. both) MSE or MAE. It is overlined (resp. double-overlined) when the vanilla version is statistically better than the post-processing one on one (resp. both) metric. Setup distribution for each model is shown under the proper case.

- **data length variation** due to differences between context and horizon size can be a performance driver. We thus condition the results on three possible setups: $(H > L)$, $(H = L)$, and $(H < L)$, and analyze distribution shifts. The setup repartition varies from one model to another ($L$ is a hyperparameter in the benchmark). For the $(H = L)$ setup, DYN and PRO are the same, except for FEDformer due to temporal embeddings and $\mathbf{X}_{trunc}^{rec}$ addition.

**Results** To perform the comparison, PRO versions are trained with the same hyperparameters as DYN ones. We count the number of cases when each DYN (resp. post-processing) model is better, equal (iso), or worse than its PRO (resp. vanilla) version on MSE and MAE. Global and conditioned distributions with cases when p-values are below 0.05 are shown in Figure 5. Detailed results are shown in Appendix G.3. Conditioned comparisons between DYN added models and their vanilla versions are presented in Appendix G.4. Prediction visualizations are shown in Appendix H.

**Preliminary observation** We observe from Figure 5 that **when models process greater data length, they get an advantage**. As the setup distribution is not uniform across each model, results are biased. In the following, if distributions are inverted from $(H > L)$ to $(H < L)$, the performance would then be driven by data length variation. If it is in

favor of DYN (resp. RQ2 vanilla) model even when it is disadvantaged, then the performance gain is mainly driven by the learnable dynamics capabilities (resp. dynamics block located at the end).

**RQ1** Results confirms that **the performance mainly comes from the dynamics**. Indeed, from Figure 5, for Informer and FEDformer, overall, DYN versions are statistically better than their PRO versions. For Informer, DYN and PRO are statistically similar on $(H < L)$ while DYN should be disadvantaged (fewer added parameters). For FEDformer, DYN is statistically better in both setups. For $(H = L)$, FEDformer DYN and PRO are statistically similar: the timestamp embedding, in line with (Zeng et al., 2023), doesn't influence the performance, and the recomputed $\mathbf{X}_{trunc}^{rec}$ doesn't give any data length advantage. On the contrary, for FiLM, the DYN version is not statistically better than the PRO one, with a symmetry in the distributions and a majority of $(H < L)$ disadvantaged cases for DYN: the performance against the vanilla version comes from the parameter addition. Indeed, the DYN layer **could be in conflict with its SSM encoding part**, which also learns a dynamics (Gu & Dao, 2024).

**RQ2** From the experiments, it emerges that predicting points based on a greater number of observations is an advantageous setup **when there are learning dynamics capabilities located at the model end**. Indeed, from Fig-

ure 5, an overall tendency emerges: vanilla models are similar to post-processing versions when ($H \geq L$) while surpassing them with statistical significance in the ($H < L$) setup. RQ1 `PRO` models, without learnable dynamics capabilities, don't surpass `DYN` models when ($H < L$), while they should be advantaged. RQ2 post-processing models don't take advantage of larger data length against vanilla ones (especially for PatchTST), confirming the superiority of the (Pre-processing)-`DYN` configuration. In addition, it can be interpreted as follows: in a `PRE-DYN` configuration, the `PRE` block serves a dynamics purpose, as it can be understood as learning a representation in which the temporal propagation becomes easier for a linear `DYN` function. This provides a possible explanation for why `PRE-DYN` performs better, whereas in `DYN-POST`, the `POST` block aims to refine a prediction towards a target, thus it is less related to dynamics-based considerations. This is consistent with (Hu et al., 2024b), which developed a physics-informed `PRE` module to improve performance.

## 5. Conclusion and Future Work

This work considers TSF models through the lens of dynamics. We propose the original `PRO-DYN` nomenclature, identify the dynamics defined by LTSF-Linear models, then assess which features can contribute the most to model performance, which emerge to be **1.** the ability to learn a dynamics, **2.** located at the end of the model. We perform experiments validating the hypothesis that **models should be able to learn dynamics**, showing they take **better advantage of a longer look-back window with learning dynamics capabilities**: they are powerful (Zeng et al., 2023).

In this paper, we focus on linear `DYN` funcions, family 1 architectures and the `PRE-DYN` configuration, constituting a foundational setting that underlies the temporal propagation mechanisms of more complex families as well, such as family 2 models with similar backbone and configuration (see Table 5 in Appendix D). This work targets this foundational layer as a first step to then extend the general `PRO-DYN` nomenclature to other configurations and `DYN` functions. Exploring the generation process of the considered datasets and how TSF models align their model dynamics with physical dynamics remains an open question for future study.

## Acknowledgements

We would like to thank the Agence de l'Innovation de Défense (AID) for funding and supporting the present work (DGA project N° 2023 29 0930), and the reviewers for their careful feedback for improving this work. The experiments were performed using computational resources from the Ruche platform of the "Mésocentre" computing center of Université Paris-Saclay, CentraleSupélec, and Ecole Normale Supérieure Paris-Saclay, supported by CNRS and Région Île-de-France.

## Impact Statement

This work is foundational research on already existing open-source models, tested on a public benchmark. This work has no impact on society at large, beyond aiming to get a better understanding of deep learning models. This work could reduce model energy consumption by guiding research on dynamics modeling.

**Responsible use of LLMs** In preparing this manuscript, we occasionally used suggestions from LLMs (GPT-5) to guide improvements in clarity, grammar, and overall readability. All scientific content, including experimental design, codebase, data analysis, results, and interpretations, is independently developed by the authors. LLMs are not involved in generating, modifying, or interpreting any experimental results, nor in producing code or analyses. Their use is strictly limited to selectively refining language to ensure clear and effective communication of our research.

**Reproducibility** For transparency and reproducibility, as detailed in the supplementary material in Appendix F, we base our code on the public repository developed by the TFB benchmark. We provide, along with the paper, the full code that includes the studied models and their hyperparameters used to evaluate them on the TSF task.

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

## A. Allen's Temporal Interval Relations

We propose a detailed visualization, in Figure 6, inspired by (Mate et al., 2019), illustrating the relations defined by Allen's interval algebra (Allen, 1983) considered in the paper. In our work, with the same notations as in Section 3.1, we study functions going from a temporal interval $\mathcal{T}_\mathcal{E}$ to $\mathcal{T}_\mathcal{F}$. If we denote `rel` an Allen's temporal interval relation, with $\mathcal{T}_\mathcal{E}$ `rel` $\mathcal{T}_\mathcal{F}$, we only consider relations `rel` that stay in the same temporal interval or go into the future (in accordance with the time series forecasting task), which are displayed in the figure below.

| Relation | Pictoral example $\quad t$ | Function type $f:\mathcal{T}_\mathcal{E} \to \mathcal{T}_\mathcal{F}$ |
|---|---|---|
| $\mathcal{T}_\mathcal{E}$ before $\mathcal{T}_\mathcal{F}$ | $\mathcal{T}_\mathcal{E}$ $\quad$ $\mathcal{T}_\mathcal{F}$ | DYN |
| $\mathcal{T}_\mathcal{E}$ equals $\mathcal{T}_\mathcal{F}$ | $\mathcal{T}_\mathcal{E}$ / $\mathcal{T}_\mathcal{F}$ | PRO |
| $\mathcal{T}_\mathcal{E}$ meets $\mathcal{T}_\mathcal{F}$ | $\mathcal{T}_\mathcal{E}$ $\quad$ $\mathcal{T}_\mathcal{F}$ | DYN |
| $\mathcal{T}_\mathcal{E}$ overlaps $\mathcal{T}_\mathcal{F}$ | $\mathcal{T}_\mathcal{E}$ $\quad$ $\mathcal{T}_\mathcal{F}$ | DYN |
| $\mathcal{T}_\mathcal{E}$ contains $\mathcal{T}_\mathcal{F}$ | $\mathcal{T}_\mathcal{E}$ / $\mathcal{T}_\mathcal{F}$ | PRO |
| $\mathcal{T}_\mathcal{E}$ starts $\mathcal{T}_\mathcal{F}$ | $\mathcal{T}_\mathcal{E}$ / $\mathcal{T}_\mathcal{F}$ | DYN |
| $\mathcal{T}_\mathcal{E}$ is started by $\mathcal{T}_\mathcal{F}$ | $\mathcal{T}_\mathcal{E}$ / $\mathcal{T}_\mathcal{F}$ | PRO |
| $\mathcal{T}_\mathcal{E}$ finished by $\mathcal{T}_\mathcal{F}$ | $\mathcal{T}_\mathcal{E}$ / $\mathcal{T}_\mathcal{F}$ | PRO |

*Figure 6.* Allen's temporal interval relations illustration. In our work, we only consider the relations that stay in the same temporal interval or go into the future. `DYN` cases are highlighted in orange. Input interval $\mathcal{T}_\mathcal{E}$ is in blue, output interval $\mathcal{T}_\mathcal{F}$ is in white. They are juxtaposed to better visualize how they relate along the time axis.

## B. Linear Dynamics when $H > L$

Following Section 3.2 with the same notations, let us suppose $H > L$. Supposing the true dynamics is $[\mathbf{x}(t_n),\ldots,\mathbf{x}(t_{n-L+1})]^T = M[\mathbf{x}(t_{n-1}),\ldots,\mathbf{x}(t_{n-L})]^T$. As $H > L \geq K$, we can extend the lookback window such that $[\mathbf{x}(t_n),\ldots,\mathbf{x}(t_{n-H+1})]^T = M_e[\mathbf{x}(t_{n-1}),\ldots,\mathbf{x}(t_{n-H})]^T$ (where $M_e$ is the extension of $M$ to a longer lookback window). Denoting $\mathbf{Y} = [\mathbf{x}(t_{L+1}),\ldots,\mathbf{x}(t_{L+H})]^T$, we then have : $\mathbf{Y} = (M_e^H)_{:,-L:}\mathbf{X}$ where $(M_e^H)_{:,-L:} \in \mathbb{R}^{H \times L}$ corresponds to $M_e^H$ where only the last $L$ columns are kept (following Python notation). We thus fall into the same situation as in Section 3.2.

## C. Basic Examples of Discrete Linear Time-Delay Dynamical Systems

Let $x \in \mathbb{R}$ denote a variable representing the state of a system governed by the following continuous evolution law:

$$\frac{dx}{dt}(t) = ax(t), a \in \mathbb{R}^*$$

If we discretize the above equation following the Euler method, with a sample rate of $\Delta \in \mathbb{R}^{*,+}$ such as $a\Delta \neq 1$ , we have:

$$x(t_n) = \frac{1}{1 - a\Delta}x(t_{n-1})$$

Here, $K = 1$ and with $L = 2$, we can define $M = \begin{pmatrix} (1 - a\Delta)^{-1} & 0 \\ 1 & 0 \end{pmatrix}$. If instead we employ the trapezoidal method so that $\frac{dx}{dt}(t_n) \approx \frac{x(t_{n+1}) - x(t_{n-1})}{2\Delta}$, then we have :

$$x(t_{n+1}) = 2a\Delta x(t_n) + x(t_{n-1})$$

Thus, $K = 2$ and with $L = 2$ we now have $M = \begin{pmatrix} 2a\Delta & 1 \\ 1 & 0 \end{pmatrix}$. These basic examples exhibit how the matrix $M$ emerges in a linear setup.

## D. Studied and Additional Models Analyzed Though the `PRO-DYN` Nomenclature

We include in this section an extended version of Table 1 which includes `PRO` backbone type and references. It shows that a majority of well-performing models from family 1 are Transformer-based. It also shows that the analysis seems to be independent from the `PRO` backbone type.

*Table 4.* Extended version of Table 1 TSF deep models through the `PRO-DYN` nomenclature, including `PRO` backbone type and references. Again, ↑ (resp. ↓) are models better (resp. worse) than NLinear. Tf. stands for Transformer, CNN for Convolution Neural Network, MLP for MultiLayer Perceptron, 0-pad. for 0-padding, discr. for discretization, NS for Non-stationary. Colors correspond to features identified as driving (green) or dragging (magenta) performance on the TSF task.

| Model | Complete learnable dynamics (RQ1) | Config. (RQ2) | `DYN` function | `PRO` backbone | Reference |
|---|---|---|---|---|---|
| **DUET** ↑ | ✓ | `PRE-DYN` | Linear | Transformer | (Qiu et al., 2025) |
| **PDF** ↑ | ✓ | `PRE-DYN` | Linear | Tf. & CNN | (Dai et al., 2024) |
| **Pathformer** ↑ | ✓ | `PRE-DYN` | Linear | Transformer | (Chen et al., 2024) |
| **iTransformer** ↑ | ✓ | `PRE-DYN` | Linear | Transformer | (Liu et al., 2024b) |
| **PatchTST** ↑ | ✓ | `PRE-DYN` | Linear | Transformer | (Nie et al., 2023) |
| **Crossformer** ↑ | ✓ | `PRE-DYN` | Linear | Transformer | (Zhang & Yan, 2023) |
| **TimeMixer** ↑ | ✓ | `PRE-DYN` | Linear | MLP | (Wang et al., 2024) |
| NLinear | ✓ | `DYN` | Linear | None | (Zeng et al., 2023) |
| *FITS* ↓ | ✗ | `PRE-DYN` | Linear & 0-pad. | Filtering | (Xu et al., 2024) |
| *TimesNet* ↓ | ✓ | `PRE-DYN-POST` | Linear | CNN | (Wu et al., 2023) |
| **Triformer** ↓ | ✓ | `PRE-DYN` | Linear | Transformer | (Cirstea et al., 2022) |
| *FEDformer* ↓ | ✗ | `PRE-DYN-POST` | Mean & 0-pad. | Transformer | (Zhou et al., 2022a) |
| *MICN* ↓ | ✗ | `PRE-DYN-POST` | Linear & 0-pad. | CNN | (Wang et al., 2023a) |
| *NS Transformer* ↓ | ✗ | `PRE-DYN-POST` | 0-padding | Transformer | (Liu et al., 2022b) |
| *FiLM* ↓ | ✗ | `PRE-DYN-POST` | Legendre discr. | SSM | (Zhou et al., 2022b) |
| *Informer* ↓ | ✗ | `PRE-DYN-POST` | 0-padding | Transformer | (Zhou et al., 2021) |

In Table 5, we provide additional models analyzed through our `PRO-DYN` nomenclature. In our paper, we mainly focus on **pattern-based TSF models**, which are models designed for the TSF task, **trained from scratch from one dataset to another**, without any prior knowledge on the time series modality (family 1). Their goal is to extract sequential patterns.

We also include two additional model families: foundation TSF models (family 2) and dynamics-based models (family 3).

The **foundation family** pushes the limits of dynamics learning. Indeed, it includes pattern-based models that are trained on a large collection of datasets, exhibiting different physical dynamics. These models aim to adapt their predictions to the specificity of the input data. A group of work (Ansari et al., 2024; Rasul et al., 2023; Cohen et al., 2025; Woo et al., 2024), for instance, approach parameters of a probability distribution and sample a prediction. It appears to be an interesting approach to adapt the `DYN` function behavior regarding the studied data. In addition, we can observe that all the considered models are in a `PRE-DYN` configuration, supporting RQ2 findings: such a configuration for pattern-based models is preferable. It should be compared against the dynamics-based family where well-performing models are in a `PRE-DYN-POST` configuration.

The **dynamics-based family** includes models designed while supposing the time series data is governed by an evolution law. Thanks to our nomenclature, it shows the theory from which the model originates (Koopman, Takens, state-space), impacts the functions design (`PRE-DYN-POST` configuration with autoregressive mechanism to move forward in time when the Koopman theory is invoked for instance). In addition, none of the models considered in this family are Transformer-based,

*Table 5.* Pattern-based TSF models trained on one dataset, foundation and dynamics-based TSF models (resp. first, second and third group block) through the `PRO-DYN` nomenclature. Models within a family are displayed in an arbitrary order. AR stands for autoregressive; AR sampling stands for sampling a distribution autoregressively, with parameters computed from previous observations. Parallel sampling means all predictions are sampled in parallel. Mixture stands for a mixture of models used for prediction.

| Model | Complete learnable dynamics (RQ1) | Config. (RQ2) | DYN function | PRO backbone | Reference |
|---|---|---|---|---|---|
| TiDE | ✓ | PRE-DYN-POST | Linear | MLP | (Das et al., 2023) |
| Pyraformer | ✓ | PRE-DYN | Linear | Transformer | (Liu et al., 2022a) |
| Autoformer | ✓ | PRE-DYN-POST | Linear | Transformer | (Wu et al., 2021) |
| PSformer | ✓ | PRE-DYN | Linear | Transformer | (Wang et al., 2025b) |
| EDformer | ✓ | PRE-DYN | Linear | Transformer | (Chakraborty et al., 2024) |
| Sensoformer | ✓ | PRE-DYN | Linear | Transformer | (Qin et al., 2025) |
| FourierGNN | ✓ | PRE-DYN | Linear | GNN | (Yi et al., 2023) |
| TimeFilter | ✓ | PRE-DYN | Linear | GNN | (Hu et al., 2025) |
| TCN | ✓ | PRE-DYN | AR | CNN | (Bai et al., 2018) |
| RNN | ✓ | PRE-DYN-POST | AR | MLP | (Elman, 1990) |
| TimeGrad | ✗ | PRE-DYN-POST | Sampling | RNN+Diffusion | (Rasul et al., 2021) |
| Chronos | ✓ | PRE-DYN | AR sampling | Transformer | (Ansari et al., 2024) |
| Moment | ✓ | PRE-DYN | Linear | Transformer | (Goswami et al., 2024) |
| Lag-llama | ✓ | PRE-DYN | AR sampling | Transformer | (Rasul et al., 2023) |
| Toto | ✓ | PRE-DYN | AR sampling | Transformer | (Cohen et al., 2025) |
| Moirai | ✓ | PRE-DYN | Parallel sampling | Transformer | (Woo et al., 2024) |
| TTMs | ✓ | PRE-DYN | Linear | MLP | (Ekambaram et al., 2024) |
| ROSE | ✓ | PRE-DYN | Linear | Transformer | (Wang et al., 2025a) |
| Synapse | ✓ | PRE-DYN | Mixture | Transformer | (Das et al., 2026) |
| TimeCopilot | ✓ | PRE-DYN | Mixture | Transformer | (Garza & Garcia, 2025) |
| Kairos | ✓ | PRE-DYN | Linear | Transformer | (Feng et al., 2025) |
| Attraos | ✓ | PRE-DYN | Linear | SSM | (Hu et al., 2024a) |
| DeepEDM | ✓ | DYN-POST | Linear | MLP | (Majeedi et al., 2025) |
| (Lusch et al., 2018) | ✓ | PRE-DYN-POST | AR | MLP | (Lusch et al., 2018) |
| Koopa | ✓ | PRE-DYN-POST | Linear+AR | MLP | (Liu et al., 2023) |
| KNF | ✓ | PRE-DYN-POST | AR | MLP | (Wang et al., 2023b) |
| MZ-AE | ✓ | PRE-DYN-POST | AR | MLP+LSTM | (Gupta et al., 2024) |
| $K^2$VAE | ✓ | PRE-DYN-POST | AR | MLP+Kalman filter | (Wu et al., 2025) |
| SpaceTime | ✓ | PRE-DYN-POST | AR | SSM | (Zhang et al., 2023) |
| DYffusion | ✓ | DYN-POST | Diffusion | CNN | (Cachay et al., 2023) |
| DyDiff | ✗ | DYN-POST | Diffusion | Diffusion | (Guo et al., 2025) |

whereas a majority of foundation models in time series are. Building TSF models on physical dynamics considerations leads to different backbone choices. **Would it be possible to define a foundation-dynamics-based model for TSF ?** As the dynamics modeling of a specific phenomenon is explicitly infused in these models, such a foundation model should be able to adapt its dynamics to one situation to another, raising questions of domain adaptation, for instance. Furthermore, our RQs focus on the nature and placement of the `DYN` layer in forecasting architectures whose `PRO`cessing units are not explicitly dynamics-informed. In this setting, RQ2 experimentally shows that a `PRE-DYN` configuration achieves better performance. Family 3 models operate in a different regime, where `PRE` and/or `POST` units are themselves motivated by dynamics considerations (e.g., Attraos or DeepEDM based on Takens representation theorem), and therefore do not fall under the scope of RQ2. Indeed, dynamics-inspired `PRO`cessing blocks allow to get better results in generative tasks (Hu et al., 2024b). As a result, well-performing `PRE-DYN-POST` models from family 3 do not contradict our findings, but rather as going beyond the specific setting studied in our paper.

More broadly, Table 5 also suggests how richer forms of physical dynamics can be described by the `PRO-DYN` nomenclature. For instance, autoregressive / memory-based `DYN` functions may relate to partially observed physical dynamics, while sampling or diffusion-style `DYN` functions may relate to stochastic physical dynamics.

## E. Details on the Datasets

We provide, in Table 6, the detailed statistics of the datasets used in our experiments. It includes the domain, the sampling frequency, the variate dimension, and the data split for training, validation, and testing.

*Table 6.* Statistics of multivariate datasets of the TFB benchmark (Qiu et al., 2024) taken from (Qiu et al., 2025).

| Dataset | Domain | Frequency | Lengths | Dim | Split | Description |
|---|---|---|---|---|---|---|
| METR-LA | Traffic | 5 mins | 34,272 | 207 | 7:1:2 | Traffic speed dataset collected from loop detectors in the LA County road network |
| PEMS-BAY | Traffic | 5 mins | 52,116 | 325 | 7:1:2 | Traffic speed dataset collected from the CalTrans PeMS |
| PEMS04 | Traffic | 5 mins | 16,992 | 307 | 6:2:2 | Traffic flow time series collected from the CalTrans PeMS |
| PEMS08 | Traffic | 5 mins | 17,856 | 170 | 6:2:2 | Traffic flow time series collected from the CalTrans PeMS |
| Traffic | Traffic | 1 hour | 17,544 | 862 | 7:1:2 | Road occupancy rates measured by 862 sensors on San Francisco Bay area freeways |
| ETTh1 | Electricity | 1 hour | 14,400 | 7 | 6:2:2 | Power transformer 1, comprising seven indicators such as oil temperature and useful load |
| ETTh2 | Electricity | 1 hour | 14,400 | 7 | 6:2:2 | Power transformer 2, comprising seven indicators such as oil temperature and useful load |
| ETTm1 | Electricity | 15 mins | 57,600 | 7 | 6:2:2 | Power transformer 1, comprising seven indicators such as oil temperature and useful load |
| ETTm2 | Electricity | 15 mins | 57,600 | 7 | 6:2:2 | Power transformer 2, comprising seven indicators such as oil temperature and useful load |
| Electricity | Electricity | 1 hour | 26,304 | 321 | 7:1:2 | Electricity records the electricity consumption in kWh every 1 hour from 2012 to 2014 |
| Solar | Energy | 10 mins | 52,560 | 137 | 6:2:2 | Solar production records collected from 137 PV plants in Alabama |
| Wind | Energy | 15 mins | 48,673 | 7 | 7:1:2 | Wind power records from 2020-2021 at 15-minute intervals |
| Weather | Environment | 10 mins | 52,696 | 21 | 7:1:2 | Recorded every for the whole year 2020, which contains 21 meteorological indicators |
| AQShunyi | Environment | 1 hour | 35,064 | 11 | 6:2:2 | Air quality datasets from a measurement station, over a period of 4 years |
| AQWan | Environment | 1 hour | 35,064 | 11 | 6:2:2 | Air quality datasets from a measurement station, over a period of 4 years |
| ZafNoo | Nature | 30 mins | 19,225 | 11 | 7:1:2 | From the Sapflux data project includes sap flow measurements and environmental variables |
| CzeLan | Nature | 30 mins | 19,934 | 11 | 7:1:2 | From the Sapflux data project includes sap flow measurements and environmental variables |
| FRED-MD | Economic | 1 month | 728 | 107 | 7:1:2 | Time series showing a set of macroeconomic indicators from the Federal Reserve Bank |
| Exchange | Economic | 1 day | 7,588 | 8 | 7:1:2 | ExchangeRate collects the daily exchange rates of eight countries |
| NASDAQ | Stock | 1 day | 1,244 | 5 | 7:1:2 | Records opening price, closing price, trading volume, lowest price, and highest price |
| NYSE | Stock | 1 day | 1,243 | 5 | 7:1:2 | Records opening price, closing price, trading volume, lowest price, and highest price |
| NN5 | Banking | 1 day | 791 | 111 | 7:1:2 | NN5 is from banking, records the daily cash withdrawals from ATMs in UK |
| ILI | Health | 1 week | 966 | 7 | 7:1:2 | Recorded indicators of patients data from Centers for Disease Control and Prevention |
| Covid-19 | Health | 1 day | 1,392 | 948 | 7:1:2 | Provide opportunities for researchers to investigate the dynamics of COVID-19 |
| Wike2000 | Web | 1 day | 792 | 2,000 | 7:1:2 | Wike2000 is daily page views of 2000 Wikipedia pages |

# F. Code, Design, and Implementation Details

**Code** The code is available online.[10] The code is based on the repository[11] developed for the TFB benchmark (Qiu et al., 2024) under the MIT license. In the `.\ts_benchmark\baselines\time_series_library` folder, we develop our models and insert them in the `models` folder to use the same methods as the studied vanilla models defined in the `adapters_for_transformers.py` file. Results from the vanilla models were taken from the OpenTS multivariate time series leaderboard,[12] a dashboard developed for the TFB benchmark, in accordance with the results as they were in May 2025 (a web archive from May 2025 of the leaderboard is provided in the code repository). Everything was done in accordance with the MIT license, which grants the right to use the code and datasets of the TFB benchmark without restriction.

**DYN added model design** To validate the modified models, we first tested the performance of the `DYN` and post-processing models on the ILI, NASDAQ, and NYSE datasets, the three lightest ones. When each model got comparable or better results to the vanilla version, while being diverse in the way the linear `DYN` layer was added, we extended the experiments to all the other datasets.

**Resources and implementation** All the introduced and rerun models were trained on CentOs 7.9.2009, on either:

- Intel Xeon Gold 6230 20C @ 2.1GHz with 768 GB memory with up to four NVIDIA Tesla V100 with 32 GB GRAM,

- Intel Xeon Gold 6346 16C @ 3.1GHz with 1024 GB memory with up to four Nvidia HGX A100 with 40 GB memory.

Training was done in PyTorch [13], using the learning procedure developed in the `adapters_for_transformers.py` file: MSE loss criterion, Adam optimizer, learning rate divided by two from one epoch to another. Hyperparameters can be found in the `.\scripts` folder presented in `.sh` files. Datasets are split following the rolling forecast method to prevent information leakage from the future to the past. We needed at most 4 GPUs with 30 CPUs per task, running for 26 hours, on the PEMS-BAY dataset (the largest of the benchmark) for FiLM-based models, for $H = 192$ and $H = 336$. Experiments on the seven largest datasets (PEMS-BAY, Traffic, Electricity, Solar, METR-LA, PEMS04, PEMS08) could run up to 15 hours

---

[10]https://github.com/ARBrachet/PRO-DYN/

[11]https://github.com/decisionintelligence/TFB

[12]https://decisionintelligence.github.io/OpenTS/

[13]Torch version was 1.12.1 with cuda toolkit 11.2.0 and Python 3.9.7

on 4 GPUs with 15 CPUs per task. Otherwise, each experiment runs in several minutes on at most 2 GPUs with 15 CPUs per task.

**PatchTST-PEMS04 resource issue**    We have not been able to run PatchTST post-processing `DYN` on the PEMS04 dataset with $H = 720$ due to memory limit issues with our experimental setup. In order to fill this gap, we set results, for the post-processing version, better by $0.001$ on both MSE and MAE than the vanilla one for the comparison in Sections 4.2 and 4.3. Doing so, we get a worst-case scenario for our analysis (see Table 12). No matter the results obtained on this particular case, the tendencies and thus the analysis remain the same for PatchTST.

## G. Detailed Results

### G.1. Raw Results

In this section, MSE and MAE results are shown. Vanilla model results are directly copied from the TFB benchmark (Qiu et al., 2024). Post-processing models from RQ2 are identified as *DYN-Post* model to refer to post-processing arrangement with a linear `DYN` layer at the beginning. When the `DYN` version is better than the vanilla one, the `DYN` score is in **bold**. For RQ1, when the `PRO` version is better than the `DYN` one, the `PRO` score is underlined. When a vanilla score is starred*, it means the shown result is a better one, obtained with our updated learning hyperparameters, than in the TFB benchmark. For each table, NLinear performance (copied from the TFB benchmark) is added as a reference. Context length $L$ and horizon length $H$ are also shown.

*Table 7.* MSE and MAE scores of Informer-based models. The lower, the better. DYN score is in **bold** when it is better than the vanilla version. PRO score is underlined when it is better than the DYN one. Vanilla score is starred* when we obtained better results than the TFB benchmark with updated learning hyperparameters.

| Dataset | Cont. L | Hor. H | Informer DYN MSE | Informer DYN MAE | Informer PRO MSE | Informer PRO MAE | Informer MSE | Informer MAE | NLinear MSE | NLinear MAE |
|---|---|---|---|---|---|---|---|---|---|---|
| ETTh1 | 96 | 96 | **0.503** | **0.480** | 0.503 | 0.480 | 0.715 | 0.571 | 0.385 | 0.403 |
| | 96 | 192 | **0.558** | **0.507** | 0.623 | 0.535 | 0.726 | 0.574 | 0.422 | 0.426 |
| | 96 | 336 | **0.597** | **0.528** | 0.727 | 0.583 | 0.741 | 0.588 | 0.431 | 0.429 |
| | 96 | 720 | **0.661** | **0.570** | 0.732 | 0.605 | 0.772 | 0.623 | 0.439 | 0.452 |
| ETTh2 | 96 | 96 | **0.345** | **0.384** | 0.345 | 0.384 | 0.362 | 0.394 | 0.276 | 0.338 |
| | 96 | 192 | **0.435** | **0.436** | 0.435 | 0.436 | 0.460 | 0.448 | 0.345 | 0.382 |
| | 336 | 336 | **0.413** | **0.447** | 0.413 | 0.447 | 0.454 | 0.464 | 0.368 | 0.408 |
| | 512 | 720 | 0.410 | **0.453** | 0.407 | 0.449 | 0.410 | 0.454 | 0.406 | 0.441 |
| ETTm1 | 96 | 96 | **0.386** | **0.408** | 0.386 | 0.408 | 0.419 | 0.422 | 0.301 | 0.343 |
| | 96 | 192 | **0.484** | **0.449** | 0.501 | 0.457 | 0.547 | 0.480 | 0.355 | 0.379 |
| | 96 | 336 | **0.521** | **0.472** | 0.528 | 0.480 | 0.654 | 0.531 | 0.372 | 0.385 |
| | 96 | 720 | **0.553** | **0.497** | 0.639 | 0.536 | 0.715 | 0.578 | 0.430 | 0.418 |
| ETTm2 | 96 | 96 | **0.204** | **0.287** | 0.204 | 0.287 | 0.216 | 0.302 | 0.163 | 0.252 |
| | 96 | 192 | **0.271** | **0.326** | 0.273 | 0.327 | 0.320 | 0.365 | 0.218 | 0.290 |
| | 96 | 336 | **0.345** | **0.381** | 0.345 | 0.381 | 0.400 | 0.414 | 0.273 | 0.326 |
| | 96 | 720 | **0.439** | **0.434** | 0.441 | 0.434 | 0.512 | 0.468 | 0.361 | 0.382 |
| Exchange | 96 | 96 | 0.126 | 0.257 | 0.126 | 0.257 | 0.125 | 0.257 | 0.085 | 0.204 |
| | 96 | 192 | 0.222 | 0.341 | 0.206 | 0.329 | 0.208 | 0.335 | 0.175 | 0.297 |
| | 96 | 336 | 0.386 | 0.453 | 0.382 | 0.453 | 0.336 | 0.426 | 0.320 | 0.409 |
| | 96 | 720 | 0.974 | 0.752 | 1.018 | 0.768 | 0.663 | 0.631 | 0.838 | 0.690 |
| Weather | 96 | 96 | **0.189** | **0.235** | 0.189 | 0.235 | 0.210 | 0.256 | 0.180 | 0.226 |
| | 96 | 192 | **0.246** | **0.284** | 0.245 | 0.282 | 0.261 | 0.300 | 0.218 | 0.261 |
| | 96 | 336 | **0.296** | **0.316** | 0.301 | 0.321 | 0.309 | 0.332 | 0.266 | 0.296 |
| | 96 | 720 | **0.368** | **0.361** | 0.375 | 0.370 | 0.390 | 0.388 | 0.334 | 0.345 |
| Electricity | 96 | 96 | **0.178** | **0.283** | 0.178 | 0.283 | 0.215 | 0.321 | 0.140 | 0.236 |
| | 96 | 192 | **0.208** | **0.311** | 0.226 | 0.328 | 0.263 | 0.362 | 0.155 | 0.248 |
| | 96 | 336 | **0.232** | **0.331** | 0.267 | 0.363 | 0.334 | 0.416 | 0.171 | 0.264 |
| | 96 | 720 | **0.243** | **0.337** | 0.343 | 0.417 | 0.502 | 0.525 | 0.210 | 0.297 |
| ILI | 104 | 24 | **2.215** | **1.015** | 2.213 | 0.994 | 2.832 | 1.174 | 1.998 | 0.919 |
| | 104 | 36 | **2.429** | **1.024** | 2.481 | 1.036 | 2.889 | 1.147 | 1.920 | 0.916 |
| | 104 | 48 | **2.278** | **0.995** | 2.177 | 0.979 | 2.944 | 1.177 | 1.895 | 0.924 |
| | 36 | 60 | **2.340** | **0.981** | 2.340 | 0.979 | 2.902 | 1.158 | 1.964 | 0.947 |
| Solar | 96 | 96 | 0.356 | 0.387 | 0.356 | 0.387 | 0.329 | 0.368 | 0.202 | 0.245 |
| | 96 | 192 | 0.378 | 0.395 | 0.395 | 0.408 | 0.370 | 0.388 | 0.223 | 0.258 |
| | 96 | 336 | **0.401** | **0.410** | 0.405 | 0.412 | 0.419 | 0.420 | 0.238 | 0.265 |
| | 96 | 720 | 0.408 | **0.389** | 0.441 | 0.431 | 0.386 | 0.405 | 0.246 | 0.268 |
| Traffic | 96 | 96 | **0.609** | **0.327** | 0.609 | 0.327 | 0.682* | 0.388* | 0.395 | 0.272 |
| | 96 | 192 | 2.953 | 1.299 | 2.716 | 1.257 | 1.561* | 0.852* | 0.407 | 0.277 |
| | 96 | 336 | **0.648** | **0.346** | 0.688 | 0.384 | 0.862 | 0.477 | 0.417 | 0.282 |
| | 96 | 720 | **3.005** | **1.224** | 3.086 | 1.268 | 3.203 | 1.294 | 0.453 | 0.302 |
| PEMS-BAY | 96 | 96 | **0.807** | **0.435** | 0.807 | 0.435 | 0.818 | 0.439 | 0.642 | 0.402 |
| | 96 | 192 | **0.893** | **0.459** | 0.884 | 0.460 | 0.900 | 0.470 | 0.687 | 0.420 |
| | 96 | 336 | **0.831** | **0.443** | 0.826 | 0.447 | 0.844 | 0.454 | 0.735 | 0.437 |
| | 96 | 720 | **0.960** | **0.481** | 0.981 | 0.494 | 1.013 | 0.508 | 0.924 | 0.514 |
| METR-LA | 96 | 96 | **1.359** | **0.656** | 1.359 | 0.656 | 1.366 | 0.679 | 1.042 | 0.651 |
| | 96 | 192 | **1.543** | **0.701** | 1.525 | 0.700 | 1.558 | 0.712 | 1.218 | 0.720 |
| | 96 | 336 | 1.609 | **0.710** | 1.600 | 0.716 | 1.595 | 0.719 | 1.334 | 0.756 |
| | 96 | 720 | 1.851 | **0.792** | 1.846 | 0.800 | 1.833 | 0.806 | 1.683 | 0.886 |
| PEMS04 | 96 | 96 | 0.255 | 0.365 | 0.255 | 0.365 | 0.249 | 0.361 | 0.208 | 0.301 |
| | 96 | 192 | **0.287** | **0.388** | 0.296 | 0.396 | 0.288 | 0.390 | 0.229 | 0.312 |
| | 96 | 336 | **0.266** | **0.366** | 0.279 | 0.378 | 0.275 | 0.374 | 0.251 | 0.331 |
| | 96 | 720 | **0.348** | **0.427** | 0.396 | 0.454 | 0.376 | 0.451 | 0.346 | 0.398 |
| PEMS08 | 96 | 96 | 0.457 | 0.467 | 0.457 | 0.467 | 0.385 | 0.419 | 0.340 | 0.327 |
| | 96 | 192 | **0.571** | **0.487** | 0.591 | 0.503 | 0.589 | 0.501 | 0.450 | 0.353 |
| | 96 | 336 | 0.570 | **0.449** | 0.602 | 0.473 | 0.568 | 0.457 | 0.454 | 0.369 |
| | 96 | 720 | **0.699** | **0.511** | 0.775 | 0.559 | 0.739 | 0.540 | 0.494 | 0.429 |
| AQShunyi | 512 | 96 | **0.693** | **0.505** | 0.688 | 0.503 | 0.754 | 0.542 | 0.653 | 0.486 |
| | 512 | 192 | **0.719** | **0.517** | 0.717 | 0.516 | 0.759 | 0.536 | 0.701 | 0.506 |
| | 96 | 336 | 0.838 | 0.560 | 0.845 | 0.562 | 0.837 | 0.560 | 0.722 | 0.519 |
| | 512 | 720 | 0.805 | 0.554 | 0.800 | 0.554 | 0.777 | 0.543 | 0.777 | 0.545 |
| AQWan | 96 | 96 | **0.865** | **0.501** | 0.865 | 0.501 | 0.902 | 0.522 | 0.758 | 0.475 |
| | 512 | 192 | **0.805** | **0.499** | 0.815 | 0.503 | 0.833 | 0.521 | 0.809 | 0.496 |
| | 512 | 336 | **0.834** | **0.511** | 0.824 | 0.506 | 0.847 | 0.525 | 0.830 | 0.508 |
| | 512 | 720 | 0.909 | 0.543 | 0.909 | 0.543 | 0.883 | 0.532 | 0.906 | 0.538 |
| Wind | 96 | 96 | 0.975 | 0.665 | 0.975 | 0.665 | 0.954 | 0.663 | 0.923 | 0.640 |
| | 96 | 192 | 1.161 | **0.756** | 1.171 | 0.762 | 1.145 | 0.764 | 1.081 | 0.734 |
| | 96 | 336 | 1.321 | 0.833 | 1.343 | 0.844 | 1.313 | 0.826 | 1.228 | 0.805 |
| | 512 | 720 | 1.332 | 0.865 | 1.335 | 0.870 | 1.318 | 0.862 | 1.328 | 0.852 |
| ZafNoo | 96 | 96 | **0.531** | **0.452** | 0.531 | 0.452 | 0.547 | 0.477 | 0.447 | 0.410 |
| | 96 | 192 | **0.610** | **0.493** | 0.614 | 0.506 | 0.698 | 0.570 | 0.503 | 0.447 |
| | 96 | 336 | **0.623** | **0.505** | 0.677 | 0.544 | 0.832 | 0.655 | 0.545 | 0.470 |
| | 96 | 720 | **0.690** | **0.536** | 0.890 | 0.687 | 0.876 | 0.699 | 0.589 | 0.497 |
| CzeLan | 96 | 96 | 0.257 | 0.305 | 0.257 | 0.305 | 0.238 | 0.296 | 0.178 | 0.228 |
| | 96 | 192 | **0.284** | **0.327** | 0.280 | 0.325 | 0.299 | 0.340 | 0.210 | 0.252 |
| | 96 | 336 | **0.307** | **0.341** | 0.321 | 0.349 | 0.351 | 0.366 | 0.243 | 0.280 |
| | 336 | 720 | **0.357** | **0.378** | 0.382 | 0.404 | 0.384 | 0.416 | 0.290 | 0.326 |
| Covid-19 | 36 | 24 | **2.115** | **0.067** | 2.122 | 0.067 | 2.687 | 0.080 | 1.139 | 0.070 |
| | 36 | 36 | **2.465** | **0.075** | 2.465 | 0.075 | 3.071 | 0.087 | 1.582 | 0.091 |
| | 36 | 48 | **2.766** | **0.083** | 2.792 | 0.082 | 3.548 | 0.094 | 1.932 | 0.099 |
| | 36 | 60 | **3.258** | **0.091** | 3.361 | 0.091 | 4.052 | 0.104 | 2.682 | 0.127 |
| NASDAQ | 36 | 24 | **0.835** | **0.681** | 0.880 | 0.700 | 1.076 | 0.727 | 0.557 | 0.522 |
| | 104 | 36 | **1.234** | **0.865** | 1.241 | 0.865 | 1.411 | 0.897 | 0.869 | 0.668 |
| | 104 | 48 | **1.152** | **0.835** | 1.142 | 0.832 | 1.289 | 0.863 | 1.152 | 0.770 |
| | 104 | 60 | **1.072** | **0.810** | 1.072 | 0.810 | 1.158 | 0.829 | 1.284 | 0.809 |
| NYSE | 36 | 24 | 0.284 | 0.363 | 0.284 | 0.363 | 0.276 | 0.349 | 0.193 | 0.283 |
| | 36 | 36 | 0.408 | 0.435 | 0.408 | 0.435 | 0.407 | 0.417 | 0.315 | 0.356 |
| | 36 | 48 | **0.546** | 0.496 | 0.543 | 0.493 | 0.567 | 0.496 | 0.464 | 0.438 |
| | 36 | 60 | **0.796** | **0.602** | 0.706 | 0.570 | 0.899 | 0.670 | 0.631 | 0.522 |
| FRED-MD | 36 | 24 | **61.700** | **1.471** | 60.306 | 1.461 | 70.055 | 1.562 | 32.125 | 0.931 |
| | 36 | 36 | **82.488** | **1.706** | 82.488 | 1.706 | 101.858 | 1.860 | 58.332 | 1.260 |
| | 36 | 48 | **116.781** | **1.998** | 114.924 | 1.984 | 137.938 | 2.147 | 82.184 | 1.609 |
| | 36 | 60 | **155.983** | **2.297** | 153.512 | 2.277 | 184.740 | 2.460 | 109.625 | 1.882 |
| NN5 | 104 | 24 | **1.289** | **0.916** | 1.290 | 0.916 | 1.362 | 0.949 | 0.758 | 0.592 |
| | 104 | 36 | **1.270** | **0.913** | 1.265 | 0.910 | 1.341 | 0.944 | 0.693 | 0.577 |
| | 104 | 48 | **1.275** | **0.920** | 1.275 | 0.920 | 1.337 | 0.942 | 0.688 | 0.587 |
| | 104 | 60 | **1.281** | **0.925** | 1.284 | 0.926 | 1.335 | 0.945 | 0.679 | 0.587 |
| Wike2000 | 36 | 24 | 1196.199 | 1.578 | 1133.634 | 1.580 | 1190.077* | 1.566* | 1135.609 | 1.350 |
| | 36 | 36 | 1315.755 | **1.648** | 1315.755 | 1.648 | 1274.502 | 1.657 | 1060.939 | 1.388 |
| | 36 | 48 | **1328.209** | **1.700** | 1333.007 | 1.741 | 1372.456 | 1.718 | 1899.937 | 1.733 |
| | 36 | 60 | 1371.842 | **1.769** | 1485.935 | 1.821 | 1368.723 | 1.808 | 1281.245 | 1.570 |

*Table 8.* MSE and MAE scores of FiLM-based models. The lower, the better. DYN score is in **bold** when it is better than the vanilla version. PRO score is underlined when it is better than the DYN one. Vanilla score is starred* when we obtained better results than the TFB benchmark with updated learning hyperparameters.

| Dataset | Cont. L | Hor. H | FiLM DYN MSE | FiLM DYN MAE | FiLM PRO MSE | FiLM PRO MAE | FiLM MSE | FiLM MAE | NLinear MSE | NLinear MAE |
|---|---|---|---|---|---|---|---|---|---|---|
| ETTh1 | 512 | 96 | 0.367 | 0.392 | 0.366 | 0.392 | 0.370 | 0.394 | 0.385 | 0.403 |
| | 512 | 192 | 0.403 | 0.416 | 0.402 | 0.415 | 0.405 | 0.416 | 0.422 | 0.426 |
| | 512 | 336 | 0.429 | 0.433 | 0.428 | 0.433 | 0.434 | 0.435 | 0.431 | 0.429 |
| | 336 | 720 | 0.459 | 0.470 | 0.464 | 0.472 | 0.463 | 0.474 | 0.439 | 0.452 |
| ETTh2 | 512 | 96 | 0.275 | 0.339 | 0.269 | 0.335 | 0.275* | 0.340* | 0.276 | 0.338 |
| | 336 | 192 | 0.337 | 0.383 | 0.338 | 0.384 | 0.340* | 0.385* | 0.345 | 0.382 |
| | 512 | 336 | 0.367 | 0.415 | 0.373 | 0.417 | 0.372 | 0.425 | 0.368 | 0.408 |
| | 336 | 720 | 0.424 | 0.449 | 0.429 | 0.455 | 0.425 | 0.455 | 0.406 | 0.441 |
| ETTm1 | 512 | 96 | 0.306 | 0.348 | 0.305 | 0.347 | 0.301 | 0.343 | 0.301 | 0.343 |
| | 336 | 192 | 0.337 | 0.365 | 0.345 | 0.371 | 0.339 | 0.365 | 0.355 | 0.379 |
| | 336 | 336 | 0.371 | 0.383 | 0.371 | 0.383 | 0.374 | 0.385 | 0.372 | 0.385 |
| | 512 | 720 | 0.422 | 0.414 | 0.433 | 0.422 | 0.423 | 0.414 | 0.430 | 0.418 |
| ETTm2 | 512 | 96 | 0.163 | 0.252 | 0.163 | 0.252 | 0.165 | 0.254 | 0.163 | 0.252 |
| | 512 | 192 | 0.220 | 0.290 | 0.218 | 0.290 | 0.220 | 0.291 | 0.218 | 0.290 |
| | 336 | 336 | 0.274 | 0.326 | 0.274 | 0.326 | 0.277 | 0.329 | 0.273 | 0.326 |
| | 512 | 720 | 0.361 | 0.385 | 0.362 | 0.385 | 0.363 | 0.386 | 0.361 | 0.382 |
| Exchange | 512 | 96 | 0.088 | 0.207 | 0.087 | 0.206 | 0.087 | 0.210 | 0.085 | 0.204 |
| | 96 | 192 | 0.173 | 0.294 | 0.172 | 0.294 | 0.182 | 0.308 | 0.175 | 0.297 |
| | 96 | 336 | 0.318 | 0.407 | 0.315 | 0.405 | 0.318 | 0.409 | 0.320 | 0.409 |
| | 96 | 720 | 0.821 | 0.682 | 0.800 | 0.672 | 0.815 | 0.681 | 0.838 | 0.690 |
| Weather | 336 | 96 | 0.174 | 0.223 | 0.177 | 0.226 | 0.178 | 0.229 | 0.180 | 0.226 |
| | 512 | 192 | 0.213 | 0.258 | 0.214 | 0.259 | 0.218 | 0.263 | 0.218 | 0.261 |
| | 336 | 336 | 0.264 | 0.293 | 0.264 | 0.293 | 0.266 | 0.295 | 0.266 | 0.296 |
| | 336 | 720 | 0.331 | 0.339 | 0.331 | 0.339 | 0.332 | 0.341 | 0.334 | 0.345 |
| Electricity | 192 | 96 | 0.153 | 0.246 | 0.153 | 0.246 | 0.154 | 0.246 | 0.140 | 0.236 |
| | 192 | 192 | 0.172 | 0.268 | 0.172 | 0.268 | 0.168 | 0.261 | 0.155 | 0.248 |
| | 192 | 336 | 0.188 | 0.283 | 0.188 | 0.283 | 0.189 | 0.284 | 0.171 | 0.264 |
| | 192 | 720 | 0.248 | 0.339 | 0.249 | 0.340 | 0.249 | 0.340 | 0.210 | 0.297 |
| ILI | 104 | 24 | 2.337 | 1.026 | 2.143 | 0.966 | 2.256 | 0.996 | 1.998 | 0.919 |
| | 104 | 36 | 2.290 | 1.038 | 2.067 | 0.972 | 2.133 | 0.992 | 1.920 | 0.916 |
| | 104 | 48 | 2.511 | 1.070 | 2.193 | 1.011 | 2.034 | 0.969 | 1.895 | 0.924 |
| | 104 | 60 | 1.981 | 0.920 | 2.021 | 0.936 | 1.974 | 0.929 | 1.964 | 0.947 |
| Solar | 512 | 96 | 0.202 | 0.245 | 0.202 | 0.244 | 0.214 | 0.259 | 0.202 | 0.245 |
| | 512 | 192 | 0.224 | 0.256 | 0.224 | 0.256 | 0.226 | 0.257 | 0.223 | 0.258 |
| | 512 | 336 | 0.238 | 0.264 | 0.238 | 0.263 | 0.241 | 0.265 | 0.238 | 0.265 |
| | 512 | 720 | 0.245 | 0.267 | 0.245 | 0.267 | 0.247 | 0.268 | 0.246 | 0.268 |
| Traffic | 512 | 96 | 0.397 | 0.276 | 0.396 | 0.274 | 0.412 | 0.284 | 0.395 | 0.272 |
| | 512 | 192 | 0.413 | 0.286 | 0.410 | 0.282 | 0.415 | 0.285 | 0.407 | 0.277 |
| | 512 | 336 | 0.428 | 0.299 | 0.426 | 0.297 | 0.430 | 0.299 | 0.417 | 0.282 |
| | 512 | 720 | 0.517 | 0.364 | 0.517 | 0.364 | 0.525 | 0.371 | 0.453 | 0.302 |
| PEMS-BAY | 512 | 96 | 0.637 | 0.392 | 0.636 | 0.391 | 0.658 | 0.421 | 0.642 | 0.402 |
| | 512 | 192 | 0.677 | 0.401 | 0.676 | 0.401 | 0.680 | 0.401 | 0.687 | 0.420 |
| | 512 | 336 | 0.725 | 0.417 | 0.725 | 0.417 | 0.728 | 0.418 | 0.735 | 0.437 |
| | 720 | 720 | 0.894 | 0.483 | 0.894 | 0.483 | 0.895 | 0.482 | 0.924 | 0.514 |
| METR-LA | 96 | 96 | 1.276 | 0.681 | 1.276 | 0.681 | 1.287 | 0.697 | 1.042 | 0.651 |
| | 720 | 192 | 1.196 | 0.710 | 1.199 | 0.718 | 1.223 | 0.733 | 1.218 | 0.720 |
| | 512 | 336 | 1.327 | 0.742 | 1.322 | 0.736 | 1.334 | 0.741 | 1.334 | 0.756 |
| | 512 | 720 | 1.590 | 0.818 | 1.592 | 0.819 | 1.602 | 0.822 | 1.683 | 0.886 |
| PEMS04 | 720 | 96 | 0.203 | 0.295 | 0.202 | 0.294 | 0.208 | 0.299 | 0.208 | 0.301 |
| | 720 | 192 | 0.218 | 0.306 | 0.218 | 0.305 | 0.221 | 0.308 | 0.229 | 0.312 |
| | 720 | 336 | 0.239 | 0.322 | 0.238 | 0.322 | 0.246 | 0.327 | 0.251 | 0.331 |
| | 720 | 720 | 0.317 | 0.383 | 0.317 | 0.383 | 0.329 | 0.391 | 0.346 | 0.398 |
| PEMS08 | 512 | 96 | 0.343 | 0.326 | 0.343 | 0.330 | 0.349 | 0.330 | 0.340 | 0.327 |
| | 512 | 192 | 0.418 | 0.352 | 0.416 | 0.351 | 0.428 | 0.352 | 0.450 | 0.353 |
| | 512 | 336 | 0.453 | 0.370 | 0.454 | 0.373 | 0.463 | 0.374 | 0.454 | 0.369 |
| | 512 | 720 | 0.493 | 0.429 | 0.493 | 0.430 | 0.498 | 0.434 | 0.494 | 0.429 |
| AQShunyi | 336 | 96 | 0.659 | 0.483 | 0.659 | 0.483 | 0.664 | 0.486 | 0.653 | 0.486 |
| | 336 | 192 | 0.700 | 0.501 | 0.700 | 0.501 | 0.705 | 0.504 | 0.701 | 0.506 |
| | 336 | 336 | 0.720 | 0.514 | 0.720 | 0.514 | 0.725 | 0.517 | 0.722 | 0.519 |
| | 336 | 720 | 0.778 | 0.542 | 0.779 | 0.543 | 0.782 | 0.544 | 0.777 | 0.545 |
| AQWan | 336 | 96 | 0.758 | 0.471 | 0.760 | 0.474 | 0.766 | 0.475 | 0.758 | 0.475 |
| | 336 | 192 | 0.804 | 0.490 | 0.804 | 0.490 | 0.809 | 0.494 | 0.809 | 0.496 |
| | 336 | 336 | 0.824 | 0.502 | 0.824 | 0.502 | 0.831 | 0.505 | 0.830 | 0.508 |
| | 336 | 720 | 0.903 | 0.534 | 0.902 | 0.533 | 0.906 | 0.536 | 0.906 | 0.538 |
| Wind | 336 | 96 | 0.926 | 0.646 | 0.927 | 0.646 | 0.933 | 0.649 | 0.923 | 0.640 |
| | 512 | 192 | 1.087 | 0.737 | 1.087 | 0.739 | 1.097 | 0.743 | 1.081 | 0.734 |
| | 512 | 336 | 1.226 | 0.801 | 1.230 | 0.803 | 1.241 | 0.809 | 1.228 | 0.805 |
| | 512 | 720 | 1.324 | 0.847 | 1.325 | 0.847 | 1.334 | 0.852 | 1.328 | 0.852 |
| ZafNoo | 336 | 96 | 0.447 | 0.409 | 0.447 | 0.408 | 0.451 | 0.411 | 0.447 | 0.410 |
| | 512 | 192 | 0.503 | 0.445 | 0.503 | 0.444 | 0.508 | 0.448 | 0.503 | 0.447 |
| | 512 | 336 | 0.544 | 0.469 | 0.545 | 0.469 | 0.549 | 0.471 | 0.545 | 0.470 |
| | 512 | 720 | 0.598 | 0.503 | 0.593 | 0.500 | 0.598 | 0.504 | 0.589 | 0.497 |
| CzeLan | 336 | 96 | 0.178 | 0.230 | 0.178 | 0.230 | 0.180 | 0.232 | 0.178 | 0.228 |
| | 336 | 192 | 0.210 | 0.253 | 0.209 | 0.253 | 0.212 | 0.255 | 0.210 | 0.252 |
| | 336 | 336 | 0.242 | 0.279 | 0.242 | 0.279 | 0.243 | 0.281 | 0.243 | 0.280 |
| | 512 | 720 | 0.281 | 0.311 | 0.281 | 0.310 | 0.282 | 0.312 | 0.290 | 0.326 |
| Covid-19 | 36 | 24 | 1.055 | 0.045 | 1.058 | 0.045 | 1.183 | 0.047 | 1.139 | 0.070 |
| | 36 | 36 | 1.371 | 0.054 | 1.371 | 0.054 | 1.470 | 0.059 | 1.582 | 0.091 |
| | 36 | 48 | 1.739 | 0.063 | 1.754 | 0.064 | 1.859 | 0.069 | 1.932 | 0.099 |
| | 36 | 60 | 2.142 | 0.073 | 2.187 | 0.074 | 2.278 | 0.078 | 2.682 | 0.127 |
| NASDAQ | 36 | 24 | 0.642 | 0.567 | 0.639 | 0.563 | 0.767 | 0.645 | 0.557 | 0.522 |
| | 36 | 36 | 1.000 | 0.707 | 1.000 | 0.707 | 1.271* | 0.800* | 0.869 | 0.668 |
| | 104 | 48 | 1.172 | 0.773 | 1.311 | 0.815 | 1.179 | 0.829 | 1.152 | 0.770 |
| | 104 | 60 | 1.342 | 0.825 | 1.380 | 0.835 | 1.303 | 0.853 | 1.284 | 0.809 |
| NYSE | 36 | 24 | 0.195 | 0.283 | 0.192 | 0.282 | 0.276* | 0.339* | 0.193 | 0.283 |
| | 36 | 36 | 0.314 | 0.358 | 0.314 | 0.358 | 0.376* | 0.406* | 0.315 | 0.356 |
| | 36 | 48 | 0.452 | 0.427 | 0.451 | 0.428 | 0.523* | 0.474* | 0.464 | 0.438 |
| | 36 | 60 | 0.613 | 0.517 | 0.611 | 0.514 | 0.698* | 0.554* | 0.631 | 0.522 |
| FRED-MD | 36 | 24 | 31.809 | 0.960 | 31.050 | 0.948 | 37.426* | 1.092* | 32.125 | 0.931 |
| | 36 | 36 | 60.333 | 1.331 | 60.333 | 1.311 | 90.434 | 1.670 | 58.332 | 1.258 |
| | 104 | 48 | 99.858 | 1.709 | 105.524 | 1.719 | 131.081 | 2.119 | 82.184 | 1.609 |
| | 36 | 60 | 126.751 | 1.941 | 130.846 | 1.974 | 180.367 | 2.397 | 109.625 | 1.882 |
| NN5 | 104 | 24 | 0.821 | 0.637 | 0.785 | 0.613 | 0.846 | 0.651 | 0.758 | 0.592 |
| | 104 | 36 | 0.812 | 0.652 | 0.716 | 0.593 | 0.883 | 0.702 | 0.693 | 0.577 |
| | 36 | 48 | 0.822 | 0.657 | 0.805 | 0.649 | 0.969 | 0.741 | 0.688 | 0.587 |
| | 104 | 60 | 0.657 | 0.572 | 0.650 | 0.567 | 0.633 | 0.556 | 0.679 | 0.587 |
| Wike2000 | 36 | 24 | 631.760 | 1.166 | 623.585 | 1.162 | 959.454 | 1.318 | 1135.609 | 1.350 |
| | 36 | 36 | 667.587 | 1.245 | 667.587 | 1.245 | 983.239 | 1.419 | 1060.939 | 1.388 |
| | 36 | 48 | 724.318 | 1.334 | 708.970 | 1.320 | 1358.260 | 1.581 | 1899.937 | 1.733 |
| | 36 | 60 | 743.718 | 1.389 | 750.659 | 1.387 | 1157.091 | 1.582 | 1281.245 | 1.570 |

*Table 9.* MSE and MAE scores of MICN-based models. The lower, the better. DYN score is in **bold** when it is better than the vanilla version. Vanilla score is starred* when we obtained better results than the TFB benchmark with updated learning hyperparameters.

| Dataset | Cont. L | Hor. H | MICN PRO MSE | MAE | MICN MSE | MAE | NLinear MSE | MAE |
|---|---|---|---|---|---|---|---|---|
| ETTh1 | 96 | 96 | **0.375** | **0.410** | 0.377* | 0.412 | 0.385 | 0.403 |
| | 512 | 192 | **0.398** | **0.428** | 0.400 | 0.430 | 0.422 | 0.426 |
| | 336 | 336 | 0.425 | 0.446 | 0.425* | 0.445* | 0.431 | 0.429 |
| | 512 | 720 | 0.479 | 0.501 | 0.474 | 0.499 | 0.439 | 0.452 |
| ETTh2 | 336 | 96 | 0.312 | **0.370** | 0.312* | 0.371* | 0.276 | 0.338 |
| | 336 | 192 | 0.393 | 0.423 | 0.393* | 0.423* | 0.345 | 0.382 |
| | 336 | 336 | 0.473 | 0.474 | 0.472* | 0.474* | 0.368 | 0.408 |
| | 336 | 720 | **0.703** | **0.591** | 0.723 | 0.600 | 0.406 | 0.441 |
| ETTm1 | 336 | 96 | **0.301** | **0.348** | 0.302* | 0.349 | 0.301 | 0.343 |
| | 336 | 192 | 0.331 | **0.368** | 0.331* | 0.369 | 0.355 | 0.379 |
| | 512 | 336 | 0.368 | **0.388** | 0.368* | 0.389* | 0.372 | 0.385 |
| | 512 | 720 | 0.416 | **0.420** | 0.410 | 0.421 | 0.430 | 0.418 |
| ETTm2 | 512 | 96 | 0.172 | 0.269 | 0.172* | 0.269* | 0.163 | 0.252 |
| | 512 | 192 | 0.223 | 0.299 | 0.223* | 0.299* | 0.218 | 0.290 |
| | 512 | 336 | **0.297** | **0.360** | 0.303 | 0.366* | 0.273 | 0.326 |
| | 512 | 720 | **0.438** | **0.446** | 0.461* | 0.477* | 0.361 | 0.382 |
| Exchange | 96 | 96 | 0.079 | 0.203 | 0.079 | 0.203 | 0.085 | 0.204 |
| | 96 | 192 | 0.159 | 0.299 | 0.158 | 0.299 | 0.175 | 0.297 |
| | 96 | 336 | 0.296 | 0.418 | 0.296 | 0.418 | 0.320 | 0.409 |
| | 96 | 720 | 0.763 | 0.677 | 0.745 | 0.675 | 0.838 | 0.690 |
| Weather | 512 | 96 | 0.172 | 0.231 | 0.172 | 0.231* | 0.180 | 0.226 |
| | 512 | 192 | 0.214 | **0.270** | 0.214 | 0.270* | 0.218 | 0.261 |
| | 512 | 336 | **0.258** | **0.307** | 0.259 | 0.309 | 0.266 | 0.296 |
| | 96 | 720 | 0.313 | 0.358 | 0.309 | 0.343 | 0.334 | 0.345 |
| Electricity | 96 | 96 | 0.160 | 0.270 | 0.158 | 0.266 | 0.140 | 0.236 |
| | 96 | 192 | 0.174 | 0.285 | 0.173* | 0.284* | 0.155 | 0.248 |
| | 96 | 336 | 0.183 | 0.295 | 0.183* | 0.294* | 0.171 | 0.264 |
| | 96 | 720 | 0.203 | 0.314 | 0.200 | 0.310 | 0.210 | 0.297 |
| ILI | 104 | 24 | **2.256** | **1.013** | 2.279 | 1.020 | 1.998 | 0.919 |
| | 104 | 36 | 2.491 | 1.095 | 2.451 | 1.085 | 1.920 | 0.916 |
| | 104 | 48 | 2.469 | 1.085 | 2.440 | 1.077 | 1.895 | 0.924 |
| | 104 | 60 | **2.273** | **1.006** | 2.303 | 1.012 | 1.964 | 0.947 |
| Solar | 96 | 96 | **0.180** | 0.252 | 0.190 | 0.250 | 0.202 | 0.245 |
| | 96 | 192 | **0.207** | **0.246** | 0.226 | 0.284 | 0.223 | 0.258 |
| | 96 | 336 | **0.224** | **0.279** | 0.259 | 0.308 | 0.238 | 0.265 |
| | 96 | 720 | **0.276** | **0.319** | 0.341 | 0.365 | 0.246 | 0.268 |
| Traffic | 96 | 96 | 0.523 | **0.305** | 0.508* | 0.299* | 0.395 | 0.272 |
| | 96 | 192 | 0.537 | 0.316 | 0.518* | 0.298* | 0.407 | 0.277 |
| | 96 | 336 | 0.550 | 0.322 | 0.545 | 0.307 | 0.417 | 0.282 |
| | 96 | 720 | 0.594 | 0.333 | 0.569 | 0.328 | 0.453 | 0.302 |
| PEMS-BAY | 96 | 96 | **0.637** | 0.419 | 0.647 | 0.393 | 0.642 | 0.402 |
| | 96 | 192 | **0.760** | **0.443** | 0.767 | 0.444 | 0.687 | 0.420 |
| | 96 | 336 | **0.835** | 0.523 | 0.903 | 0.468 | 0.735 | 0.437 |
| | 96 | 720 | **1.046** | **0.613** | 1.119 | 0.634 | 0.924 | 0.514 |
| METR-LA | 96 | 96 | **1.161** | **0.724** | 1.195 | 0.729 | 1.042 | 0.651 |
| | 96 | 192 | 1.359 | 0.735 | 1.309 | 0.729 | 1.218 | 0.720 |
| | 96 | 336 | 1.342 | 0.778 | 1.332 | 0.773 | 1.334 | 0.756 |
| | 96 | 720 | 1.502 | 0.805 | 1.501 | 0.793 | 1.683 | 0.886 |
| PEMS04 | 96 | 96 | 0.257 | 0.362 | 0.241 | 0.350 | 0.208 | 0.301 |
| | 96 | 192 | **0.290** | **0.390** | 0.358 | 0.444 | 0.229 | 0.312 |
| | 96 | 336 | **0.310** | **0.413** | 0.383 | 0.460 | 0.251 | 0.331 |
| | 96 | 720 | 0.749 | 0.679 | 0.675 | 0.633 | 0.346 | 0.398 |
| PEMS08 | 96 | 96 | 0.477 | 0.450 | 0.391 | 0.412 | 0.340 | 0.327 |
| | 96 | 192 | **0.462** | **0.457** | 0.517 | 0.507 | 0.450 | 0.353 |
| | 96 | 336 | **0.472** | **0.445** | 0.494 | 0.479 | 0.454 | 0.369 |
| | 96 | 720 | 0.792 | **0.631** | 0.776 | 0.647 | 0.494 | 0.429 |
| AQShunyi | 96 | 96 | 0.693 | **0.511** | 0.688* | 0.512 | 0.653 | 0.486 |
| | 96 | 192 | 0.716 | 0.528 | 0.710* | 0.528* | 0.701 | 0.506 |
| | 96 | 336 | **0.724** | **0.539** | 0.729 | 0.544 | 0.722 | 0.519 |
| | 336 | 720 | **0.724** | **0.530** | 0.732* | 0.531* | 0.777 | 0.545 |
| AQWan | 336 | 96 | 0.773 | **0.492** | 0.773* | 0.493* | 0.758 | 0.475 |
| | 96 | 192 | 0.819 | 0.518 | 0.819 | 0.518 | 0.809 | 0.496 |
| | 96 | 336 | **0.823** | 0.522 | 0.826 | 0.522* | 0.830 | 0.508 |
| | 96 | 720 | **0.872** | **0.542** | 0.875 | 0.544 | 0.906 | 0.538 |
| Wind | 96 | 96 | **0.864** | 0.623 | 0.865* | 0.623* | 0.923 | 0.640 |
| | 96 | 192 | **1.024** | **0.709** | 1.066 | 0.719 | 1.081 | 0.734 |
| | 512 | 336 | **1.152** | **0.778** | 1.160* | 0.781* | 1.228 | 0.805 |
| | 512 | 720 | 1.247 | 0.820 | 1.245 | 0.820 | 1.328 | 0.852 |
| ZafNoo | 96 | 96 | **0.437** | **0.418** | 0.442 | 0.421 | 0.447 | 0.410 |
| | 336 | 192 | 0.507 | 0.464 | 0.493 | 0.455 | 0.503 | 0.447 |
| | 96 | 336 | **0.497** | **0.451** | 0.504 | 0.455 | 0.545 | 0.470 |
| | 96 | 720 | **0.532** | **0.470** | 0.540 | 0.476 | 0.589 | 0.497 |
| CzeLan | 512 | 96 | 0.194 | 0.275 | 0.194 | 0.275 | 0.178 | 0.228 |
| | 336 | 192 | **0.254** | **0.322** | 0.275 | 0.362 | 0.210 | 0.252 |
| | 512 | 336 | 0.310 | **0.356** | 0.309 | 0.359 | 0.243 | 0.280 |
| | 512 | 720 | **0.339** | **0.387** | 0.358 | 0.395 | 0.290 | 0.326 |
| Covid-19 | 36 | 24 | **35.655** | **0.448** | 38.500 | 0.469 | 1.139 | 0.070 |
| | 104 | 36 | 47.277 | 0.519 | 45.425 | 0.489 | 1.582 | 0.091 |
| | 104 | 48 | **30.860** | **0.434** | 46.565 | 0.566 | 1.932 | 0.099 |
| | 104 | 60 | 80.919 | 0.645 | 77.972 | 0.616 | 2.682 | 0.127 |
| NASDAQ | 36 | 24 | **0.743** | **0.618** | 0.769* | 0.624* | 0.557 | 0.522 |
| | 36 | 36 | **1.277** | **0.816** | 1.319* | 0.834* | 0.869 | 0.668 |
| | 36 | 48 | **1.693** | **0.940** | 1.788 | 0.977 | 1.152 | 0.770 |
| | 36 | 60 | 2.262 | **1.117** | 2.280* | 1.124* | 1.284 | 0.809 |
| NYSE | 36 | 24 | 0.411 | 0.461 | 0.408* | 0.458* | 0.193 | 0.283 |
| | 36 | 36 | **0.640** | **0.600** | 0.733 | 0.635 | 0.315 | 0.356 |
| | 104 | 48 | 0.843 | 0.695 | 0.768 | 0.663 | 0.464 | 0.438 |
| | 36 | 60 | 1.218 | 0.854 | 1.204 | 0.843 | 0.631 | 0.522 |
| FRED-MD | 36 | 24 | **63.213** | **1.518** | 63.217 | 1.521 | 32.125 | 0.931 |
| | 36 | 36 | 102.826 | 1.947 | 102.800 | 1.945 | 58.332 | 1.258 |
| | 36 | 48 | **147.335** | **2.360** | 147.405 | 2.362 | 82.184 | 1.609 |
| | 36 | 60 | **202.442** | **2.784** | 202.988 | 2.800 | 109.625 | 1.882 |
| NN5 | 36 | 24 | 0.736 | 0.593 | 0.728 | 0.589 | 0.758 | 0.592 |
| | 104 | 36 | **0.656** | **0.561** | 0.658 | 0.562 | 0.693 | 0.577 |
| | 104 | 48 | **0.626** | **0.550** | 0.646 | 0.562 | 0.688 | 0.587 |
| | 104 | 60 | **0.623** | **0.552** | 0.645 | 0.564 | 0.679 | 0.587 |
| Wike2000 | 36 | 24 | 507.237 | 1.369 | 506.895 | 1.365 | 1135.609 | 1.350 |
| | 36 | 36 | **577.394** | **1.596** | 577.706 | 1.596 | 1060.939 | 1.388 |
| | 36 | 48 | **612.004** | **1.684** | 612.170 | 1.688 | 1899.937 | 1.733 |
| | 36 | 60 | **654.285** | 1.819 | 654.767 | 1.804 | 1281.245 | 1.570 |

*Table 10.* MSE and MAE scores of FEDformer-based models. The lower, the better. DYN score is in **bold** when it is better than the vanilla version. PRO score is underlined when it is better than the DYN one. Vanilla score is starred* when we obtained better results than the TFB benchmark with updated learning hyperparameters.

| Dataset | Cont. L | Hor. H | FEDformer DYN MSE | MAE | FEDformer PRO MSE | MAE | FEDformer MSE | MAE | NLinear MSE | MAE |
|---|---|---|---|---|---|---|---|---|---|---|
| ETTh1 | 96 | 96 | **0.378** | **0.419** | 0.389 | 0.429 | 0.379 | 0.419 | 0.385 | 0.403 |
| | 96 | 192 | **0.419** | **0.442** | 0.420 | 0.443 | 0.420 | 0.444 | 0.422 | 0.426 |
| | 96 | 336 | **0.456** | **0.464** | 0.457 | 0.463 | 0.458 | 0.466 | 0.431 | 0.429 |
| | 96 | 720 | 0.479 | 0.491 | 0.495 | 0.497 | 0.474 | 0.488 | 0.439 | 0.452 |
| ETTh2 | 96 | 96 | 0.338 | 0.380 | 0.337 | 0.380 | 0.337 | 0.380 | 0.276 | 0.338 |
| | 96 | 192 | **0.414** | **0.427** | 0.416 | 0.429 | 0.415 | 0.428 | 0.345 | 0.382 |
| | 512 | 336 | **0.372** | **0.444** | 0.376 | 0.449 | 0.389 | 0.457 | 0.368 | 0.408 |
| | 96 | 720 | **0.475** | **0.483** | 0.481 | 0.486 | 0.483 | 0.486 | 0.406 | 0.441 |
| ETTm1 | 96 | 96 | 0.464 | 0.465 | 0.464 | 0.465 | 0.463 | 0.463 | 0.301 | 0.343 |
| | 96 | 192 | **0.571** | 0.522 | 0.571 | 0.510 | 0.575 | 0.516 | 0.355 | 0.379 |
| | 512 | 336 | 0.632 | 0.554 | 0.746 | 0.603 | 0.618 | 0.544 | 0.372 | 0.385 |
| | 336 | 720 | 0.615 | 0.554 | 0.616 | 0.551 | 0.612 | 0.551 | 0.430 | 0.418 |
| ETTm2 | 96 | 96 | 0.218 | 0.319 | 0.280 | 0.370 | 0.216 | 0.309 | 0.163 | 0.252 |
| | 96 | 192 | 0.318 | 0.377 | 0.321 | 0.382 | 0.297 | 0.360 | 0.218 | 0.290 |
| | 96 | 336 | **0.357** | **0.393** | 0.364 | 0.399 | 0.366 | 0.400 | 0.273 | 0.326 |
| | 96 | 720 | **0.455** | **0.447** | 0.458 | 0.449 | 0.459 | 0.450 | 0.361 | 0.382 |
| Exchange | 96 | 96 | 0.151 | 0.288 | 0.152 | 0.281 | 0.138 | 0.268 | 0.085 | 0.204 |
| | 96 | 192 | **0.269** | **0.376** | 0.234 | 0.355 | 0.273 | 0.379 | 0.175 | 0.297 |
| | 96 | 336 | **0.437** | **0.484** | 0.441 | 0.487 | 0.437 | 0.485 | 0.320 | 0.409 |
| | 96 | 720 | 1.162 | 0.830 | 1.139 | 0.826 | 1.158 | 0.828 | 0.838 | 0.690 |
| Weather | 96 | 96 | 0.267 | 0.339 | 0.274 | 0.342 | 0.229 | 0.298 | 0.180 | 0.226 |
| | 96 | 192 | 0.261 | **0.319** | 0.281 | 0.345 | 0.258* | 0.323* | 0.218 | 0.261 |
| | 336 | 336 | 0.334 | 0.372 | 0.357 | 0.403 | 0.329* | 0.371* | 0.266 | 0.296 |
| | 96 | 720 | **0.418** | **0.411** | 0.420 | 0.414 | 0.423 | 0.418 | 0.334 | 0.345 |
| Electricity | 96 | 96 | 0.192 | 0.307 | 0.191 | 0.306 | 0.191 | 0.305 | 0.140 | 0.236 |
| | 96 | 192 | **0.201** | **0.314** | 0.202 | 0.315 | 0.202* | 0.315* | 0.155 | 0.248 |
| | 96 | 336 | **0.220** | **0.332** | 0.221 | 0.334 | 0.221 | 0.333 | 0.171 | 0.264 |
| | 96 | 720 | **0.254** | **0.360** | 0.256 | 0.362 | 0.256* | 0.362* | 0.210 | 0.297 |
| ILI | 36 | 24 | **2.389** | **1.020** | 2.362 | 1.012 | 2.398 | 1.020 | 1.998 | 0.919 |
| | 36 | 36 | 2.627 | 1.056 | 2.479 | 1.016 | 2.410 | 1.005 | 1.920 | 0.916 |
| | 36 | 48 | 2.673 | 1.059 | 2.771 | 1.079 | 2.591 | 1.033 | 1.895 | 0.924 |
| | 36 | 60 | 2.541 | 1.079 | 2.512 | 1.078 | 2.539 | 1.070 | 1.964 | 0.947 |
| Solar | 96 | 96 | 0.462 | 0.560 | 0.471 | 0.557 | 0.462* | 0.547* | 0.202 | 0.245 |
| | 96 | 192 | **0.403** | 0.514 | 0.379 | 0.469 | 0.408* | 0.477* | 0.223 | 0.258 |
| | 96 | 336 | 1.009 | 0.857 | 0.950 | 0.799 | 1.008 | 0.839 | 0.238 | 0.265 |
| | 96 | 720 | **0.329** | **0.423** | 0.327 | 0.421 | 0.343* | 0.443* | 0.246 | 0.268 |
| Traffic | 96 | 96 | 0.594 | 0.366 | 0.593 | 0.366 | 0.593 | 0.365 | 0.395 | 0.272 |
| | 96 | 192 | 0.612 | **0.379** | 0.611 | 0.379 | 0.614 | 0.381 | 0.407 | 0.277 |
| | 96 | 336 | **0.608** | **0.371** | 0.618 | 0.378 | 0.613* | 0.376* | 0.417 | 0.282 |
| | 96 | 720 | 0.652 | 0.397 | 0.656 | 0.400 | 0.646 | 0.394 | 0.453 | 0.302 |
| PEMS-BAY | 96 | 96 | **0.776** | **0.529** | 0.980 | 0.625 | 0.854 | 0.586 | 0.642 | 0.402 |
| | 96 | 192 | **0.909** | **0.583** | 0.884 | 0.572 | 1.217 | 0.714 | 0.687 | 0.420 |
| | 96 | 336 | **1.173** | **0.677** | 1.175 | 0.695 | 1.184 | 0.689 | 0.735 | 0.437 |
| | 96 | 720 | **1.061** | **0.623** | 1.021 | 0.632 | 1.082 | 0.642 | 0.924 | 0.514 |
| METR-LA | 96 | 96 | **1.308** | **0.787** | 1.285 | 0.792 | 1.464 | 0.813 | 1.042 | 0.651 |
| | 96 | 192 | **1.573** | **0.884** | 1.550 | 0.865 | 2.042 | 0.999 | 1.218 | 0.720 |
| | 96 | 336 | **1.516** | **0.877** | 1.567 | 0.921 | 1.672 | 0.889 | 1.334 | 0.756 |
| | 96 | 720 | **1.676** | **0.883** | 1.778 | 0.902 | 2.015 | 0.963 | 1.683 | 0.886 |
| PEMS04 | 96 | 96 | **0.539** | **0.561** | 0.556 | 0.557 | 0.598 | 0.571 | 0.208 | 0.301 |
| | 96 | 192 | **0.753** | **0.696** | 0.798 | 0.714 | 0.996 | 0.804 | 0.229 | 0.312 |
| | 96 | 336 | 1.826 | 1.073 | 1.869 | 1.081 | 1.817 | 1.067 | 0.251 | 0.331 |
| | 96 | 720 | **0.501** | **0.552** | 0.588 | 0.598 | 1.047 | 0.795 | 0.346 | 0.398 |
| PEMS08 | 96 | 96 | 0.738 | 0.609 | 0.664 | 0.570 | 0.722 | 0.585 | 0.340 | 0.327 |
| | 96 | 192 | **0.937** | **0.705** | 0.861 | 0.670 | 1.159 | 0.810 | 0.450 | 0.353 |
| | 96 | 336 | **2.109** | **1.114** | 2.225 | 1.145 | 2.222 | 1.142 | 0.454 | 0.369 |
| | 96 | 720 | **0.844** | **0.645** | 0.914 | 0.682 | 1.243 | 0.825 | 0.494 | 0.429 |
| AQShunyi | 512 | 96 | **0.694** | **0.517** | 0.694 | 0.516 | 0.706 | 0.525 | 0.653 | 0.486 |
| | 336 | 192 | **0.728** | 0.531 | 0.731 | 0.535 | 0.729 | 0.531 | 0.701 | 0.506 |
| | 96 | 336 | 0.825 | 0.571 | 0.827 | 0.569 | 0.824 | 0.569 | 0.722 | 0.519 |
| | 512 | 720 | **0.791** | **0.559** | 0.793 | 0.561 | 0.794 | 0.561 | 0.777 | 0.545 |
| AQWan | 336 | 96 | **0.790** | **0.507** | 0.796 | 0.508 | 0.796 | 0.508 | 0.758 | 0.475 |
| | 336 | 192 | **0.824** | **0.517** | 0.824 | 0.517 | 0.825 | 0.517 | 0.809 | 0.496 |
| | 336 | 336 | 0.863 | 0.537 | 0.863 | 0.537 | 0.863 | 0.537 | 0.830 | 0.508 |
| | 512 | 720 | **0.905** | **0.551** | 0.907 | 0.552 | 0.907 | 0.552 | 0.906 | 0.538 |
| Wind | 96 | 96 | **0.967** | **0.688** | 1.024 | 0.706 | 1.034 | 0.699 | 0.923 | 0.640 |
| | 96 | 192 | **1.200** | **0.795** | 1.198 | 0.792 | 1.280 | 0.805 | 1.081 | 0.734 |
| | 336 | 336 | **1.284** | **0.850** | 1.318 | 0.868 | 1.302 | 0.864 | 1.228 | 0.805 |
| | 336 | 720 | **1.361** | 0.881 | 1.327 | 0.871 | 1.373 | 0.881 | 1.328 | 0.852 |
| ZafNoo | 96 | 96 | **0.472** | **0.446** | 0.477 | 0.456 | 0.476 | 0.450 | 0.447 | 0.410 |
| | 96 | 192 | **0.541** | **0.479** | 0.545 | 0.482 | 0.544 | 0.479 | 0.503 | 0.447 |
| | 96 | 336 | 0.619 | 0.515 | 0.619 | 0.515 | 0.619* | 0.515* | 0.545 | 0.470 |
| | 512 | 720 | 0.648 | 0.557 | 0.672 | 0.577 | 0.647* | 0.557* | 0.589 | 0.497 |
| CzeLan | 96 | 96 | 0.239 | 0.311 | 0.234 | 0.308 | 0.231 | 0.311 | 0.178 | 0.228 |
| | 96 | 192 | **0.282** | **0.348** | 0.270 | 0.344 | 0.283 | 0.349 | 0.210 | 0.252 |
| | 512 | 336 | **0.294** | **0.362** | 0.295 | 0.362 | 0.298 | 0.363 | 0.243 | 0.280 |
| | 96 | 720 | **0.368** | **0.410** | 0.376 | 0.418 | 0.426 | 0.449 | 0.290 | 0.326 |
| Covid-19 | 36 | 24 | 2.125 | **0.174** | 2.124 | 0.175 | 2.125 | 0.176* | 1.139 | 0.070 |
| | 36 | 36 | 2.463 | 0.177 | 2.489 | 0.228 | 2.463 | 0.177 | 1.582 | 0.091 |
| | 36 | 48 | 2.847 | 0.192 | 2.847 | 0.193 | 2.847 | 0.192 | 1.932 | 0.099 |
| | 36 | 60 | **3.273** | **0.205** | 3.272 | 0.205 | 3.275 | 0.212 | 2.682 | 0.127 |
| NASDAQ | 36 | 24 | **0.462** | **0.453** | 0.481 | 0.473 | 0.537 | 0.481 | 0.557 | 0.522 |
| | 36 | 36 | 0.844 | 0.636 | 0.818 | 0.629 | 0.808 | 0.628 | 0.869 | 0.668 |
| | 36 | 48 | **1.105** | **0.738** | 1.138 | 0.742 | 1.126* | 0.742* | 1.152 | 0.770 |
| | 36 | 60 | 1.252 | 0.786 | 1.229 | 0.783 | 1.251 | 0.783 | 1.284 | 0.809 |
| NYSE | 36 | 24 | 0.229 | 0.337 | 0.261 | 0.359 | 0.159 | 0.254 | 0.193 | 0.283 |
| | 36 | 36 | **0.283** | **0.335** | 0.279 | 0.334 | 0.289 | 0.344 | 0.315 | 0.356 |
| | 36 | 48 | 0.498 | 0.472 | 0.469 | 0.451 | 0.477 | 0.457 | 0.464 | 0.438 |
| | 36 | 60 | **0.649** | **0.549** | 0.669 | 0.551 | 0.693 | 0.586 | 0.631 | 0.522 |
| FRED-MD | 36 | 24 | 66.029 | 1.613 | 66.031 | 1.614 | 66.023* | 1.613* | 32.125 | 0.931 |
| | 36 | 36 | 94.379 | 1.872 | 94.334 | 1.863 | 94.359 | 1.878 | 58.332 | 1.258 |
| | 36 | 48 | 129.920 | **2.132** | 129.865 | 2.130 | 129.798 | 2.135 | 82.184 | 1.609 |
| | 36 | 60 | **173.275** | **2.426** | 173.648 | 2.430 | 173.596* | 2.432* | 109.625 | 1.882 |
| NN5 | 36 | 24 | 0.792 | 0.623 | 0.799 | 0.626 | 0.785 | 0.618 | 0.758 | 0.592 |
| | 36 | 36 | **0.706** | **0.591** | 0.733 | 0.608 | 0.727 | 0.606 | 0.693 | 0.577 |
| | 36 | 48 | 0.648 | 0.568 | 0.660 | 0.570 | 0.623 | 0.555 | 0.688 | 0.587 |
| | 36 | 60 | **0.626** | **0.555** | 0.634 | 0.562 | 0.630 | 0.559 | 0.679 | 0.587 |
| Wike2000 | 36 | 24 | **681.084** | **3.742** | 681.118 | 3.751 | 681.306 | 3.775 | 1135.609 | 1.350 |
| | 36 | 36 | 717.057 | 3.260 | 716.522 | 3.254 | 716.194 | 3.237 | 1060.939 | 1.388 |
| | 36 | 48 | **748.880** | **3.030** | 748.927 | 3.040 | 750.039 | 3.071 | 1899.937 | 1.733 |
| | 36 | 60 | **786.595** | **2.950** | 786.670 | 2.952 | 786.667 | 2.953 | 1281.245 | 1.570 |

*Table 11.* MSE and MAE scores of iTransformer-based models. The lower, the better. DYN-Post-processing score is in **bold** when it is better than the vanilla version.

| Dataset | Cont. $L$ | Hor. $H$ | iTransformer DYN-Post MSE | MAE | iTransformer MSE | MAE | NLinear MSE | MAE |
|---|---|---|---|---|---|---|---|---|
| ETTh1 | 96 | 96 | 0.386 | **0.404** | 0.386 | 0.405 | 0.385 | 0.403 |
| | 512 | 192 | **0.423** | **0.438** | 0.424 | 0.440 | 0.422 | 0.426 |
| | 512 | 336 | **0.439** | **0.452** | 0.449 | 0.460 | 0.431 | 0.429 |
| | 96 | 720 | 0.552 | 0.518 | 0.495 | 0.487 | 0.439 | 0.452 |
| ETTh2 | 96 | 96 | 0.306 | 0.353 | 0.297 | 0.348 | 0.276 | 0.338 |
| | 512 | 192 | **0.370** | **0.402** | 0.372 | 0.403 | 0.345 | 0.382 |
| | 336 | 336 | **0.384** | **0.413** | 0.388 | 0.417 | 0.368 | 0.408 |
| | 96 | 720 | **0.420** | **0.442** | 0.424 | 0.444 | 0.406 | 0.441 |
| ETTm1 | 336 | 96 | 0.301 | 0.356 | 0.300 | 0.353 | 0.301 | 0.343 |
| | 336 | 192 | 0.351 | 0.386 | 0.341 | 0.380 | 0.355 | 0.379 |
| | 336 | 336 | 0.379 | 0.401 | 0.374 | 0.396 | 0.372 | 0.385 |
| | 512 | 720 | **0.428** | **0.429** | 0.429 | 0.430 | 0.430 | 0.418 |
| ETTm2 | 336 | 96 | 0.179 | 0.269 | 0.175 | 0.266 | 0.163 | 0.252 |
| | 336 | 192 | **0.236** | **0.309** | 0.242 | 0.312 | 0.218 | 0.290 |
| | 336 | 336 | 0.283 | 0.337 | 0.282 | 0.337 | 0.273 | 0.326 |
| | 336 | 720 | 0.386 | 0.396 | 0.375 | 0.394 | 0.361 | 0.382 |
| Exchange | 96 | 96 | 0.087 | 0.207 | 0.086 | 0.205 | 0.085 | 0.204 |
| | 96 | 192 | 0.177 | 0.299 | 0.177 | 0.299 | 0.175 | 0.297 |
| | 96 | 336 | 0.331 | 0.417 | 0.331 | 0.417 | 0.320 | 0.409 |
| | 96 | 720 | **0.843** | **0.691** | 0.846 | 0.693 | 0.838 | 0.690 |
| Weather | 336 | 96 | 0.158 | 0.207 | 0.157 | 0.207 | 0.180 | 0.226 |
| | 512 | 192 | 0.203 | 0.249 | 0.200 | 0.248 | 0.218 | 0.261 |
| | 336 | 336 | **0.251** | **0.286** | 0.252 | 0.287 | 0.266 | 0.296 |
| | 512 | 720 | **0.319** | 0.336 | 0.320 | 0.336 | 0.334 | 0.345 |
| Electricity | 336 | 96 | **0.132** | **0.229** | 0.134 | 0.230 | 0.140 | 0.236 |
| | 336 | 192 | **0.153** | **0.248** | 0.154 | 0.250 | 0.155 | 0.248 |
| | 336 | 336 | 0.169 | 0.265 | 0.169 | 0.265 | 0.171 | 0.264 |
| | 336 | 720 | **0.193** | **0.287** | 0.194 | 0.288 | 0.210 | 0.297 |
| ILI | 104 | 24 | **1.632** | **0.793** | 1.783 | 0.846 | 1.998 | 0.919 |
| | 104 | 36 | **1.718** | 0.886 | 1.746 | 0.860 | 1.920 | 0.916 |
| | 104 | 48 | 1.822 | 0.932 | 1.716 | 0.898 | 1.895 | 0.924 |
| | 36 | 60 | **2.151** | **0.953** | 2.183 | 0.963 | 1.964 | 0.947 |
| Solar | 336 | 96 | 0.200 | 0.247 | 0.190 | 0.244 | 0.202 | 0.245 |
| | 512 | 192 | 0.193 | 0.260 | 0.193 | 0.257 | 0.223 | 0.258 |
| | 512 | 336 | **0.200** | **0.264** | 0.203 | 0.266 | 0.238 | 0.265 |
| | 512 | 720 | **0.221** | **0.279** | 0.223 | 0.281 | 0.246 | 0.268 |
| Traffic | 512 | 96 | **0.362** | **0.264** | 0.363 | 0.265 | 0.395 | 0.272 |
| | 512 | 192 | **0.384** | **0.272** | 0.384 | 0.273 | 0.407 | 0.277 |
| | 512 | 336 | **0.395** | 0.278 | 0.396 | 0.277 | 0.417 | 0.282 |
| | 512 | 720 | **0.440** | **0.304** | 0.445 | 0.308 | 0.453 | 0.302 |
| PEMS-BAY | 336 | 96 | 0.562 | 0.339 | 0.498 | 0.319 | 0.642 | 0.402 |
| | 512 | 192 | 0.577 | 0.366 | 0.547 | 0.340 | 0.687 | 0.420 |
| | 512 | 336 | 0.677 | 0.401 | 0.580 | 0.355 | 0.735 | 0.437 |
| | 512 | 720 | 0.741 | 0.426 | 0.662 | 0.391 | 0.924 | 0.514 |
| METR-LA | 512 | 96 | 1.117 | 0.650 | 1.077 | 0.633 | 1.042 | 0.651 |
| | 336 | 192 | 1.284 | **0.672** | 1.267 | 0.690 | 1.218 | 0.720 |
| | 512 | 336 | 1.401 | 0.787 | 1.397 | 0.730 | 1.334 | 0.756 |
| | 512 | 720 | **1.636** | 0.828 | 1.676 | 0.826 | 1.683 | 0.886 |
| PEMS04 | 512 | 96 | 0.127 | 0.235 | 0.125 | 0.230 | 0.208 | 0.301 |
| | 512 | 192 | 0.163 | 0.261 | 0.145 | 0.249 | 0.229 | 0.312 |
| | 512 | 336 | 0.164 | 0.266 | 0.158 | 0.263 | 0.251 | 0.331 |
| | 512 | 720 | 0.204 | 0.305 | 0.195 | 0.300 | 0.346 | 0.398 |
| PEMS08 | 336 | 96 | 0.214 | 0.254 | 0.194 | 0.235 | 0.340 | 0.327 |
| | 336 | 192 | 0.366 | 0.315 | 0.288 | 0.255 | 0.450 | 0.353 |
| | 336 | 336 | 0.440 | 0.343 | 0.345 | 0.276 | 0.454 | 0.369 |
| | 512 | 720 | **0.353** | **0.309** | 0.366 | 0.318 | 0.494 | 0.429 |
| AQShunyi | 512 | 96 | **0.647** | 0.481 | 0.650 | 0.479 | 0.653 | 0.486 |
| | 336 | 192 | **0.689** | 0.500 | 0.693 | 0.498 | 0.701 | 0.506 |
| | 336 | 336 | 0.713 | 0.511 | 0.713 | 0.510 | 0.722 | 0.519 |
| | 512 | 720 | 0.776 | 0.547 | 0.766 | 0.537 | 0.777 | 0.545 |
| AQWan | 512 | 96 | **0.745** | **0.469** | 0.747 | 0.470 | 0.758 | 0.475 |
| | 512 | 192 | 0.793 | 0.492 | 0.787 | 0.486 | 0.809 | 0.496 |
| | 336 | 336 | **0.808** | 0.498 | 0.814 | 0.497 | 0.830 | 0.508 |
| | 336 | 720 | 0.899 | 0.534 | 0.889 | 0.529 | 0.906 | 0.538 |
| Wind | 512 | 96 | **0.900** | 0.648 | 0.901 | 0.646 | 0.923 | 0.640 |
| | 512 | 192 | 1.089 | 0.742 | 1.085 | 0.740 | 1.081 | 0.734 |
| | 512 | 336 | 1.228 | 0.807 | 1.222 | 0.805 | 1.228 | 0.805 |
| | 512 | 720 | 1.348 | 0.860 | 1.325 | 0.850 | 1.328 | 0.852 |
| ZafNoo | 336 | 96 | 0.439 | 0.410 | 0.439 | 0.408 | 0.447 | 0.410 |
| | 336 | 192 | **0.503** | 0.449 | 0.505 | 0.443 | 0.503 | 0.447 |
| | 336 | 336 | 0.573 | 0.489 | 0.555 | 0.473 | 0.545 | 0.470 |
| | 512 | 720 | 0.619 | 0.519 | 0.591 | 0.501 | 0.589 | 0.497 |
| CzeLan | 512 | 96 | **0.170** | **0.234** | 0.177 | 0.239 | 0.178 | 0.228 |
| | 512 | 192 | 0.203 | 0.259 | 0.201 | 0.257 | 0.210 | 0.252 |
| | 512 | 336 | 0.234 | 0.287 | 0.232 | 0.282 | 0.243 | 0.280 |
| | 512 | 720 | **0.258** | **0.308** | 0.261 | 0.311 | 0.290 | 0.326 |
| Covid-19 | 36 | 24 | **0.932** | **0.035** | 1.001 | 0.038 | 1.139 | 0.070 |
| | 36 | 36 | **1.227** | **0.041** | 1.236 | 0.042 | 1.582 | 0.091 |
| | 36 | 48 | **1.658** | **0.053** | 1.710 | 0.056 | 1.932 | 0.099 |
| | 36 | 60 | 2.036 | **0.061** | 2.005 | 0.062 | 2.682 | 0.127 |
| NASDAQ | 104 | 24 | **0.506** | **0.513** | 0.570 | 0.540 | 0.557 | 0.522 |
| | 104 | 36 | 0.855 | 0.695 | 0.691 | 0.600 | 0.869 | 0.668 |
| | 104 | 48 | **1.091** | **0.758** | 1.188 | 0.773 | 1.152 | 0.770 |
| | 36 | 60 | 1.425 | 0.839 | 1.325 | 0.820 | 1.284 | 0.809 |
| NYSE | 36 | 24 | 0.225 | 0.304 | 0.225 | 0.302 | 0.193 | 0.283 |
| | 36 | 36 | **0.390** | 0.418 | 0.392 | 0.409 | 0.315 | 0.356 |
| | 36 | 48 | **0.510** | **0.460** | 0.529 | 0.480 | 0.464 | 0.438 |
| | 36 | 60 | 0.700 | **0.554** | 0.687 | 0.557 | 0.631 | 0.522 |
| FRED-MD | 36 | 24 | **27.040** | **0.903** | 28.581 | 0.917 | 32.125 | 0.931 |
| | 36 | 36 | **48.389** | **1.226** | 54.221 | 1.276 | 58.332 | 1.258 |
| | 36 | 48 | **85.573** | **1.563** | 89.574 | 1.607 | 82.184 | 1.609 |
| | 36 | 60 | **124.354** | **1.918** | 130.061 | 1.947 | 109.625 | 1.882 |
| NN5 | 104 | 24 | **0.719** | 0.570 | 0.727 | 0.568 | 0.758 | 0.592 |
| | 104 | 36 | **0.658** | 0.552 | 0.664 | 0.552 | 0.693 | 0.577 |
| | 104 | 48 | **0.632** | 0.546 | 0.633 | 0.543 | 0.688 | 0.587 |
| | 104 | 60 | 0.617 | 0.541 | 0.615 | 0.537 | 0.679 | 0.587 |
| Wike2000 | 36 | 24 | 458.891 | 1.017 | 453.475 | 1.011 | 1135.609 | 1.350 |
| | 36 | 36 | 635.855 | 1.225 | 515.830 | 1.132 | 1060.939 | 1.388 |
| | 36 | 48 | **539.686** | **1.176** | 578.335 | 1.214 | 1899.937 | 1.733 |
| | 104 | 60 | 644.895 | **1.398** | 634.947 | 1.402 | 1281.245 | 1.570 |

*Table 12.* MSE and MAE scores of PatchTST-based models. The lower, the better. DYN-Post-processing score is in **bold** when it is better than the vanilla version.

| Dataset | Cont. L | Hor. H | PatchTST DYN-Post MSE | MAE | PatchTST MSE | MAE | NLinear MSE | MAE |
|---|---|---|---|---|---|---|---|---|
| ETTh1 | 96 | 96 | 0.379 | **0.395** | 0.377 | 0.397 | 0.385 | 0.403 |
| | 512 | 192 | **0.401** | **0.418** | 0.409 | 0.425 | 0.422 | 0.426 |
| | 512 | 336 | **0.423** | **0.433** | 0.431 | 0.444 | 0.431 | 0.429 |
| | 512 | 720 | 0.457 | **0.474** | 0.457 | 0.477 | 0.439 | 0.452 |
| ETTh2 | 512 | 96 | 0.281 | 0.345 | 0.274 | 0.337 | 0.276 | 0.338 |
| | 512 | 192 | **0.341** | **0.382** | 0.348 | 0.384 | 0.345 | 0.382 |
| | 512 | 336 | **0.372** | **0.413** | 0.377 | 0.416 | 0.368 | 0.408 |
| | 336 | 720 | 0.415 | 0.445 | 0.406 | 0.441 | 0.406 | 0.441 |
| ETTm1 | 336 | 96 | 0.290 | 0.345 | 0.289 | 0.343 | 0.301 | 0.343 |
| | 336 | 192 | **0.328** | **0.367** | 0.329 | 0.368 | 0.355 | 0.379 |
| | 336 | 336 | **0.361** | **0.386** | 0.362 | 0.390 | 0.372 | 0.385 |
| | 512 | 720 | 0.417 | **0.421** | 0.416 | 0.423 | 0.430 | 0.418 |
| ETTm2 | 336 | 96 | 0.168 | 0.256 | 0.165 | 0.255 | 0.163 | 0.252 |
| | 336 | 192 | 0.223 | 0.296 | 0.221 | 0.293 | 0.218 | 0.290 |
| | 512 | 336 | **0.274** | 0.332 | 0.276 | 0.327 | 0.273 | 0.326 |
| | 336 | 720 | 0.367 | 0.389 | 0.362 | 0.381 | 0.361 | 0.382 |
| Exchange | 96 | 96 | **0.084** | **0.201** | 0.087 | 0.204 | 0.085 | 0.204 |
| | 96 | 192 | 0.177 | **0.298** | 0.177 | 0.300 | 0.175 | 0.297 |
| | 512 | 336 | 0.338 | 0.424 | 0.297 | 0.399 | 0.320 | 0.409 |
| | 96 | 720 | 0.885 | 0.709 | 0.843 | 0.692 | 0.838 | 0.690 |
| Weather | 336 | 96 | 0.156 | 0.207 | 0.150 | 0.200 | 0.180 | 0.226 |
| | 512 | 192 | 0.195 | 0.246 | 0.191 | 0.239 | 0.218 | 0.261 |
| | 512 | 336 | 0.247 | 0.283 | 0.242 | 0.279 | 0.266 | 0.296 |
| | 512 | 720 | 0.319 | 0.336 | 0.312 | 0.330 | 0.334 | 0.345 |
| Electricity | 512 | 96 | 0.151 | 0.257 | 0.143 | 0.247 | 0.140 | 0.236 |
| | 512 | 192 | 0.162 | 0.266 | 0.158 | 0.260 | 0.155 | 0.248 |
| | 512 | 336 | 0.167 | **0.264** | 0.168 | 0.267 | 0.171 | 0.264 |
| | 512 | 720 | 0.215 | 0.307 | 0.214 | 0.307 | 0.210 | 0.297 |
| ILI | 104 | 24 | 2.204 | 0.998 | 1.932 | 0.872 | 1.998 | 0.919 |
| | 104 | 36 | 2.162 | 0.995 | 1.869 | 0.866 | 1.920 | 0.916 |
| | 104 | 48 | 2.193 | 0.998 | 1.891 | 0.883 | 1.895 | 0.924 |
| | 104 | 60 | 2.177 | 1.000 | 1.914 | 0.896 | 1.964 | 0.947 |
| Solar | 512 | 96 | 0.181 | 0.239 | 0.170 | 0.234 | 0.202 | 0.245 |
| | 512 | 192 | 0.207 | **0.289** | 0.203* | 0.302 | 0.223 | 0.258 |
| | 512 | 336 | 0.216 | 0.305 | 0.212 | 0.293 | 0.238 | 0.265 |
| | 512 | 720 | 0.223 | 0.309 | 0.215 | 0.307 | 0.246 | 0.268 |
| Traffic | 512 | 96 | 0.385 | 0.275 | 0.370 | 0.262 | 0.395 | 0.272 |
| | 512 | 192 | 0.397 | 0.281 | 0.386 | 0.269 | 0.407 | 0.277 |
| | 512 | 336 | 0.409 | 0.288 | 0.396 | 0.275 | 0.417 | 0.282 |
| | 512 | 720 | 0.446 | 0.309 | 0.435 | 0.295 | 0.453 | 0.302 |
| PEMS-BAY | 512 | 96 | 0.586 | 0.367 | 0.566 | 0.355 | 0.642 | 0.402 |
| | 512 | 192 | 0.654 | 0.401 | 0.649 | 0.394 | 0.687 | 0.420 |
| | 512 | 336 | 0.701 | 0.416 | 0.700 | 0.412 | 0.735 | 0.437 |
| | 512 | 720 | **0.830** | 0.462 | 0.843 | 0.460 | 0.924 | 0.514 |
| METR-LA | 512 | 96 | 1.036 | 0.647 | 1.028 | 0.643 | 1.042 | 0.651 |
| | 512 | 192 | 1.209 | 0.714 | 1.156 | 0.683 | 1.218 | 0.720 |
| | 512 | 336 | 1.324 | 0.743 | 1.252 | 0.736 | 1.334 | 0.756 |
| | 512 | 720 | 1.557 | 0.845 | 1.433 | 0.793 | 1.683 | 0.886 |
| PEMS04 | 336 | 96 | 0.174 | 0.284 | 0.169 | 0.282 | 0.208 | 0.301 |
| | 336 | 192 | 0.205 | 0.309 | 0.202 | 0.309 | 0.229 | 0.312 |
| | 336 | 336 | 0.218 | 0.320 | 0.211 | 0.315 | 0.251 | 0.331 |
| | 336 | 720 | **0.256** worst case | **0.351** worst case | 0.257 | 0.352 | 0.346 | 0.398 |
| PEMS08 | 512 | 96 | **0.240** | **0.281** | 0.248 | 0.287 | 0.340 | 0.327 |
| | 512 | 192 | 0.330 | **0.301** | 0.319 | 0.310 | 0.450 | 0.353 |
| | 512 | 336 | 0.382 | **0.323** | 0.361 | 0.324 | 0.454 | 0.369 |
| | 512 | 720 | **0.396** | **0.351** | 0.399 | 0.368 | 0.494 | 0.429 |
| AQShunyi | 512 | 96 | 0.651 | 0.483 | 0.646 | 0.478 | 0.653 | 0.486 |
| | 512 | 192 | 0.691 | 0.502 | 0.688 | 0.498 | 0.701 | 0.506 |
| | 512 | 336 | 0.713 | 0.515 | 0.710 | 0.513 | 0.722 | 0.519 |
| | 512 | 720 | 0.769 | 0.540 | 0.768 | 0.539 | 0.777 | 0.545 |
| AQWan | 512 | 96 | 0.750 | 0.472 | 0.745 | 0.468 | 0.758 | 0.475 |
| | 512 | 192 | 0.797 | 0.493 | 0.793 | 0.490 | 0.809 | 0.496 |
| | 512 | 336 | 0.820 | 0.504 | 0.819 | 0.502 | 0.830 | 0.508 |
| | 512 | 720 | 0.894 | 0.535 | 0.890 | 0.533 | 0.906 | 0.538 |
| Wind | 512 | 96 | 0.908 | 0.653 | 0.877 | 0.643 | 0.923 | 0.640 |
| | 512 | 192 | 1.090 | 0.743 | 1.065 | 0.741 | 1.081 | 0.734 |
| | 512 | 336 | 1.229 | 0.808 | 1.202 | 0.802 | 1.228 | 0.805 |
| | 512 | 720 | 1.337 | 0.861 | 1.300 | 0.846 | 1.328 | 0.852 |
| ZafNoo | 512 | 96 | 0.443 | 0.419 | 0.429 | 0.405 | 0.447 | 0.410 |
| | 512 | 192 | 0.502 | 0.456 | 0.494 | 0.449 | 0.503 | 0.447 |
| | 512 | 336 | 0.545 | 0.479 | 0.538 | 0.475 | 0.545 | 0.470 |
| | 512 | 720 | 0.583 | 0.499 | 0.573 | 0.486 | 0.589 | 0.497 |
| CzeLan | 512 | 96 | 0.184 | 0.240 | 0.176 | 0.232 | 0.178 | 0.228 |
| | 512 | 192 | 0.211 | 0.268 | 0.205 | 0.263 | 0.210 | 0.252 |
| | 512 | 336 | **0.232** | **0.279** | 0.236 | 0.286 | 0.243 | 0.280 |
| | 512 | 720 | **0.269** | **0.313** | 0.270 | 0.316 | 0.290 | 0.326 |
| Covid-19 | 36 | 24 | **1.016** | 0.042 | 1.045 | 0.042 | 1.139 | 0.070 |
| | 36 | 36 | **1.372** | 0.052 | 1.397 | 0.051 | 1.582 | 0.091 |
| | 36 | 48 | **1.692** | **0.060** | 1.769 | 0.062 | 1.932 | 0.099 |
| | 36 | 60 | **2.161** | 0.069 | 2.216 | 0.068 | 2.682 | 0.127 |
| NASDAQ | 36 | 24 | 0.677 | 0.607 | 0.649 | 0.567 | 0.557 | 0.522 |
| | 104 | 36 | 0.834 | **0.677** | 0.821 | 0.682 | 0.869 | 0.668 |
| | 104 | 48 | **1.021** | **0.752** | 1.169 | 0.793 | 1.152 | 0.770 |
| | 104 | 60 | 1.247 | **0.842** | 1.247 | 0.843 | 1.284 | 0.809 |
| NYSE | 36 | 24 | 0.232 | 0.305 | 0.226 | 0.296 | 0.193 | 0.283 |
| | 36 | 36 | **0.350** | **0.370** | 0.380 | 0.389 | 0.315 | 0.356 |
| | 36 | 48 | **0.483** | **0.436** | 0.575 | 0.492 | 0.464 | 0.438 |
| | 36 | 60 | **0.642** | **0.523** | 0.749 | 0.572 | 0.631 | 0.522 |
| FRED-MD | 36 | 24 | 33.754 | 1.007 | 32.808 | 0.962 | 32.125 | 0.931 |
| | 36 | 36 | 61.611 | 1.358 | 61.035 | 1.345 | 58.332 | 1.258 |
| | 36 | 48 | 92.930 | 1.657 | 91.835 | 1.648 | 82.184 | 1.609 |
| | 36 | 60 | 130.365 | 2.026 | 127.018 | 1.958 | 109.625 | 1.882 |
| NN5 | 104 | 24 | **0.726** | **0.590** | 0.740 | 0.596 | 0.758 | 0.592 |
| | 104 | 36 | **0.679** | **0.582** | 0.694 | 0.595 | 0.693 | 0.577 |
| | 104 | 48 | **0.648** | **0.571** | 0.667 | 0.585 | 0.688 | 0.587 |
| | 104 | 60 | **0.643** | **0.576** | 0.653 | 0.582 | 0.679 | 0.587 |
| Wike2000 | 36 | 24 | **449.087** | 1.040 | 457.183 | 1.023 | 1135.609 | 1.350 |
| | 36 | 36 | **494.895** | 1.184 | 511.944 | 1.115 | 1060.939 | 1.388 |
| | 36 | 48 | **508.306** | 1.219 | 531.900 | 1.179 | 1899.937 | 1.733 |
| | 36 | 60 | 629.172 | 1.336 | 554.829 | 1.244 | 1281.245 | 1.570 |

*Table 13.* MSE and MAE scores of Crossformer-based models. The lower, the better. DYN-Post-processing score is in **bold** when it is better than the vanilla version. Vanilla score is starred* when we obtained better results than the TFB benchmark with updated learning hyperparameters.

| Dataset | Cont. $L$ | Hor. $H$ | Crossformer DYN-Post MSE | MAE | Crossformer MSE | MAE | NLinear MSE | MAE |
|---|---|---|---|---|---|---|---|---|
| ETTh1 | 96 | 96 | 0.439 | 0.452 | 0.411 | 0.435 | 0.385 | 0.403 |
| | 512 | 192 | 0.436 | 0.457 | 0.409 | 0.438 | 0.422 | 0.426 |
| | 512 | 336 | 0.460 | 0.472 | 0.433 | 0.457 | 0.431 | 0.429 |
| | 512 | 720 | 0.515 | 0.525 | 0.501 | 0.514 | 0.439 | 0.452 |
| ETTh2 | 336 | 96 | **0.670** | **0.579** | 0.728 | 0.603 | 0.276 | 0.338 |
| | 336 | 192 | 0.843 | 0.659 | 0.723 | 0.607 | 0.345 | 0.382 |
| | 336 | 336 | 0.845 | 0.659 | 0.740 | 0.628 | 0.368 | 0.408 |
| | 336 | 720 | **1.090** | **0.781** | 1.386 | 0.882 | 0.406 | 0.441 |
| ETTm1 | 512 | 96 | 0.336 | 0.386 | 0.314 | 0.367 | 0.301 | 0.343 |
| | 512 | 192 | 0.406 | 0.431 | 0.374 | 0.410 | 0.355 | 0.379 |
| | 512 | 336 | 0.415 | 0.435 | 0.413 | 0.432 | 0.372 | 0.385 |
| | 336 | 720 | **0.720** | **0.582** | 0.753 | 0.613 | 0.430 | 0.418 |
| ETTm2 | 512 | 96 | **0.266** | **0.361** | 0.296 | 0.391 | 0.163 | 0.252 |
| | 512 | 192 | 0.474 | 0.472 | 0.369 | 0.416 | 0.218 | 0.290 |
| | 96 | 336 | **0.530** | **0.518** | 0.588 | 0.600 | 0.273 | 0.326 |
| | 512 | 720 | 1.563 | 0.861 | 0.750 | 0.612 | 0.361 | 0.382 |
| Exchange | 96 | 96 | **0.222** | **0.350** | 0.231 | 0.356 | 0.085 | 0.204 |
| | 96 | 192 | 0.480 | 0.522 | 0.460 | 0.509 | 0.175 | 0.297 |
| | 96 | 336 | **0.769** | **0.701** | 1.034 | 0.825 | 0.320 | 0.409 |
| | 336 | 720 | **1.253** | **0.949** | 1.576 | 1.021 | 0.838 | 0.690 |
| Weather | 512 | 96 | 0.154 | 0.229 | 0.143 | 0.210 | 0.180 | 0.226 |
| | 336 | 192 | 0.198 | 0.264 | 0.195 | 0.260 | 0.218 | 0.261 |
| | 96 | 336 | 0.276 | 0.340 | 0.254 | 0.319 | 0.266 | 0.296 |
| | 512 | 720 | 0.346 | 0.388 | 0.335 | 0.385 | 0.334 | 0.345 |
| Electricity | 512 | 96 | 0.136 | 0.235 | 0.134 | 0.231 | 0.140 | 0.236 |
| | 512 | 192 | 0.151 | 0.248 | 0.146 | 0.243 | 0.155 | 0.248 |
| | 512 | 336 | 0.186 | 0.280 | 0.165 | 0.264 | 0.171 | 0.264 |
| | 512 | 720 | 0.237 | **0.313** | 0.237 | 0.314 | 0.210 | 0.297 |
| ILI | 104 | 24 | 3.222 | 1.197 | 2.981 | 1.096 | 1.998 | 0.919 |
| | 104 | 36 | 3.903 | 1.356 | 3.549 | 1.196 | 1.920 | 0.916 |
| | 104 | 48 | 4.032 | 1.377 | 3.851 | 1.288 | 1.895 | 0.924 |
| | 104 | 60 | 4.896 | 1.509 | 4.692 | 1.450 | 1.964 | 0.947 |
| Solar | 512 | 96 | **0.164** | 0.231 | 0.183 | 0.208 | 0.202 | 0.245 |
| | 512 | 192 | **0.206** | 0.244 | 0.208 | 0.226 | 0.223 | 0.258 |
| | 512 | 336 | **0.206** | 0.260 | 0.212 | 0.239 | 0.238 | 0.265 |
| | 512 | 720 | **0.201** | **0.237** | 0.215 | 0.256 | 0.246 | 0.268 |
| Traffic | 96 | 96 | 0.547 | 0.311 | 0.526 | 0.288 | 0.395 | 0.272 |
| | 336 | 192 | **0.489** | 0.273 | 0.503 | 0.263 | 0.407 | 0.277 |
| | 512 | 336 | 0.528 | 0.298 | 0.505 | 0.276 | 0.417 | 0.282 |
| | 336 | 720 | 0.600 | 0.338 | 0.552 | 0.301 | 0.453 | 0.302 |
| PEMS-BAY | 512 | 96 | 0.454 | 0.305 | 0.435 | 0.297 | 0.642 | 0.402 |
| | 512 | 192 | 0.472 | 0.319 | 0.470 | 0.317 | 0.687 | 0.420 |
| | 512 | 336 | **0.493** | **0.323** | 0.495 | 0.326 | 0.735 | 0.437 |
| | 512 | 720 | 0.607 | **0.363** | 0.605 | 0.364 | 0.924 | 0.514 |
| METR-LA | 512 | 96 | 1.212 | 0.648 | 1.069 | 0.606 | 1.042 | 0.651 |
| | 512 | 192 | 1.392 | **0.660** | 1.166 | 0.689 | 1.218 | 0.720 |
| | 512 | 336 | **1.383** | **0.776** | 1.405 | 0.777 | 1.334 | 0.756 |
| | 512 | 720 | 1.566 | **0.782** | 1.421 | 0.808 | 1.683 | 0.886 |
| PEMS04 | 336 | 96 | 0.123 | 0.237 | 0.122 | 0.236 | 0.208 | 0.301 |
| | 336 | 192 | **0.168** | **0.267** | 0.173 | 0.271 | 0.229 | 0.312 |
| | 336 | 336 | **0.188** | **0.298** | 0.208 | 0.307 | 0.251 | 0.331 |
| | 512 | 720 | **0.239** | **0.333** | 0.279 | 0.362 | 0.346 | 0.398 |
| PEMS08 | 512 | 96 | **0.201** | **0.235** | 0.230 | 0.260 | 0.340 | 0.327 |
| | 512 | 192 | **0.226** | **0.249** | 0.239 | 0.264 | 0.450 | 0.353 |
| | 512 | 336 | **0.245** | **0.268** | 0.272 | 0.289 | 0.454 | 0.369 |
| | 512 | 720 | **0.301** | **0.304** | 0.320 | 0.316 | 0.494 | 0.429 |
| AQShunyi | 512 | 96 | 0.666 | 0.503 | 0.652 | 0.484 | 0.653 | 0.486 |
| | 512 | 192 | 0.725 | 0.519 | 0.674 | 0.499 | 0.701 | 0.506 |
| | 336 | 336 | 0.710 | 0.517 | 0.704 | 0.515 | 0.722 | 0.519 |
| | 512 | 720 | 0.764 | 0.535 | 0.747 | 0.518 | 0.777 | 0.545 |
| AQWan | 512 | 96 | 0.757 | 0.473 | 0.750 | 0.465 | 0.758 | 0.475 |
| | 512 | 192 | 0.799 | 0.495 | 0.762 | 0.479 | 0.809 | 0.496 |
| | 336 | 336 | 0.805 | 0.505 | 0.802 | 0.504 | 0.830 | 0.508 |
| | 512 | 720 | 0.837 | 0.515 | 0.829 | 0.512 | 0.906 | 0.538 |
| Wind | 512 | 96 | 0.836 | 0.630 | 0.784 | 0.590 | 0.923 | 0.640 |
| | 512 | 192 | 1.097 | 0.739 | 0.977 | 0.697 | 1.081 | 0.734 |
| | 512 | 336 | 1.162 | 0.776 | 1.073 | 0.755 | 1.228 | 0.805 |
| | 96 | 720 | 1.201 | 0.808 | 1.191 | 0.803 | 1.328 | 0.852 |
| ZafNoo | 96 | 96 | 0.433 | 0.421 | 0.430 | 0.418 | 0.447 | 0.410 |
| | 96 | 192 | 0.482 | **0.448** | 0.479 | 0.449 | 0.503 | 0.447 |
| | 96 | 336 | 0.513 | 0.469 | 0.505 | 0.464 | 0.545 | 0.470 |
| | 96 | 720 | 0.565 | 0.503 | 0.560 | 0.494 | 0.589 | 0.497 |
| CzeLan | 512 | 96 | 0.705 | 0.496 | 0.581 | 0.443 | 0.178 | 0.228 |
| | 336 | 192 | 0.810 | 0.562 | 0.705 | 0.503 | 0.210 | 0.252 |
| | 336 | 336 | 1.016 | 0.625 | 0.971 | 0.596 | 0.243 | 0.280 |
| | 96 | 720 | 1.655 | 0.797 | 1.566 | 0.762 | 0.290 | 0.326 |
| Covid-19 | 104 | 24 | 1771.717 | 2.427 | 1768.817 | 2.314 | 1.139 | 0.070 |
| | 104 | 36 | 1773.242 | 2.507 | 1770.939 | 2.346 | 1.582 | 0.091 |
| | 36 | 48 | **1772.789** | **2.443** | 1773.447 | 2.450 | 1.932 | 0.099 |
| | 104 | 60 | **1772.296** | **2.456** | 1772.833 | 2.486 | 2.682 | 0.127 |
| NASDAQ | 36 | 24 | **1.123** | **0.737** | 1.149 | 0.745 | 0.557 | 0.522 |
| | 36 | 36 | 1.716 | 0.969 | 1.414 | 0.885 | 0.869 | 0.668 |
| | 36 | 48 | 2.204 | 1.143 | 2.108 | 1.136 | 1.152 | 0.770 |
| | 104 | 60 | 2.801 | 1.355 | 2.276 | 1.201 | 1.284 | 0.809 |
| NYSE | 36 | 24 | **0.478** | **0.631** | 0.820 | 0.841 | 0.193 | 0.283 |
| | 36 | 36 | **0.854** | **0.830** | 0.942 | 0.904 | 0.315 | 0.356 |
| | 36 | 48 | **0.758** | **0.751** | 1.049 | 0.955 | 0.464 | 0.438 |
| | 36 | 60 | 1.353 | 1.018 | 1.121 | 0.937 | 0.631 | 0.522 |
| FRED-MD | 104 | 24 | **381.897** | 3.785 | 385.599 | 3.559 | 32.125 | 0.931 |
| | 104 | 36 | **395.499** | 3.887 | 398.728 | 3.716 | 58.332 | 1.258 |
| | 104 | 48 | 414.542 | 4.300 | 414.353 | 3.939 | 82.184 | 1.609 |
| | 104 | 60 | 423.996 | 4.690 | 422.864 | 4.093 | 109.625 | 1.882 |
| NN5 | 104 | 24 | **0.732** | **0.590** | 0.734* | 0.584* | 0.758 | 0.592 |
| | 104 | 36 | **0.674** | **0.570** | 0.678* | 0.589 | 0.693 | 0.577 |
| | 104 | 48 | 0.660 | 0.570 | 0.634* | 0.551* | 0.688 | 0.587 |
| | 104 | 60 | 0.665 | 0.580 | 0.660* | 0.570* | 0.679 | 0.587 |
| Wike2000 | 104 | 24 | 639.111 | 1.873 | 638.794 | 1.734 | 1135.609 | 1.350 |
| | 104 | 36 | **692.271** | 1.845 | 692.485 | 1.808 | 1060.939 | 1.388 |
| | 104 | 48 | **717.538** | 1.976 | 718.072 | 1.899 | 1899.937 | 1.733 |
| | 104 | 60 | 730.390 | **1.983** | 730.368 | 2.021 | 1281.245 | 1.570 |

## G.2. Detailed Results of Section 4.2

Detailed results of Section 4.2 are presented here. For Figure 4, counts of the distribution are shown, with the p-values $p_v$ of MSE and MAE. For RQ1 models, in Table 14, to assess statistical significance, we perform the following unilateral Wilcoxon test: $H_0$: *"DYN added model MAE (or MSE) is not statistically lower than its vanilla version"*, and $H_1$: *"DYN added model MAE (or MSE) is statistically lower than its vanilla version"*. For RQ2 models, in Table 15, the performed unilateral Wilcoxon test is: $H_0$: *"Vanilla model MAE (or MSE) is not statistically lower than its post-processing version"*, and $H_1$: *"Vanilla model MAE (or MSE) is statistically lower than its post-processing version"*.

*Table 14.* Detailed counts of RQ1 `DYN` added models relative performance against their vanilla version with p-values $p_v$ (in **bold** if less than 0.05).

| Model | Better | Iso | Worse $\leq 1\%$ | Worse $> 1\%$ | $p_{v,MSE}$ | $p_{v,MAE}$ |
|---|---|---|---|---|---|---|
| Informer | 153 (77%) | 4 (2%) | 13 (6%) | 30 (15%) | **4.0**e**−8** | **9.7**e**−11** |
| FiLM | 169 (85%) | 11 (5%) | 6 (3%) | 14 (7%) | **7.6**e**−11** | **1.2**e**−11** |
| MICN | 101 (51%) | 31 (15%) | 28 (14%) | 40 (20%) | **1.3**e**−2** | 6.5e−2 |
| FEDformer | 113 (57%) | 20 (10%) | 34 (17%) | 33 (16%) | **3.8**e**−3** | **2.3**e**−2** |

*Table 15.* Detailed counts of RQ2 post-processing models relative performance against their vanilla version with p-values $p_v$ (in **bold** if less than 0.05).

| Model | Better | Iso | Worse $\leq 1\%$ | Worse $> 1\%$ | $p_{v,MSE}$ | $p_{v,MAE}$ |
|---|---|---|---|---|---|---|
| iTransformer | 85 (43%) | 16 (8%) | 31 (15%) | 68 (34%) | 0.18 | **1.1**e**−2** |
| PatchTST | 60 (30%) | 7 (4%) | 38 (19%) | 95 (47%) | **1.0**e**−3** | **4.8**e**−5** |
| Crossformer | 65 (32%) | 1 (1%) | 29 (15%) | 105 (52%) | **7.9**e**−3** | **2.4**e**−4** |

Figure 7 is a detailed version of Table 3 where it proposes a visual of the performance shift due to the addition of a linear `DYN` layer. It also adds the average gain and loss of the dynamics addition, normalized by NLinear scores. It shows that **the mean performance is better and the average gain is greater than the average loss** with the `DYN` layer addition, supporting our model update. The average gain is greater on Informer, which is the vanilla model with the most basic non learnable `DYN` function (0-padding).

Then, in Figure 8, we show the same scatter plot with the outliers that were removed for visualization and consistency in Figure 7. Outliers, with $q_\alpha$ the $\alpha$-quantile where $\alpha \in [0, 1]$, were the ones:

- upper than $q_{0.985}$ and lower than $q_{0.015}$ along both $x$ and $y$ axis for Informer,

- none for FiLM,

- upper than $q_{0.965}$ and lower than $q_{0.035}$ along both $x$ and $y$ axis for MICN,

- upper than $q_{0.99}$ and lower than $q_{0.01}$ along both $x$ and $y$ axis for FEDformer.

Tendencies are the same as in Figure 7, except for Informer, where the average loss is greater than the average gain. That is due to the Traffic case when $H = 192$ where the `DYN` added model is worse than the vanilla version by a large margin, which is an isolated case, removed for consistency.

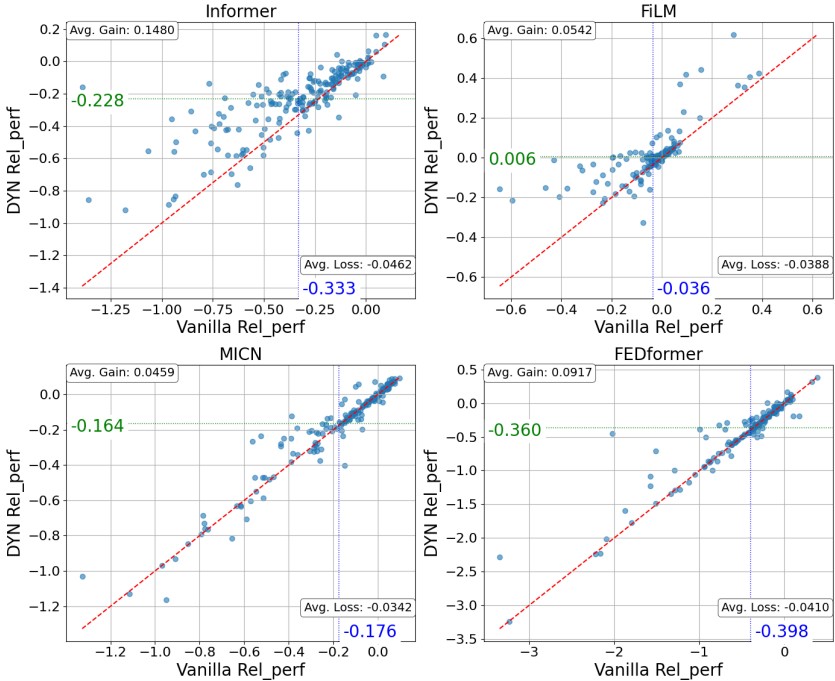

*Figure 7.* Comparison between DYN added model performance against their vanilla version, normalized by NLinear scores. Each point corresponds to $(x; y) : (\text{Rel\_perf}(\text{Vanilla}\|\text{NLinear}); \text{Rel\_perf}(\text{DYN}\|\text{NLinear}))$, where $\text{Rel\_perf}(.\|\text{NLinear}) = \frac{\text{score}(\text{NLinear})-\text{score}(.)}{\text{score}(\text{NLinear})}$, where score is MSE or MAE, for each dataset and forecasting horizon. Points above the diagonal indicate improvement with DYN addition. The mean is shown on the corresponding axis (reported in Table 3). Average gain is $\mathbb{E}[\,y-x \mid y > x\,]$, while average loss is $\mathbb{E}[\,y-x \mid y < x\,]$. Outliers are removed for clarity and consistency (see Figure 8 below with the outliers).

.

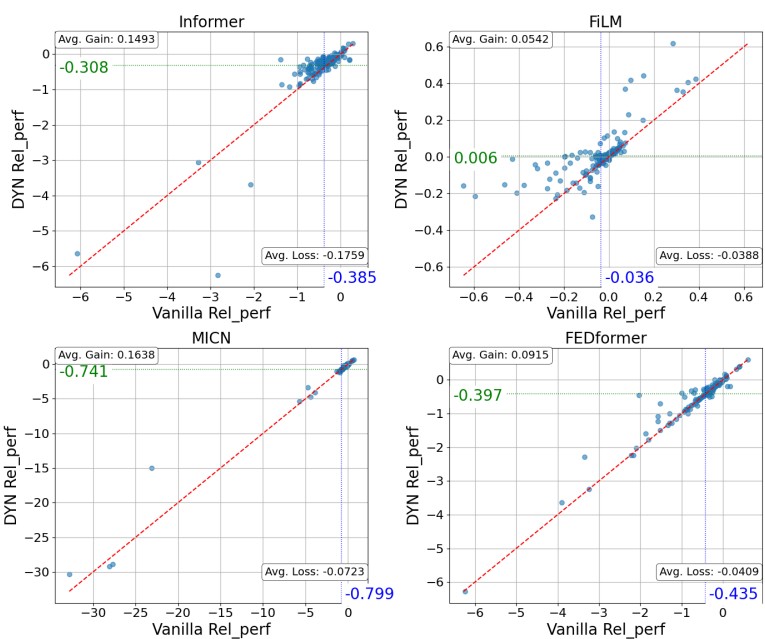

*Figure 8.* Same scatter plot as in Figure 7 above with outliers included.

.

### G.3. Detailed Results of Section 4.3

Detailed results of Section 4.3 are presented here. Again, RQ1 `DYN` added models are compared to their `PRO` versions, while RQ2 post-processing models are compared to their vanilla versions. For Figure 5, counts of the distribution are shown, with the p-values $p_v$ of MSE and MAE in Table 16 for RQ1 and in Table 17 for RQ2. For RQ1 models, in Table 16, the unilateral Wilcoxon test is: $H_0$: *"DYN added model MAE (or MSE) is not statistically lower than its `PRO` version"*, and $H_1$: *"DYN added model MAE (or MSE) is statistically lower than its `PRO` version"*. For RQ2 models, in Table 17, the unilateral Wilcoxon test is the same as in Section 4.2, conditioned by the setup.

*Table 16.* Detailed results of RQ1 `DYN` added models relative performance with setup conditioning against their `PRO` version with p-values $p_v$ (in **bold** if less than 0.05).

| Model | Informer | | | FiLM | | | FEDformer | | |
|---|---|---|---|---|---|---|---|---|---|
| | Better | Iso | Worse | Better | Iso | Worse | Better | Iso | Worse |
| DYN VS PRO | 89 | 65 | 46 | 54 | 75 | 71 | 119 | 18 | 63 |
| % | 45% | 32% | 23% | 27% | 38% | 35% | 60% | 9% | 31% |
| $p_{v,MSE}; p_{v,MAE}$ | **3.1**e−**3**; **9.1**e−**5** | | | 0.74; 0.83 | | | **9.1**e−**3**; **6.0**e−**3** | | |
| DYN VS PRO $\mid H > L$ | 77 | 13 | 28 | 23 | 13 | 18 | 77 | 6 | 37 |
| % | 65% | 11% | 24% | 43% | 24% | 33% | 64% | 5% | 31% |
| $p_{v,MSE}; p_{v,MAE}$ | **3.7**e−**4**; **1.4**e−**5** | | | 0.27; 0.60 | | | **2.7**e−**2**; **3.0**e−**2** | | |
| DYN VS PRO $\mid H = L$ | 0 | 44 | 0 | 0 | 30 | 0 | 25 | 6 | 21 |
| % | 0% | 100% | 0% | 0% | 100% | 0% | 48% | 12% | 40% |
| $p_{v,MSE}; p_{v,MAE}$ | − | | | − | | | 0.30; 8.1e−2 | | |
| DYN VS PRO $\mid H < L$ | 12 | 8 | 18 | 31 | 32 | 53 | 17 | 6 | 5 |
| % | 32% | 21% | 47% | 27% | 27% | 46% | 61% | 21% | 18% |
| $p_{v,MSE}; p_{v,MAE}$ | 0.77; 0.81 | | | 0.89; 0.85 | | | **2.0**e−**2**; 0.11 | | |

*Table 17.* Detailed results of RQ2 post-processing models relative performance with setup conditioning against their vanilla version with p-values $p_v$ (in **bold** if less than 0.05).

| Model | iTransformer | | | PatchTST | | | Crossformer | | |
|---|---|---|---|---|---|---|---|---|---|
| | Better | Iso | Worse | Better | Iso | Worse | Better | Iso | Worse |
| DYN-Post VS van. | 85 | 19 | 99 | 60 | 7 | 133 | 65 | 2 | 133 |
| % | 43% | 8% | 49% | 30% | 4% | 66% | 32% | 1% | 67% |
| $p_{v,MSE}; p_{v,MAE}$ | 0.18; **1.1**e−**2** | | | **1.0**e−**3**; **4.8**e−**5** | | | **7.9**e−**3**; **2.4**e−**4** | | |
| DYN-Post VS van. $\mid H > L$ | 32 | 5 | 21 | 18 | 3 | 33 | 24 | 1 | 31 |
| % | 55% | 9% | 36% | 33% | 6% | 61% | 43% | 2% | 55% |
| $p_{v,MSE}; p_{v,MAE}$ | 0.63; 0.66 | | | 0.15; **4.3**e−**2** | | | 0.46; 0.64 | | |
| DYN-Post VS van. $\mid H = L$ | 11 | 5 | 16 | 9 | 0 | 7 | 6 | 0 | 16 |
| % | 34% | 16% | 50% | 56% | 0% | 44% | 27% | 0% | 73% |
| $p_{v,MSE}; p_{v,MAE}$ | 0.30; 6.1e−2 | | | 0.77; 0.42 | | | 0.10; 0.12 | | |
| DYN-Post VS van. $\mid H < L$ | 42 | 6 | 62 | 33 | 4 | 93 | 35 | 0 | 87 |
| % | 38% | 6% | 56% | 25% | 3% | 72% | 29% | 0% | 71% |
| $p_{v,MSE}; p_{v,MAE}$ | 0.12; **4.2**e−**3** | | | **2.6**e−**4**; **3.1**e−**4** | | | **7.8**e−**3**; **2.0**e−**5** | | |

### G.4. Additional Results

We include here the setup conditioning for the comparison of RQ1 `DYN` added models against their vanilla versions. Distributions are shown in Figure 9, with detailed results (counts and p-values $p_v$) in Table 18. The performed unilateral Wilcoxon test is the same as in Section 4.2, conditioned by the setup.

We can observe that distributions are influenced by the setups but remain quite similar across each, without symmetry in performance for `DYN` models. It confirms the positive effect of the added learnable dynamics over the data length variation side-effect. In addition, there is a performance drop of `DYN` Informer, MICN, and FEDformer, for the $(H = L)$ setup, which are the three models where performance is driven by the added dynamics. As the `DYN` layer can be seen as a simple `PRO` feed-forward, it is less impactful. The drop is particularly pronounced for FEDformer, where the recomputation of $\mathbf{X}_{trunc}$ seems to harm the performance, confirming that it does not give any data length advantage to FEDformer `DYN`.

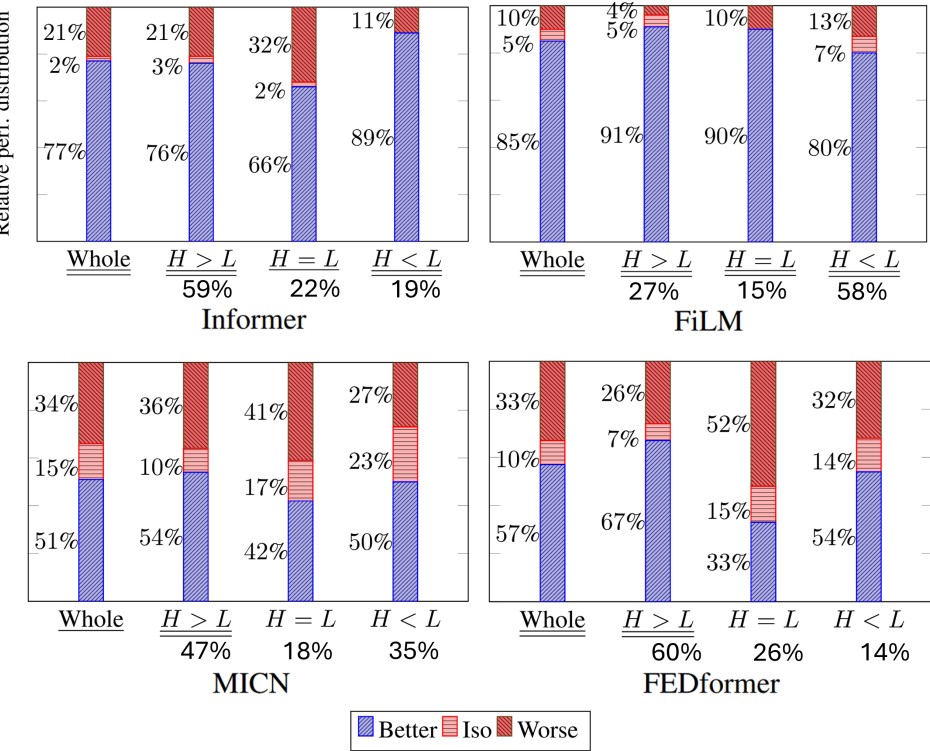

*Figure 9.* RQ1 `DYN` added models performance distribution against their vanilla version with setup conditioning. As in Figure 4, a setup is underlined (resp. double-underlined) when the `DYN` added model (left) is statistically better than its vanilla version on either MSE or MAE (resp. both). Setup distribution for each model is shown under the proper case.

.

*Table 18.* Detailed results of RQ1 `DYN` added models relative performance with setup conditioning against their vanilla version with p-values $p_v$ (in **bold** if less than 0.05).

| Model | Informer | | | FiLM | | | MICN | | | FEDformer | | |
|---|---|---|---|---|---|---|---|---|---|---|---|---|
| | Better | Iso | Worse | Better | Iso | Worse | Better | Iso | Worse | Better | Iso | Worse |
| DYN VS van. | 153 | 4 | 43 | 169 | 11 | 20 | 101 | 31 | 68 | 113 | 20 | 67 |
| % | 77% | 2% | 21% | 85% | 5% | 10% | 51% | 15% | 34% | 57% | 10% | 33% |
| $p_{v,MSE}; p_{v,MAE}$ | **4.0e−8; 9.7e−11** | | | **7.6e−11; 1.2e−11** | | | **1.3e−2; 6.5e−2** | | | **3.8e−3; 2.3e−2** | | |
| DYN VS van. $\mid H > L$ | 90 | 3 | 25 | 49 | 3 | 2 | 51 | 9 | 34 | 81 | 8 | 31 |
| % | 76% | 3% | 21% | 91% | 5% | 4% | 54% | 10% | 36% | 67% | 7% | 26% |
| $p_{v,MSE}; p_{v,MAE}$ | **3.7e−5; 8.4e−7** | | | **2.8e−5; 6.3e−6** | | | **4.9e−3; 4.9e−2** | | | **2.1e−4; 2.0e−3** | | |
| DYN VS van. $\mid H = L$ | 29 | 1 | 14 | 27 | 0 | 3 | 15 | 6 | 15 | 17 | 8 | 27 |
| % | 66% | 2% | 32% | 90% | 0% | 10% | 42% | 17% | 41% | 33% | 15% | 52% |
| $p_{v,MSE}; p_{v,MAE}$ | 5.6e−2; **9.9e−3** | | | **4.3e−4; 1.3e−3** | | | 0.24; 0.71 | | | 0.65; 0.73 | | |
| DYN VS van. $\mid H < L$ | 34 | 0 | 4 | 93 | 8 | 15 | 35 | 16 | 19 | 15 | 4 | 9 |
| % | 89% | 0% | 11% | 80% | 7% | 13% | 50% | 23% | 27% | 54% | 14% | 32% |
| $p_{v,MSE}; p_{v,MAE}$ | **5.9e−4; 1.3e−5** | | | **2.5e−5; 6.5e−6** | | | 0.38; 0.18 | | | 0.35; 0.36 | | |

# H. Prediction Visualizations

In this section, for each model, predictions on three different setups (dataset and horizon length) are shown, where the performance of RQ1 DYN added models and RQ2 post-processing models, in blue, are compared on the same sample to their vanilla version (on the left) and, for RQ1 models, to their PRO version (on the right), both in orange. The target prediction is in a dotted gray line. We chose two setups where the vanilla model performs well and one setup where it struggles more, to visualize different starting point situations. The dataset name and the horizon length $H$ are specified in the right column.

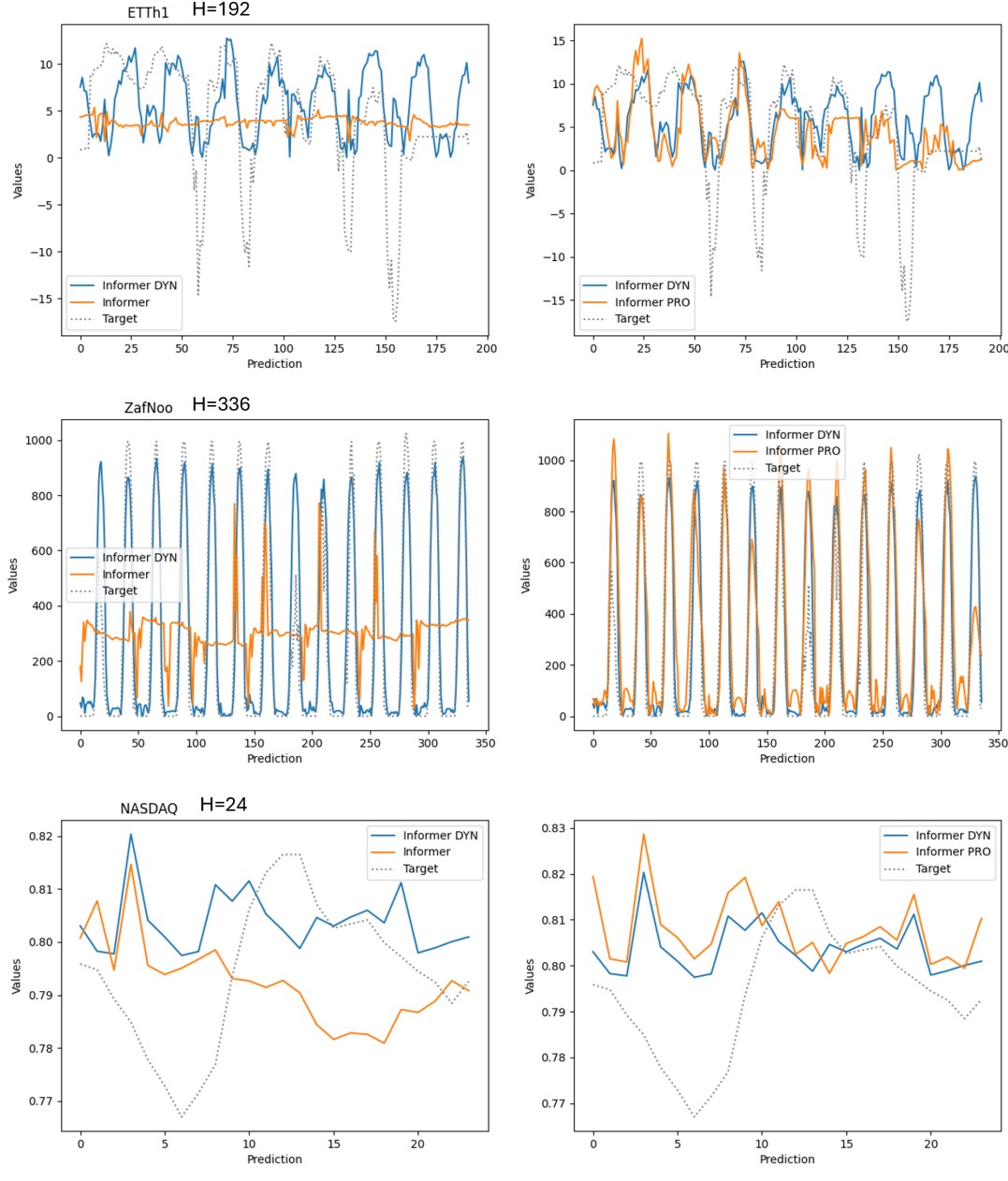

*Figure 10.* Informer prediction examples.

.

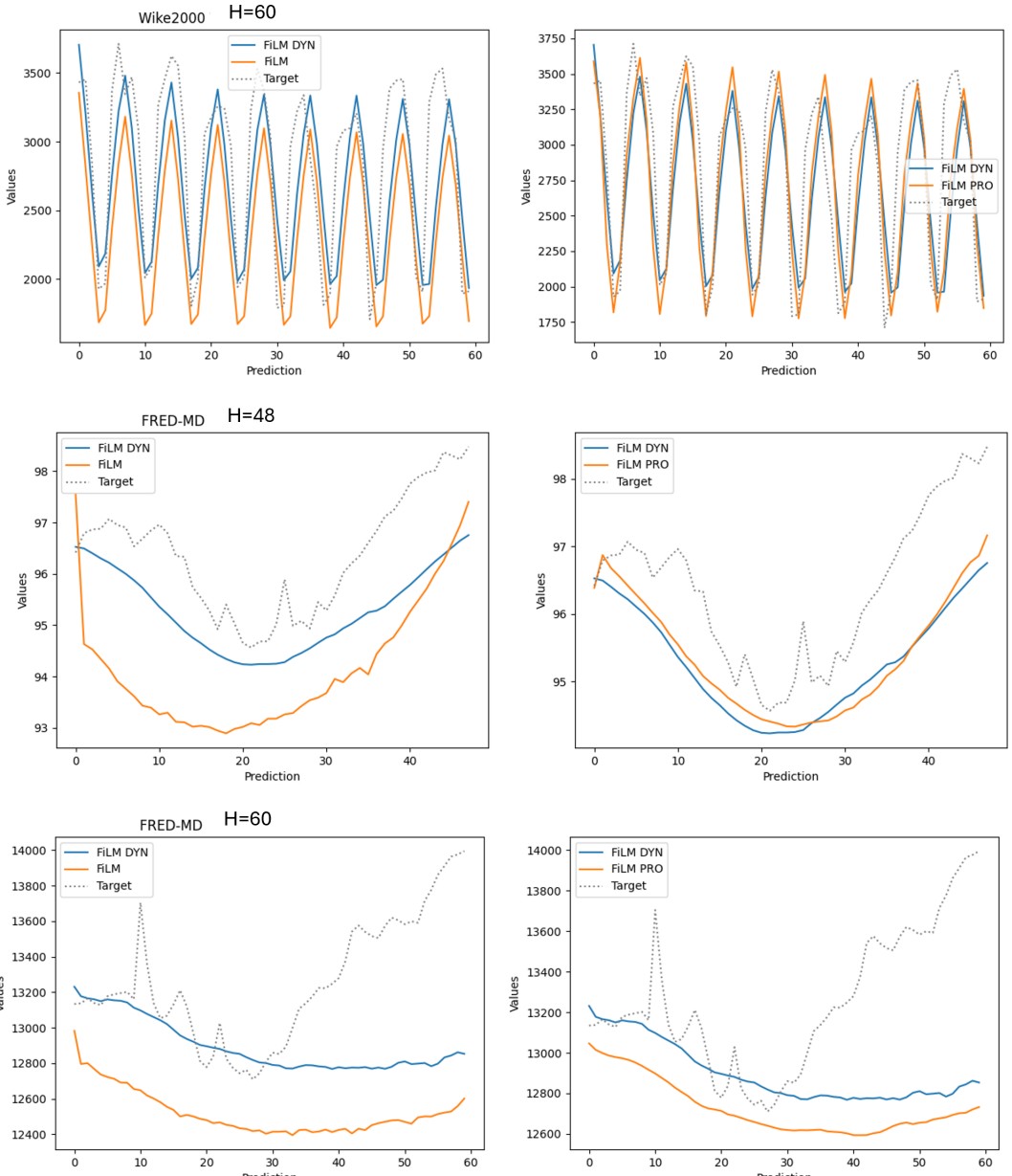

*Figure 11.* FiLM prediction examples.
.

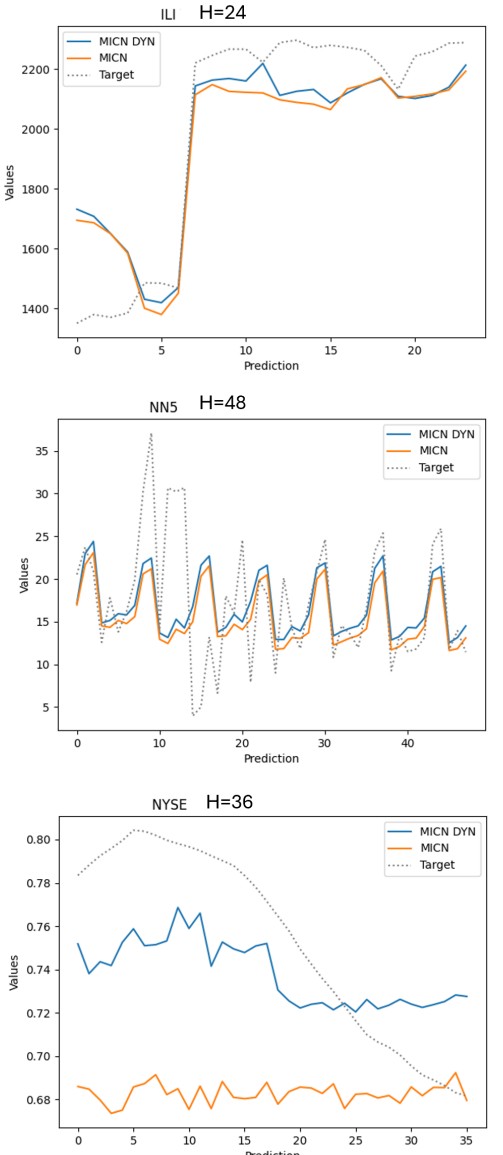

*Figure 12.* MICN prediction examples.

.

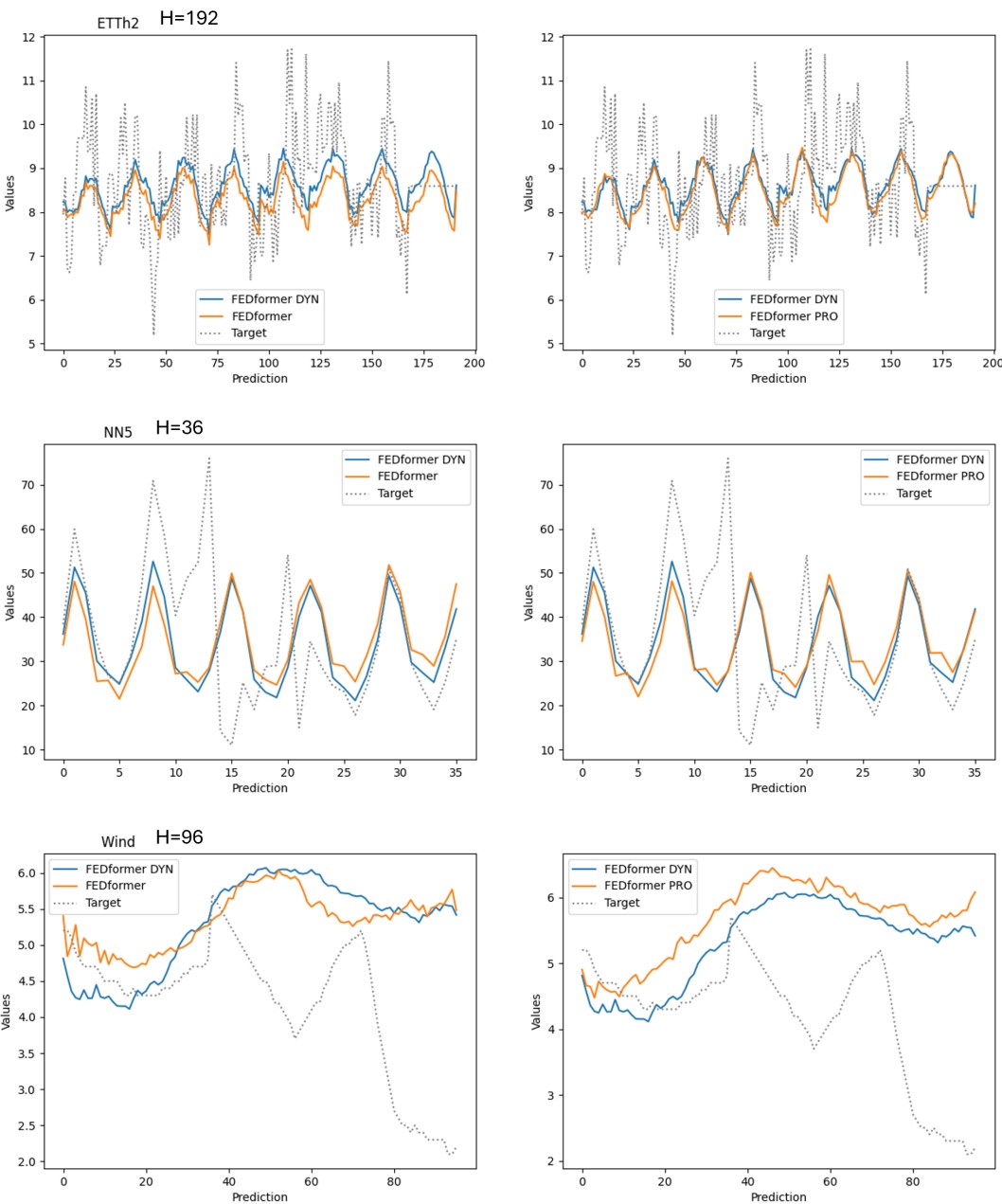

*Figure 13.* FEDformer prediction examples.
.

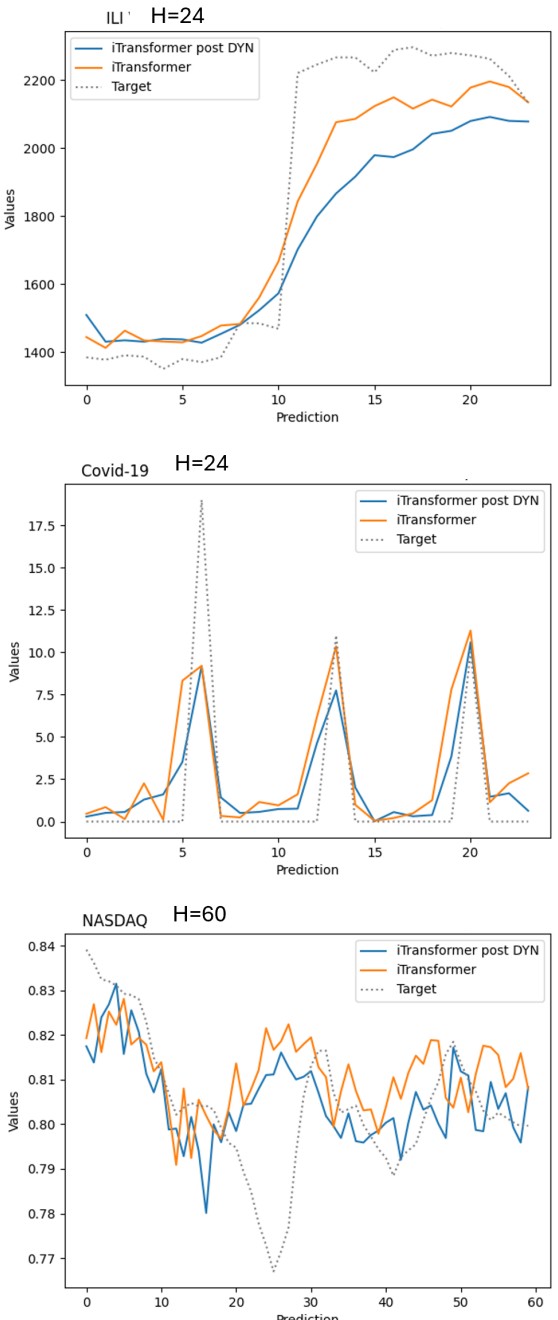

*Figure 14.* iTransformer prediction examples.

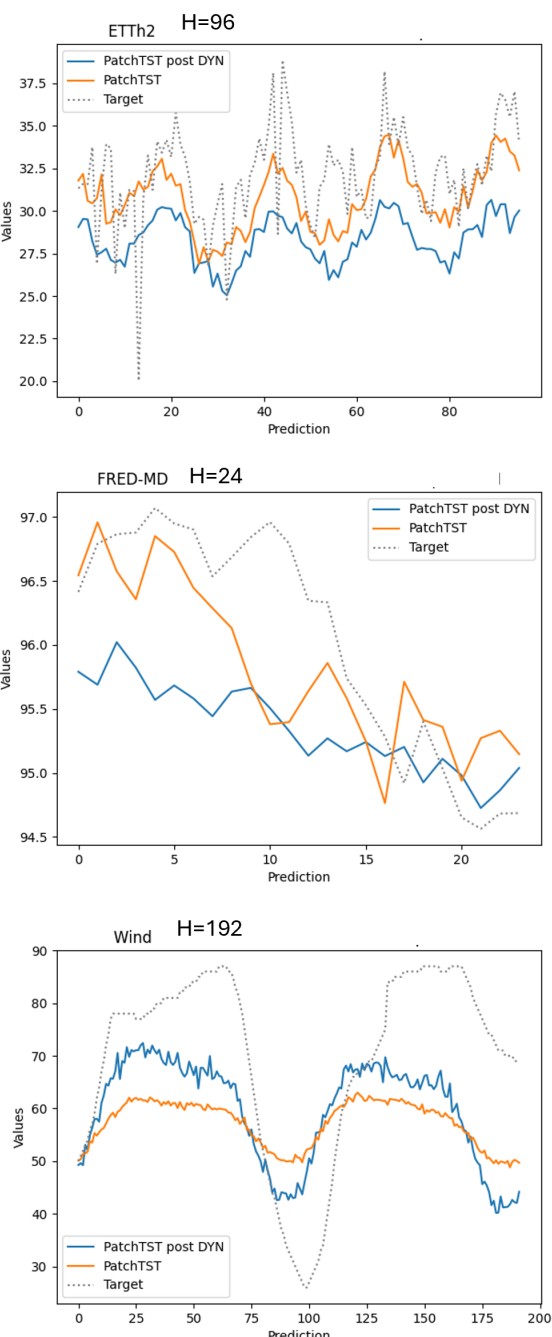

*Figure 15.* PatchTST prediction examples.

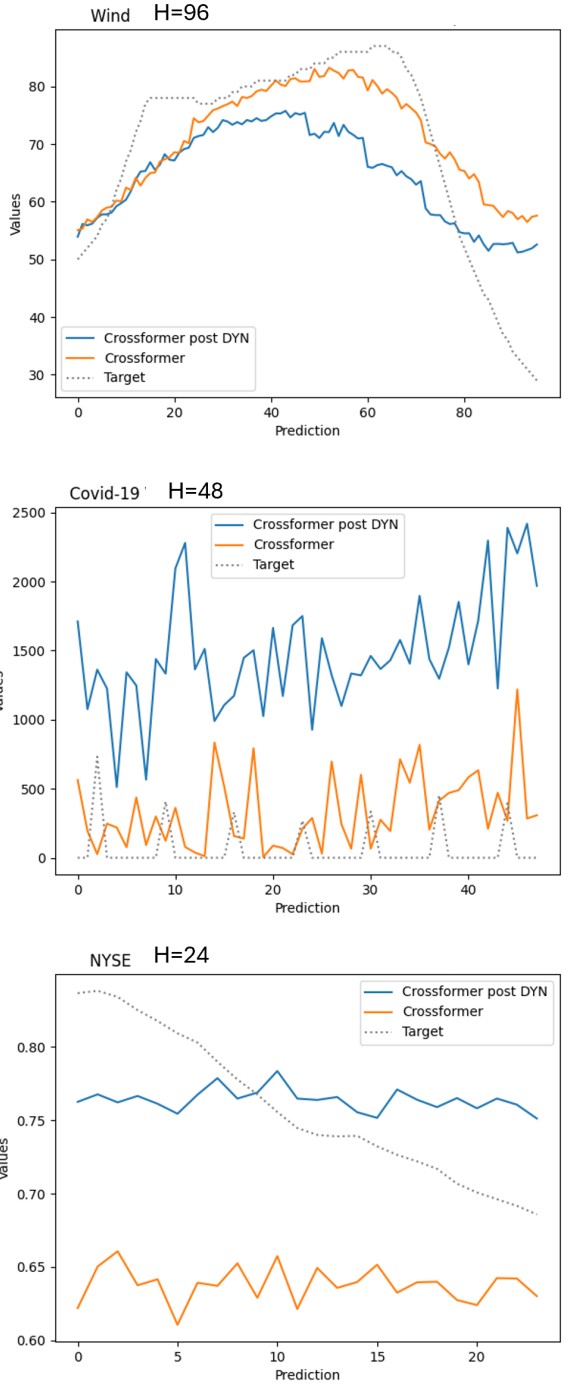

*Figure 16.* Crossformer prediction examples.

.

