# OpenReview forum: "Time Series Forecasting Through the Lens of Dynamics"
_ICML.cc/2026/Conference — ICML 2026 regular_

### Official Review · Reviewer_1CZk · 2026-03-05

**Soundness:** 3
**Presentation:** 2
**Significance:** 2
**Originality:** 2
**Overall Recommendation:** 4
**Confidence:** 4

**Summary:**

This paper performs an empirical study on the broader architectural design structure for modern time series foundation models. The authors roughly pose two core questions:
1. Does having a fully learnable dynamics module help?
2. Does it matter where the learnable dynamics module is relative to pre and post processing modules in the architecture?

The authors find through a set of forecasting experiments over a large selection of TSF models, that it indeed helps to have fully learnable dynamics modules and that it is mostly better to place the dynamics module towards the end of the architecture pipeline.

**Compliance With Llm Reviewing Policy:**

Affirmed.

**Final Justification:**

The authors effectively addressed my uncertainties and concerns in their rebuttal. My initial assessment of my confidence in the review was likely miscalibrated since I probably did not fully understand some of the nuances the authors were trying to convey in the work. Since the reported conclusions were experimentally validated, provide useful guidance to practitioners, and even seem to extend to the case of TSFMs (which is recently of practical interest in the community), I'm inclined to recommend an acceptance.

**Key Questions For Authors:**

#### Questions

1. What are the "PRO added" versions of the models for the parameter addition experiments in section 4.3? I understand the DYN added versions are the ones in fig. 2. Overall, the description of the models and setup are confusing to read.
2. How do these results explain the reported superior performance of diffusion TSF models without a learnable DYN module (e.g. DyDiff)?
3. Do you think these results will extend to time series foundation models (TSFMs)?

**Limitations:**

yes

**Strengths And Weaknesses:**

#### Strengths

1. The results suggest simple general principles over the large architecture design space for TSFMs that practitioners can rely on
2. The experiments are careful, cover a wide range of TSF models, and support the claims with statistical significance tests
3. The experiments (including vanilla baselines) are hyperparameter tuned

#### Weaknesses

1. The results appear to be limited to models that use linear fixed-length DYN functions and no modern autoregressive, or sampling based DYN functions - although this is acknowledged in the conclusion
2. Experiments only consider pointwise error metrics, rather than more practical ensemble/distributional metrics (for probabilistic forecasts) relevant to actual time series forecasting practice
3. The conclusions of RQ1 seem highly predictable - it is well known that baking in non-parametric strong inductive biases in deep learning does not work as well as learning from data

---

> ### Author Rebuttal · Authors · 2026-03-30
>
> We thank the reviewer for the detailed and constructive feedback and for recognizing the strengths of the simplicity and generality of our nomenclature and the experiments that validate our claims. In the following, we address the weaknesses and key questions sections.
>
> ## Weaknesses
>
> ### On the limitation to linear fixed-length DYN functions
> We agree that the present empirical study is restricted to linear DYN functions in family 1 models. The PRO-DYN nomenclature is intended to provide a general framework for TSF models, applicable to broader models with different backbones and DYN functions. Our current analysis to validate the PRO-DYN nomenclature focuses on family 1, as a first step, **to study pioneer deep TSF models, the transformer backbone, and the performance gap with simple linear models**. It emerges that linear DYN functions are sufficient to analyze the main architectures of family 1 and provide a minimal test case for the dynamics lens. Indeed, a linear layer is a general computational unit that maps two vectors with different dimensions and carries little explicit dynamical inductive bias, compared, for example, to autoregressive mechanisms. For this reason, the linear setting is a minimal and relatively unfavorable one for a dynamics-based interpretation, making the emergence of a dynamical bias in this minimal setting already a meaningful result.
>
> In addition, **RQ2 findings can extend to TSFMs models** (see the specific point on foundation models below).
>
> More broadly, we believe the PRO-DYN nomenclature may also help describe richer forms of temporal propagation: autoregressive / memory-based DYN functions may relate to partially observed dynamics, sampling-based ones may relate to stochastic dynamics. We discuss such cases in the appendix to illustrate how the framework may apply beyond the linear/family 1 setting. A proper comparison of such DYN functions would require a dedicated benchmark beyond the scope of this submission.
>
> ### On the point-wise metrics
> We agree. Our experiments follow the TFB benchmark, which uses point-wise metrics well-suited to the deterministic models we consider in our paper. Probabilistic or distributional metrics would indeed be valuable when studying family 2 and 3 models, especially for the analysis of richer DYN functions we plan to perform.
>
> ### On the predictability of RQ1
> We agree that, at a high level, one may expect learnable modules to outperform fixed placeholders. Our point is more specific: **our paper frames this as an architectural temporal propagation question** and validates it experimentally.
>
> This is precisely why in Section 4.3, we compare DYN-added models to their PRO-added counterparts. These comparisons help separate the effect of a learnable direct past-to-future propagation block from the effect of adding a learnable layer more generally. In particular, for Informer and FEDformer, the DYN versions perform better than the PRO versions with statistical significance, supporting the dynamical lens framing.
>
> ## Key questions
>
> ### Clarification on the PRO versions
> The PRO versions correspond to the DYN-added models, where the added linear DYN layer (direct past-to-future mapping) is replaced by a feed-forward PRO layer that does not itself advance the time interval, keeping the temporal dimension unchanged. **This comparison is meant to isolate the contribution of direct temporal propagation from parameter addition alone**. We will add an illustration of these PRO versions to the appendix to clarify this.
>
> ### On DyDiff
> In the paper introducing DyDiff, the TSF evaluation primarily compares it to the original DDPM, which follows the same overall PRO-DYN organization, and to 10 other models mostly introduced before 2020, across only 6 datasets and a single setup. In our view, a broader benchmark, covering more datasets, prediction horizons, and diffusion-based TSF models such as DYffusion, would be needed to assess this question more reliably; this is a direction we plan to explore in future work.
>
> We also note that the physics-inspired aspect of DyDiff mainly lies in its POST module rather than in the type of linear DYN. More generally, richer PRO functions informed by dynamical considerations deserve a dedicated analysis, especially in connection with family 3 models (e.g., Attraos or DeepEDM).
>
> ### On the extension to TSFMs
> As discussed in the appendix, several foundation models can also be described in a PRE-DYN configuration and still rely on a transformer-based backbone, similarly to the RQ2 models in our paper. This suggests that **the architectural question raised in the paper, especially RQ2, remains relevant beyond the specific family 1 benchmark**.

---

> > ### Author Rebuttal · Reviewer_1CZk · 2026-04-01
> >
> > I thank the authors for fully addressing my questions and concerns.
> >
> > I am convinced by the results and find that the empirical answers to the 2 proposed research questions presented in this work provide useful guidance in designing time series forecasting models that also seems to extend to the foundation model scale. Although more experiments are needed to investigate the same questions under probabilistic forecasts and distributional metrics.
> >
> > I am willing to raise my score to an accept.

---

> > > ### Author Response · Authors · 2026-04-05
> > >
> > > We sincerely thank the reviewer for their thoughtful follow-up and for raising their score. We are very pleased that our discussion addressed their concerns, and we appreciate the valuable suggestion to further investigate these questions under probabilistic forecasts and distributional metrics in future work.

---

### Official Review · Reviewer_bHWx · 2026-03-06

**Soundness:** 3
**Presentation:** 4
**Significance:** 3
**Originality:** 4
**Overall Recommendation:** 5
**Confidence:** 2

**Summary:**

The paper proposes a conceptual framework for analyzing time-series forecasting (TSF) models and argues that learnable dynamics modules are a key driver of forecasting performance.

**Compliance With Llm Reviewing Policy:**

Affirmed.

**Final Justification:**

I appreciate the authors' work on this manuscript. Unfortunately, I am not sufficiently familiar with some of the general time series forecasting literature.
To be responsible for the review process, I intend to retain my current confidence score.

**Key Questions For Authors:**

N/A

**Limitations:**

Yes.

**Strengths And Weaknesses:**

Strengths:
1. The proposed framework offers a new way to analyze how models propagate information from historical observations to future predictions. This helps explain why simple linear models remain competitive in TSF.
2. The paper is well structured and clearly written. The presentation of the framework, experimental setup, and analysis is easy to follow.
3. Experiments are comprehensive, where linear dynamics modules are inserted into various existing models (e.g., Informer, FEDformer) to validate the main claim.

Weaknesses:
The study mainly focuses on pattern-recognition TSF models (family 1), and it remains unclear whether the conclusions generalize to these broader classes of models. However, I consider this to be trivial and largely negligible.

---

> ### Author Rebuttal · Authors · 2026-03-30
>
> We thank the reviewer for the detailed and positive feedback, and for recognizing the strengths of our nomenclature, the overall perspective it offers, the clarity of the paper, and the comprehensiveness of our experiments. In the following, we address the question of generalization beyond family 1 models.
>
> The conclusions of the studied RQs in our paper are still connected to broader TSF model classes. In the appendix, we show that several family 2 (foundation) models share important architectural commonalities with the models studied in the main paper: they are still largely based on transformers, and many of them also follow a PRE-DYN structure. In that sense, **the RQ2 architectural insights of the paper remain relevant beyond the specific family 1 benchmark**.
>
> More broadly, the appendix also suggests how richer forms of dynamics can be described by the PRO-DYN nomenclature. For instance, **autoregressive / memory-based DYN functions may relate to partially observed dynamics**, while **sampling or diffusion-style DYN functions may relate to stochastic dynamics**.

---

> > ### Author Rebuttal · Reviewer_bHWx · 2026-04-01
> >
> > I thank the authors for the clarifications regarding the generalization beyond family 1 models.
> > The additional discussion on architectural commonalities and the extension of the framework to broader TSF paradigms is helpful.
> >
> > Overall, I remain positive about the paper's contributions and its potential impact.
> > However, I would like to reiterate that my confidence is limited, as noted in my confidence score.
> > To be honest, I am less familiar with some parts of the general TSF literature.
> > I would prefer to defer a more definitive judgment to other reviewers with deeper expertise in this area.

---

> > > ### Author Response · Authors · 2026-04-05
> > >
> > > We sincerely thank the reviewer for their positive evaluation and thoughtful comments. We are glad that the added discussion on the extension beyond family 1 models was helpful, and we appreciate their recognition of the paper’s contributions and potential impact.

---

### Official Review · Reviewer_kJwz · 2026-03-13

**Soundness:** 2
**Presentation:** 2
**Significance:** 2
**Originality:** 2
**Overall Recommendation:** 2
**Confidence:** 4

**Summary:**

The authors propose postulate that for forecasting a timeseries the underlying dynamics should be learned. The introduce a motivate their claim through linear dynamical systems.

**Compliance With Llm Reviewing Policy:**

Affirmed.

**Key Questions For Authors:**

cf. Strengths And Weaknesses

**Limitations:**

cf. Strengths And Weaknesses

**Strengths And Weaknesses:**

# Presentation
The paper introduces a PRO-DYN nomenclature, but does not list this nomenclauture in a single place.

# Originality
The approach in the paper seems novel, but not well motivated.

# Soundness
The underlying question, whether dynamical elements are necessary for predicting dynamical processes or whether unconstrained models are sufficient or even better suited is definately significant, in particular there are different classes of dynamical processes (deterministic, chaotic, partically observable, stochastic, smooth, ..., cf. [2]), which one would not expect to require the same modeling. The authors do not address this question, but focus on the question whether a dynamics block is helpful empirically and motivate this with linear systems theory, a theory that does not capture all of the properties listed above.

# Significance
The paper offers a shallow theoretical introduction and a classification based on whether models admit a dynamics component. This classification seems rather trivial. In terms of linear systems theory the "TSF task" does not require identifying the dynamics, c.f. ie [1].




[1] Markovsky, Ivan, and Florian Dörfler. "Behavioral systems theory in data-driven analysis, signal processing, and control." Annual Reviews in Control 52 (2021): 42-64.
[2] Lasota, Andrzej, and Michael C. Mackey. Chaos, fractals, and noise: stochastic aspects of dynamics. Vol. 97. Springer Science & Business Media, 2013.

---

> ### Author Rebuttal · Authors · 2026-03-30
>
> We thank the reviewer for the insightful comments. We address the reviewer's concerns below.
>
> ## Presentation
> We agree and will include a summary table gathering the main definitions and components of the PRO-DYN nomenclature to improve readability and flow in Section 3.
>
> ## Originality
> Our intent is to make explicit that TSF models can be interpreted through a temporal propagation perspective connected to dynamical formalisms. This perspective helps explain why such simple linear models remain competitive, an issue that remains insufficiently clarified in the TSF literature. Prior work typically motivates these models from a statistical or empirical standpoint, but does not explicitly frame them in terms of **how architectures propagate information over time**.
>
> Our contribution is therefore to propose a **common nomenclature** for analyzing TSF architectures based on their temporal propagation role, and to empirically validate it in the linear/family 1 setting.
>
> ## Soundness
> We agree with the reviewer that real-world dynamical processes can exhibit diverse properties (e.g., stochastic, partially observable, chaotic), and that these may require different modeling assumptions.
>
> Our work addresses a different question. **We do not aim to determine which class of dynamics is required to model a given process, nor to perform system identification**. Instead, **we study how neural architectures implement temporal propagation mechanisms, and whether architectural updates motivated by this perspective improve performance through RQ1 and RQ2**.
>
> Importantly, **we do not claim that linear systems theory captures all these regimes**. The PRO-DYN nomenclature is broader than the present empirical study; again, the paper first validates it in the linear/family 1 setting, where linear DYN functions are sufficient to characterize the main architectural differences highlighted by PRO-DYN.
>
> Our current analysis to validate the PRO-DYN nomenclature focuses on family 1, as a first step, **to study pioneer deep TSF models, the transformer backbone, and the performance gap with simple linear models**. It emerges that linear DYN functions are sufficient to analyze the main architectures of family 1 and provide a minimal test case for the dynamics lens. Indeed, a linear layer mapping two vectors with different dimensions is a general computation unit and carries little explicit dynamical inductive bias compared, for example, to autoregressive mechanisms. For this reason, the linear setting is a minimal and relatively unfavorable one for a dynamics-based interpretation, making the emergence of a dynamical bias in this minimal setting already a meaningful result.
>
> More broadly, we believe that PRO-DYN may also help to describe TSF models with richer forms of temporal propagation. In particular:
>
> * stochastic dynamics may relate to sampling-based or diffusion-style mechanisms,
> * partially observed dynamics may relate to autoregressive models with memory,
> * chaotic dynamics may involve representation-learning aspects that connect to processing blocks.
>
> We consider such mechanisms in the appendix to illustrate how the vocabulary may apply beyond the linear/family 1 setting and to motivate future work. A proper comparison of such DYN functions would require a dedicated benchmark beyond the scope of this submission.
>
> ## Significance
> Our claim is not that TSF requires identifying the true underlying dynamics of the data-generating process. Indeed, the connection to dynamical systems should be understood as a **descriptive architectural analogy: we highlight that architectural units implement specific forms of temporal propagation that can be interpreted through a dynamics lens**.
>
> In this sense, our approach is consistent with a broader trend in sequence modeling where dynamical structures help **organize computation** rather than recover true physical laws. For example, state-space models adopt the form of linear dynamical systems, but their parameters are not used to model a true physical LTI system; instead, they define computational mechanisms for storing and propagating information over time (e.g., HiPPO-like constructions).
>
> The reviewer points out that, from behavioral systems theory, the TSF task does not require identifying LTI systems; trajectory-level representations suffice to characterize them. Behavioral systems theory addresses what information suffices to represent trajectories; it does not prescribe how to structure a neural architecture to exploit that information efficiently. These questions are orthogonal, and our work operates at the architectural level.

---

> > ### Author Rebuttal · Reviewer_kJwz · 2026-04-04
> >
> > Dear authors, it seems like i misunderstood some of the claims. However, I still see little novelty in the ProDyn nomenclature, as the models mentioned above all explicitly discuss their modes of computation for storing information. I recognize, that computation is important to obtain efficient models. Yes, I still believe that the data-generating process is of prime importance when comparing forecasting models, as inductive biases that match data generation will outperform those that conflict with it.

---

> > > ### Author Response · Authors · 2026-04-07
> > >
> > > We thank the reviewer for this follow-up comment. It helped us clarify our claims and better position the paper. Below, we explain why we chose to conduct the study from this perspective and clarify the novelty of PRO-DYN.
> > > ## On the paper scope choice
> > > We first note an important **common ground** in your follow-up comment: forecasting models should benefit from inductive biases that align with the data-generating mechanism. **This high-level premise is precisely what guides our contribution**. Your last remark then raises a crucial question: if forecasting performance depends on such alignment, why does our paper not directly analyze the actual generating processes of the benchmark datasets? This is indeed an important question, but not the first one we choose to address in this paper.
> > >
> > > Our starting point is the following hypothesis: in time-series forecasting, the mechanism generating a series can be viewed through the lens of underlying dynamics. This hypothesis is motivated by related dynamics-based TSF works (family 3). For example, Koopa is built from a non-stationary dynamics perspective, while Attraos is motivated by a chaotic dynamical-system perspective. **These works differ in their specific assumptions, but they support a broader common idea: time-series generation can be modeled through a dynamics lens**. In this paper, our goal is to first examine whether this hypothesis can also help analyze TSF models developed without explicit dynamics considerations to explain current empirical results. Accordingly, our experiments are designed to isolate the role of DYN itself: we compare controlled variants of the same base architectures, varying only the nature and/or location of the DYN function (RQs 1 and 2), in order to study how models react when a dynamics bias is introduced, **as a first step before comparing broader full-model inductive biases against the data-generating process**.
> > >
> > > Once heterogeneous TSF models are described under a shared dynamics-based lens, whose relevance is also experimentally validated on non-dynamics-based models, they become much more directly comparable: one can ask where temporal propagation happens, which DYN function is used, and to what extent PRE blocks may transform the data into a representation that better connects the DYN function to the underlying data-generation mechanism. This is the intent of our paper. It does not yet determine whether such an alignment is achieved for each dataset, but **it provides a common dynamics-based framework needed to study how model architectures, DYN functions, and inductive biases match data generation**.
> > > ## On the novelty
> > > In this context, relating a linear prediction layer to a relaxed discrete linear time-delay dynamical system provides, to our knowledge, a novel dynamics-based interpretation of this simple computational unit, which is widely used in family 1 models. This opens the way to a broader study of generation mechanisms in TSF and of how different DYN functions may relate to them.
> > >
> > > To validate the dynamics lens on family 1 models, we study whether this inductive bias can help explain performance, in particular, why such a simple predictor can remain highly competitive. **Providing a dynamics-based explanation for the success of Linear predictors**, which had previously been mostly justified from empirical or statistical viewpoints, **supports the relevance of this lens for initially non-dynamics-informed TSF models**.
> > >
> > > This connection then directly shapes our research questions. In RQ1, **it justifies adding a linear DYN layer as a minimal way to endow models with an explicit learnable temporal propagation mechanism**. In RQ2, it provides a dynamics interpretation of the empirical superiority of the PRE-DYN configuration: under this lens, PRE modules can be understood as **learning a representation that facilitates the final temporal propagation performed by the DYN block**, here a linear one, even when the true underlying mechanism is not itself linear, while in DYN-POST, the POST block mainly aims to refine a prediction and is therefore less directly related to dynamics-based considerations. In that sense, **PRO-DYN provides a shared dynamics-based interpretation of TSF model modules beyond the model-specific motivations given in prior work**, making TSF models comparable through a common lens.
> > > ## On the revised version
> > > Following our discussion, we will revise the paper to make our positioning more explicit. In particular, we will clarify that:
> > > * this study is a first step toward considering different classes of data-generating processes and comparing TSF models within a shared dynamics-based framework;
> > > * our object of study is temporal propagation in TSF architectures, not system identification.
> > >
> > > We thank the reviewer again for this constructive discussion. It helped us better articulate both the scope and the novelty of PRO-DYN, thereby improving the clarity of our contribution.

---

### Official Review · Reviewer_K7kF · 2026-03-14

**Soundness:** 2
**Presentation:** 3
**Significance:** 2
**Originality:** 2
**Overall Recommendation:** 4
**Confidence:** 4

**Summary:**

The authors assume that TSF models must be capable of learning dynamics, and they observe that models propagate past observations to future predictions in a different way. They analyze that the reason complex Transformers underperform simple linear models is not due to attention, but because they fail to properly learn dynamics. Furthermore, the authors suggest that for a TSF model to perform well, a direct mapping (linear layer) from the past to the future should be placed at the end of the model.

**Compliance With Llm Reviewing Policy:**

Affirmed.

**Final Justification:**

Thank you for the detailed rebuttal responses. The rebuttal has partially addressed my concerns and I have revised my rating accordingly. Though I remain at the borderline between 3 and 4 recommendations, leaning toward 4 (weak accept), so I raise it to 4. At the same time, I raised the Soundness and Presentation scores (from 1 to 2 and from 2 to 3).

The authors have committed to concrete revisions, which I believe will meaningfully improve the paper's presentation/clarity.
That said, two concerns remain. First, the condition that "RQ2 applies only when the PRO unit is not dynamics-informed" was not present in the original paper and was introduced in the rebuttal without a clear plan for how it would be incorporated into the revision. Second, the concrete predictive advantages of connecting LTSF-Linear to dynamical systems theory have not been fully demonstrated.

Overall, I find the systematic empirical study across diverse architectures to be a good contribution. I increase my recommendation and encourage the authors to implement the promised revisions carefully.

**Key Questions For Authors:**

1. Is reinterpreting LTSF-Linear as a “relaxed discrete linear time-delay dynamical system” a simplistic interpretation? Does this interpretation offer any theoretical advantages for the TSF model?
2. The paper’s  claim is that learnable dynamics drive performance, yet all the DYN functions used in the experiments consist of linear layers. Does this mean the paper is actually demonstrating that the “ability to learn dynamics” is important, or simply that “linear direct mapping” is important?
3. The relations of Allen’s algebra seem to reduce to a binary classification of PRO and DYN. What additional explanatory power or predictive power does introducing this algebra provide?
4. The meaning of $M^H$ in Section 3.2 is not clearly defined.

minor comments:
- There is a typo on line 169: “connexions.”

**Limitations:**

yes

**Strengths And Weaknesses:**

### Strengths

- In terms of soundness, the authors’ experiments are comprehensive and have the advantage of including statistical significance tests via the Wilcoxon test.
- In terms of presentation, the authors present visualizations of the architectures in an intuitive manner to facilitate quick understanding by the reader. Furthermore, in Section 3.3, they use RQs to explain the setup of the two analyses they observe, ensuring readers can easily grasp the content.
- In terms of significance, the authors effectively classify the three families of TSF models, clearly outline their characteristics, and provide motivation for future research. Furthermore, the classification based on Allen’s algebra offers a new perspective. It reveals the commonalities and differences among models with distinct backbones from a fresh viewpoint.
- In terms of originality, the attempt to classify the time interval relationships of functions in TSF using Allen’s algebra could be a strength.

### Weaknesses

- In terms of soundness, the following weaknesses exist.
    - While the authors emphasize in the introduction the need to model dynamical systems in physics and economics, the DYN function they propose is simply a linear layer. While the authors strongly emphasize the connection to dynamical systems, their observations appear far from the content of the introduction.
    - The authors reinterpret LTSF-Linear as a “relaxed discrete linear time-delay dynamical system,” but it is unclear what theoretical advantages or contributions arise from this interpretation.
    - Family 2 (Foundation models) and Family 3 (dynamics-based models), mentioned in the introduction, appear to be presented in the Appendix only as examples of nomenclature application, with no experimental validation.
    - The authors state in Section 3.1 that they rely on Allen’s interval algebra, but it is difficult for readers to understand why they are using it. There is a lack of context in the introduction regarding its use, and the justification for why these temporal relations are necessary is insufficient.


- In terms of terminology, the following weaknesses exist:
    - In the paper, the term “dynamics” is used interchangeably to refer to several distinct concepts. Since it is not clearly distinguished which meaning is intended in each context, this may cause confusion for the reader. In my case, as I am familiar with the research of Raissi et al. and this field, I can grasp what the authors mean by “Dynamics” from the abstract onward; however, for time series researchers and readers unfamiliar with this field, the definition of “dynamics” seems too simplistic, which may make it difficult to understand.
    - In Figure 1, there is a lack of consistency because the notation $\mathcal{M}_\theta$ in the caption does not match the notation used in the figure.
    - When reading through Sections 3.1, 3.2, and 3.3 in Section 3, the subsections tend not to flow well together, which reduces readability.


- In terms of significance, the paper has the following weaknesses:
    - Considering the foundation models in “Family 2” mentioned by the authors, the “Family 1” models primarily analyzed in this paper are already becoming less of a practical option, which means the paper’s practical impact may diminish over time.
    - There is a significant gap between the theoretical content and the practical implications of the authors’ message. The PRO-DYN nomenclature and Allen’s algebra appear to function only as theoretical frameworks designed to justify the simple empirical observation that “the linear layer should be placed at the end,” which is a major weakness of this paper.

- In terms of originality, the following weaknesses exist:
    - The authors’ PRO-DYN nomenclature attempts to interpret LTSF-Linear as a linear system, but it appears to formalize the existing community’s understanding rather than representing a new discovery, and there are limitations to viewing this interpretation as having yielded theoretical advantages.
    - While presenting a “dynamics perspective,” the proposed method lacks originality. It is unclear whether the “dynamics lens” implied by the paper’s title is actually evident in the main text.

---

> ### Author Rebuttal · Authors · 2026-03-30
>
> We thank the reviewer for their detailed comments. In the following, we address the reviewer's concerns.
> ## Soundness
> ### On the connection to dynamical systems and the use of linear DYN
> We agree that the introduction may suggest a broader scope than the empirical study in the main paper, and we will clarify this point. The broader claim concerns the PRO-DYN nomenclature itself: it provides a **general framework to characterize TSF computing units by how they propagate information over time, and is therefore not restricted to linear dynamics**. The empirical validation focuses on linear DYN functions because they are sufficient to analyze the main architectures of family 1 and provide a minimal test case for the dynamics lens. A linear layer mapping two vectors with different dimensions is a general computation unit and carries little explicit dynamical inductive bias compared, for instance, to autoregressive mechanisms. For this reason, the linear setting is a **minimal and relatively unfavorable one for a dynamics-based interpretation**, making the emergence of a dynamical bias in this setting already a meaningful result. The framework is thus broader than the present empirical study; the paper first validates it in the linear/family 1 setting, where linear DYN functions are sufficient to characterize the main architectural differences highlighted by PRO-DYN.
>
> Richer DYN functions can also arise: some autoregressive processes with memory may relate to partially observed dynamics, and sampling methods to stochastic dynamics. We view these as ongoing extensions of PRO-DYN.
> ### On the interpretation of LTSF-Linear
> We explicit an interpretive link between computing blocks in TSF and dynamical formalisms. This perspective helps explain why such simple models remain competitive. Importantly, prior works motivate the use of linear layers for prediction from a statistical or empirical standpoint, but do not explicitly link them to dynamical systems.
> ### On Family 2 & 3
> We state in the paper that these families are not experimentally explored. The broader claim concerns the PRO-DYN nomenclature, which can also be applied to TSF models with richer temporal propagation, as illustrated in the appendix. However, the empirical validation in this paper is restricted to family 1, as a first step, **to study pioneer deep TSF models, the transformer backbone, and the performance gap with simple linear models**. The empirical extension to families 2 and 3 is beyond the present scope and is left for future work. In particular, comparing the DYN functions of families 1 and 3 would require a dedicated benchmark. We will clarify this positioning more explicitly.
> ### On the use of Allen’s interval algebra
> Allen’s algebra is a formal and comprehensible language to describe temporal relationships between inputs and outputs of computational blocks. It provides a **consistent and rigorous way to characterize how models manipulate temporal information**. Allen’s algebra provides a precise formal criterion for the PRO/DYN distinction by specifying whether a computational block preserves a temporal interval or advances it.
> ## Terminology
> ### On *dynamics* terminology
> Thanks for pointing this out. We agree and will explicitly distinguish between model dynamics (how a model propagates temporal information) and physical dynamics (physical evolution laws), and we will add a precise definition of it in the paper.
> ### On presentation
> We will correct Figure 1, improve the flow of Section 3 (grouping 3.1 and 3.3), clarify missing definitions (e.g., the meaning of $M^H$ as the repeated application of the matrix $M$), and fix the reported typo.
> ## Significance
> The field is indeed evolving toward foundation models. While our empirical study is restricted to family 1, we do not believe this makes the contribution obsolete, since we study **architectural temporal propagation mechanisms** rather than a single model generation. In particular, several recent TSF models, including those in family 2, still follow transformer-based and PRE-DYN configuration studied in the paper, so **the practical message concerns design principles that are relevant beyond the considered models**.
>
> We agree that the contribution should not be framed as a theoretical wrapper around the observation that *a linear layer at the end helps*. Rather, the PRO-DYN decomposition turns this observation into a **predictable and testable architectural question** across several architectures, which we view as a contribution in itself.
> ## Originality
> Our claim is not that dynamical ideas are new in time-series modeling, but that we make them **explicit at the architectural level**. PRO-DYN decomposition analyzes TSF computing blocks according to their temporal-propagation role, a framework that, to our knowledge, has not been explicitly applied to modern TSF architectures under a common lens.
> ## Key questions
> Please refer to the above developments.

---

> > ### Author Rebuttal · Reviewer_K7kF · 2026-04-02
> >
> > Thank you for the rebuttal. I list my remaining questions and suggestions below:
> >
> > ### On the connection to dynamical systems and the use of linear DYN
> > It would have been helpful if the paper had made the research scope more explicit. In particular, the argument that "a linear setting is a minimal and relatively unfavorable setting for a dynamics-based interpretation, ..." is a good point. Still, this framing does not appear clearly in the current paper. I would encourage the authors to make this argument more explicit in the revised version.
> >
> > ### On the interpretation of LTSF-Linear
> > Thank you for the response. However, I would like to ask further: what new advantages arise specifically from connecting LTSF-Linear to the dynamical systems concept? Before this connection, what could not be predicted can now be predicted?
> >
> > ### On Family 2 & 3
> > Thank you for clarifying the scope. Dynamics-based models in Family 3 (e.g., Koopa, Attraos) adopt a PRE-DYN or PRE-DYN-POST configuration and achieve competitive performance. Could the authors provide a more detailed explanation of how this observation relates to the conclusion of RQ2, which argues for the superiority of the PRE-DYN configuration?
> >
> > ### On the use of Allen's interval algebra
> > I understand the use of Allen's interval algebra. However, since this framework may be unfamiliar to many readers in the machine learning community, I would suggest that the authors add at least a footnote that explains the rationale for adopting Allen's algebra in this context, similar to the clarification provided in the rebuttal.
> >
> > ### On dynamics terminology and presentation
> > I have read the authors' responses regarding the clarification of "dynamics" terminology and the correction of notation.
> >
> > ### On significance
> > I do not consider the restriction to Family 1 models to make the contribution obsolete. However, I believe there is a writing-level weakness in the current manuscript. Since all three families (1, 2, and 3) are introduced and discussed,  readers will naturally develop curiosity about Families 2 and 3.  I would suggest that the authors represent the positioning more explicitly. For instance, by arguing, as was done in the rebuttal, that Family 1 architectures and the PRE-DYN configuration are a foundational setting that underlies the temporal propagation mechanisms of more complex families as well, and that this paper targets this foundational layer as a first step. Such representation would make the scoping decision feel principled rather than limiting.
> >
> > ### On originality
> > I would suggest that the distinction between the proposed PRO-DYN framework and prior work that also applies dynamics concepts at the architectural level be made more explicit. In particular, works such as Family 3 models (e.g., Koopa, Attraos) and related research that analyzes time series from a dynamics perspective. For example, Hu et al.,  already operate at the intersection of dynamics and architecture design.
> >
> > > Hu, Jiaxi, et al. "Toward physics-guided time series embedding." arXiv preprint arXiv:2410.06651 (2024).

---

> > > ### Author Response · Authors · 2026-04-03
> > >
> > > We thank the reviewer for this follow-up response, which offers an insightful discussion that improves our work. We address below the follow-up questions and concerns:
> > > ## On the revised version
> > > According to our discussion and your comments, in the revised version, we will:
> > > * explicit the framing on the linear setting as being minimal,
> > > * explain the rationale for adopting Allen's algebra for our nomenclature,
> > > * include the distinction of model and physical dynamics, and the reported details on presentation,
> > > * explicit our positioning about the foundational setting of family 1 and the PRE-DYN configuration study,
> > > * clarify our positioning with dynamics-based related work (see below).
> > > ## On the interpretation of LTSF-Linear
> > > Our theoretical contribution is to provide a dynamics explanation and interpretation for this general computational unit. Before this connection, the strong performance of linear layers as predictors could be empirically and statistically supported, but **it was not justified from a dynamics-based perspective**.
> > >
> > > Indeed, based on the hypotheses stated in the paper that (a) a generative model should reproduce the mechanism generating the data it processes, and (b) this mechanism is viewed as an underlying temporal dynamics (with family 3 works supporting (b)), **by showing that a linear layer implements a simple learnable evolution mechanism, it explains why such models can remain highly competitive despite their simplicity**. This connection then motivates our choice in RQ1 of a linear layer to add a learnable DYN function from a dynamics perspective. Regarding RQ2, we use it to interpret the empirical result: **in a PRE-DYN configuration, the PRE block serves a dynamics purpose**, as it can be understood as learning a representation in which the temporal propagation becomes easier for a linear DYN function. This provides a possible explanation for why PRE-DYN performs better experimentally in our setting, whereas in DYN-POST, the POST block aims to refine a prediction towards a target, thus it is less related to dynamics-based considerations. We will add this point in the revised paper.
> > > ## On Family 2&3
> > > Our paper focuses on the nature and placement of the DYN layer in forecasting architectures whose PROcessing units are not explicitly dynamics-informed. In this setting, RQ2 experimentally shows that a PRE-DYN configuration achieves better performance. Family 3 models operate in a different regime, where PRE and/or POST units are themselves motivated by dynamics considerations (e.g., Attraos or DeepEDM based on Takens representation theorem), and therefore do not fall under the scope of RQ2. As a result, **we do not view well-performing PRE-DYN-POST models from family 3 as contradicting our findings, but rather as going beyond the specific setting studied in our paper**.
> > >
> > > This distinction is also consistent with [1], which introduces an embedding PRE function motivated by dynamics considerations. Their results suggest that improving the representation through a dynamics-informed PRE unit can be particularly beneficial for forecasting (experiments are also conducted on PatchTST). In addition, in PRE-DYN-POST family 3 models where the DYN function maps data within the same latent space, the DYN block can become more interpretable/identifiable, as in Koopman-inspired approaches discussed in our appendix.
> > > ## On originality
> > > As mentioned above, works such as Koopa or Attraos getting superior performance from dynamics-informed design choices support hypothesis (b): **such works motivate the development of PRO-DYN**. While they propose a specific model, we aim to analyze, compare, and reinterpret diverse TSF architectures under a common dynamics lens.
> > >
> > > We also find [1] highly relevant and complementary. They introduce their dynamics-informed embedding module within an Embedding–Encoder–Decoder formulation. **Their structural choice of developing a PRE embedding function to improve performance is consistent with our RQ2 findings**; their design and results support the same direction: for forecasting (and imputation), improving the representation before the temporal propagation step appears particularly effective. **Their strongest gains are reported in generative-related settings, which align with our hypothesis about time-series generative models** (see point above). In addition, the improvements from a dynamics-informed PRO unit give insights for ongoing work on family 3 models (see previous section).
> > >
> > > Overall, our paper and [1] propose **complementary rationales for dynamics considerations in the TSF task, while getting similar tendencies on the generative tasks**. We will add this paper to the related work section, refer to it to support why we focus on the forecasting task, and clarify its connection with family 3 models in the appendix.
> > >
> > > [1]Hu et al. *Toward physics-guided time series embedding* (2024).

---

### Decision · Program_Chairs · 2026-04-30

**Decision:**

Accept (regular)

**Comment:**

This paper studies time-series forecasting from a dynamics perspective and presents a simple yet effective framework grounded in linear dynamical assumptions. Reviewers raised concerns on novelty and positioning, but also recognized the empirical rigor and consistent improvements across benchmarks. The rebuttal satisfactorily clarified the motivation and strengthened the connection to prior work. Overall, while incremental, the contribution is technically sound and practically relevant. I recommend acceptance.